# A dicarbonate solvent electrolyte for high performance 5 V-Class Lithium-based batteries

Xiaozhe Zhang[1,4], Pan Xu[2,4], Jianing Duan[2,4], Xiaodong Lin [1,4] ✉, Juanjuan Sun[2], Wenjie Shi[3], Hewei Xu[1], Wenjie Dou[2], Qingyi Zheng[2], Ruming Yuan[2], Jiande Wang[1], Yan Zhang[1], Shanshan Yu[2], Zehan Chen[1], Mingsen Zheng [2], Jean-François Gohy[1], Quanfeng Dong [2] ✉ & Alexandru Vlad [1] ✉

Rechargeable lithium batteries using 5 V positive electrode materials can deliver considerably higher energy density as compared to state-of-the-art lithium-ion batteries. However, their development remains plagued by the lack of electrolytes with concurrent anodic stability and Li metal compatibility. Here we report a new electrolyte based on dimethyl 2,5-dioxahexanedioate solvent for 5 V-class batteries. Benefiting from the particular chemical structure, weak interaction with lithium cation and resultant peculiar solvation structure, the resulting electrolyte not only enables stable, dendrite-free lithium plating-stripping, but also displays anodic stability up to 5.2 V (vs. Li/Li$^+$), in additive or co-solvent-free formulation, and at low salt concentration of 1 M. Consequently, the Li||LiNi$_{0.5}$Mn$_{1.5}$O$_4$ cells using the 1 M LiPF$_6$ in 2,5-dioxahexanedioate based electrolyte retain >97% of the initial capacity after 250 cycles, outperforming the conventional carbonate-based electrolyte formulations, making this, and potentially other dicarbonate solvents promising for future Lithium-based battery practical explorations.

Since their inception, lithium-ion batteries (LIBs) have attracted tremendous attention due to their high energy content and versatility, achieving unprecedent commercial success in the portable power supply market after decades of rapid development, and are currently serious candidates for implementation in large-scale applications such as electric vehicles and residential storage[1]. However, the energy density of current commercial LIBs in the range of 300 Wh kg$^{-1}$, cannot satisfy the application requirements (>500 Wh kg$^{-1}$) of the targeted large-scale energy storage devices[2,3]. To meet the future high energy density demands, a pragmatic and effective approach is to increase the working voltage of batteries[4,5]. High-voltage lithium metal batteries (LMBs), that employ high-voltage materials as positive and metallic lithium as negative electrode materials, are one such key technology that can supply the high energy density requested for large-scale energy storage devices[4,6]. Nevertheless, numerous fundamental challenges, mainly resulting from the high reactivity of both, the lithium metal and the high potential of the positive electrode materials to electrolyte, still hinder the development and practical application of these rechargeable high-voltage LMBs[6,7].

[1]Institute of Condensed Matter and Nanosciences, Molecular Chemistry, Materials and Catalysis, Université Catholique de Louvain, Louvain-la-Neuve B-1348, Belgium. [2]Collaborative Innovation Center of Chemistry for Energy Materials (iChEM), State Key Laboratory of Physical Chemistry of Solid Surfaces, Department of Chemistry, College of Chemistry and Chemical Engineering, Engineering Research Centre of Electrochemical Technologies of Ministry of Education, Innovation Laboratory for Sciences and Technologies of Energy Materials of Fujian Province (IKKEM), Xiamen University, Xiamen 361005, China. [3]Institute for New Energy Materials & Low Carbon Technologies, School of Material Science & Engineering, School of Chemistry & Chemical Engineering, Tianjin University of Technology, Tianjin 300384, China. [4]These authors contributed equally: Xiaozhe Zhang, Pan Xu, Jianing Duan, Xiaodong Lin. ✉e-mail: xiaodong.lin@uclouvain.be; qfdong@xmu.edu.cn; alexandru.vlad@uclouvain.be

On one side, the irreversible reactions between lithium metal and electrolyte will consume the active lithium and electrolyte, resulting in low Coulombic efficiency (CE) and short cycle life of LMBs[8,9]. Although a passivation layer called solid-electrolyte interphase (SEI) can be generated at the surface of lithium metal, its morphology and composition is generally inhomogeneous, which will induce the growth of lithium dendrites and thus degrade the cycle life and safety of LMBs[9,10]. On the other side, due to the strong oxidizing conditions of high operating voltage, electrolyte decomposition will occur at the positive electrode-electrolyte interface, further resulting in the consumption of the electrolyte[6,11]. Although the electrolyte oxidation can also lead to the formation of a cathode-electrolyte interphase (CEI), the quality of CEI generated from most used electrolytes thus far is generally not sufficient to stabilize the electrolyte against continuous oxidation at high voltage[12].

Among all the electrolyte systems, ether-based electrolytes are the most promising in LMBs because of their good interface stability with lithium metal and high efficiencies attained for Li-metal plating/stripping[13–16]. However, common ether-based electrolytes have a low anodic stability (<4.0 V vs. Li/Li$^+$) making these incompatible with most current 4 V-class positive electrode chemistries, restricted thus to mainly LiFePO$_4$ based cells[17]. By contrast, carbonate solvents exhibit much better anodic stability (4.5 V vs. Li/Li$^+$) and have been successfully applied in commercial LIBs. The drawback however is that the carbonates are poorly compatible with lithium metal, showing low lithium plating/stripping CE (<80% without additives/co-solvents) and uncontrollable lithium dendrite growth, arising from the inherent reactivity of the carbonyl group at low reduction potentials[6,16]. Additionally, when targeting cell voltages higher than 4.5 V, the carbonate-based electrolytes will undergo anodic decomposition to release $CO_2$ and $O_2$[18], which generally hinders their application to 5 V-class positive electrode chemistries (e.g., LiNi$_{0.5}$Mn$_{1.5}$O$_4$, LiCoPO$_4$). Hence, the cells based on 5 V-class positive electrodes and carbonate solvent electrolytes generally show low CE and fast capacity decay[6,19,20]. Recently, highly concentrated electrolytes have been proposed to overcome these issues and significantly improvements have been attained[21–23]. However, the high salt concentration comes with the increased manufacture costs and high viscosity of the electrolyte. The latter one will lead to low wettability of electrodes and separators, as well as low ionic conductivity resulting in poor rate performance and low utilization of active materials[24,25]. Thus, exploiting suitable and efficient electrolyte systems is the top priority for the development of high-voltage LMBs.

In this work, we report a dicarbonate-based electrolyte consisting of 1 M lithium hexafluorophosphate salt (LiPF$_6$) in a dimethyl 2,5-dioxahexanedioate (DMDOHD) solvent, which shows enhanced stability towards both challenges: the highly reactive lithium metal negative electrode and high-anodic stability. Different from the previously reported carbonate solvent based electrolytes, which display low lithium plating/stripping CE (<80%) and severe dendritic growth in the absence of additives or co-solvents, the use of DMDOHD solvent allows to achieve a relatively high lithium plating/stripping CE of 92% while suppressing lithium dendrite growth even without any additives or co-solvents, which represents a significant advance in the use of carbonate solvent based electrolytes for LMBs (Supplementary Table 1). This behavior is attributed to the ability of 1 M LiPF$_6$ – DMDOHD electrolyte formulation to favor the deposition of a robust SEI with homogeneous inorganic-organic mixed components. Additionally, the 1 M LiPF$_6$ – DMDOHD formulation is also found highly compatible with the high-voltage LiNi$_{0.5}$Mn$_{1.5}$O$_4$ positive electrode material, with excellent room-temperature cycling stability of 250 cycles (with a high-capacity retention of ~94%) at a rate of C/3 (1 C = 142 mA g$^{-1}$) attained, that is better than most previously reported carbonate-based electrolyte formulations (Supplementary Table 2). Again, the improvements are explained by effectively suppressing the electrolyte oxidation through the formation of a durable, conformal and dense CEI with a thickness

of ~5 nm. Further improvements are attained by adding fluoroethylene carbonate (FEC) additive to the baseline 1 M LiPF$_6$ – DMDOHD formulation, wherein the lithium plating/stripping CE and the capacity retention of Li||LiNi$_{0.5}$Mn$_{1.5}$O$_4$ cell rise to ~98% and ~97%, respectively, under similar testing conditions, which again outperforms most of the previously reported carbonate-based electrolytes containing additives or co-solvents (Supplementary Tables 3 and 4). It has to be mentioned that the use of DMDOHD as electrolyte additive (in a baseline 1 M LiPF$_6$ – EC/DMC, 1/1 by vol. electrolyte formulation) has been reported by T. Hanemann et al. and found to improve the performances of graphite||LiNi$_{0.5}$Mn$_{1.5}$O$_4$ cells. The bulk properties of the baseline 1 M LiPF$_6$ – EC/DMC electrolyte not being significantly affected (i.e., flammability and safety), the use of DMDOHD pointed towards the effective film forming behavior of this particular chemistry[26,27]. Finally, in the final preparation stages of this manuscript, J. Dahn and his team reported on the use of DMDOHD electrolyte solvent with LiFSI salt for 4 V class Li-ion cells such as graphite||LiFePO$_4$, graphite||Li[Ni$_{0.5}$Mn$_{0.3}$Co$_{0.2}$]O$_2$, and graphite||Li[Ni$_{0.83}$Mn$_{0.06}$Co$_{0.11}$]O$_2$[27]. Through electrochemical tests and analysis, they found that the cells with DMDOHD-based electrolyte can attain a significantly improved capacity retention with minimal gas generation under low-voltage (<4 V) and high temperature (70 and 85 °C) operation as compared to EC-based electrolytes, demonstrating the advantage of this new electrolyte solvent. Compared with their studies, while further confirming the suitability of this electrolyte formulation for Li-ion cells, our work provides additional fundamental differences and insight: (1) Different electrode chemistries: Li-metal vs. graphite negative, and 5 V-class (vs. 4 V-class) positive electrode materials; (2) Different lithium salt system: LiPF$_6$ (vs. LiFSI) that could also affect the Al corrosion at high anodic potentials; (3) Different working conditions: room temperature (vs. 70/85 °C) and high voltage up to 5 V (vs. 3.8 V). Moreover, our work here also explores and discloses the underlying improving mechanism of DMDOHD, which was not discussed in the previous developments in their work.

## Results

### Physical properties and anodic stability of the 1 M LiPF$_6$ – DMDOHD baseline electrolyte formulation

The state-of-the-art electrolytes used in commercial LIBs consist of low concentration lithium salts dissolved in mixed solvents of linear and cyclic carbonates, such as 1.3 M LiPF$_6$ in ethylene carbonate/dimethyl carbonate (EC/DMC). The reason why commercial LIBs use this electrolyte formulation as well as the disadvantages associated are discussed in the Supplementary Note 1. In this work, the choice of DMDOHD as electrolyte solvent to replace the conventional mixed linear/cyclic carbonate solvents such as EC/DMC, thereby ameliorating their performances for high-voltage LMBs, was based on the following considerations (Fig. 1 and Supplementary Fig. 1).

DMDOHD was found to be a by-product of EC/DMC-based electrolyte system, formed from via the transesterification reaction between DMC and ring-opening product of EC (Supplementary Fig. 1)[28,29]. Therefore, DMDOHD structurally integrates the features of EC and DMC, and also shares similar structural features with other linear carbonates such as ethyl methyl carbonate (EMC, Fig. 1). Generally, the structural similarity will also lead to the similarity in physicochemical properties. Thus, we speculate that DMDOHD may integrate the SEI film forming ability of EC[30], the high anodic stability of EMC[31,32], and the good desolvation ability of DMC[33,34]. Specifically:

(1) for the SEI film forming ability, our theoretical calculation results show that the energy of the lowest unoccupied molecular orbital (LUMO) in DMDOHD is lower than that of conventional linear carbonates, (e.g., diethyl carbonate – DMC, ethyl methyl carbonate – EMC, or diethyl carbonate – DEC), and close to that of ethylene carbonate – EC (Supplementary Table 5 and Supplementary Note 2). Moreover, our Nernst equation calculations (Supplementary Fig. 2)

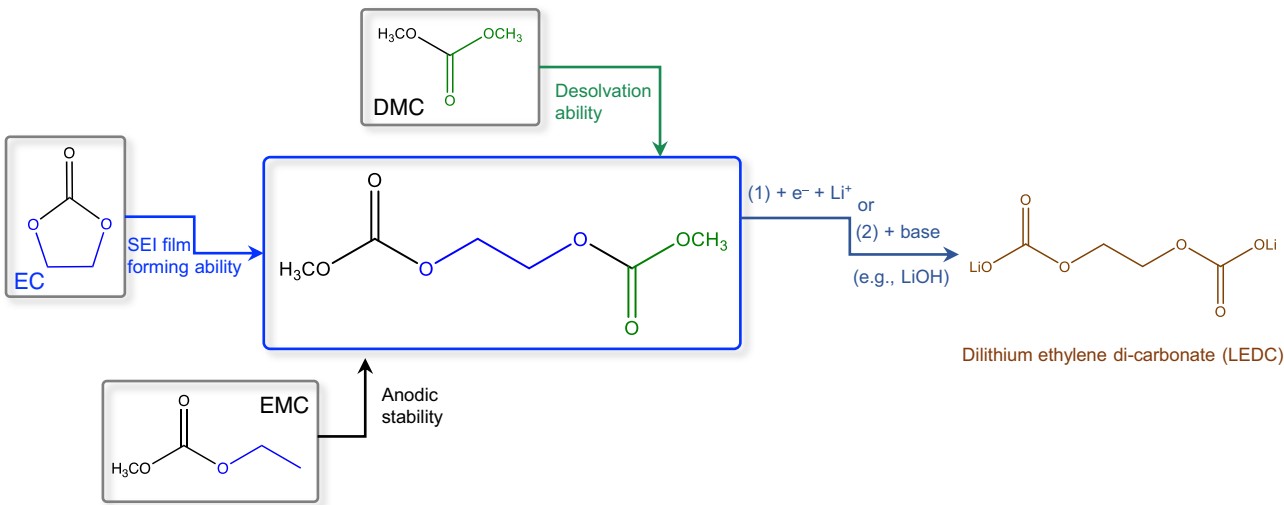

**Fig. 1 | Design rationale for selecting dimethyl 2,5-dioxahexanedioate (DMDOHD) as electrolyte solvent.** EC has good SEI film forming ability, EMC has high anodic stability, while DMC has good desolvation ability. As shown in Fig. 1, DMDOHD structurally combines the features of EC, EMC and DMC. Generally, the similarity in structural features will also lead to the similarity in physicochemical properties. Thus, it can be speculated that DMDOHD integrates the SEI film forming ability of EC, the high anodic stability of EMC, and the good desolvation ability of DMC.

illustrate that the reduction potential of Li+(EC) stands at 0.43 or 0.58 V vs. Li/Li+, as determined through B3LYP/6-311 + G(d,p) and M05-2X/6-31 + G(d,p) methods, respectively. These findings closely align with prior results obtained by Borodin and his coworkers[35–37], demonstrating the reliability of our computational methodology. As for the DMDOHD system, the calculations indicate that the lowest energy Li+(DMDOHD) conformer exhibits a reduction potential of 0.3 V vs. Li/Li+ (computed via B3LYP/6-311 + G(d,p)) or 0.23 V vs. Li/Li+ (calculated using M05-2X/6-31 + G(d,p)). Although these values are slightly lower than that of Li+(EC), they are still higher than the lithium electroplating potential (theoretical value: 0 V vs. Li/Li+), implying that DMDOHD has the ability to participate in SEI formation. The above analysis indicates that DMDOHD is preferentially reduced to form the SEI under cathodic polarization with an essential role to play in the modulation of the SEI composition and interfacial chemistry, like EC. In addition, the structure of DMDOHD is very similar to that of dilithium ethylene di-carbonate (LEDC, Fig. 1 and Supplementary Fig. 3), found recently as a key component of SEI generated from EC-based carbonate solvents[30,38]. DMDOHD can be also converted to LEDC through electrochemical reaction or saponification reaction with a base (such as LiOH) in LMBs. We presumed thus that DMDOHD will generate larger amounts of LEDC in the SEI, which is beneficial for stable cycling.

(2) For the anodic stability, DMDOHD also exhibits a lower than DMC, EMC and DEC highest occupied molecular orbital (HOMO) energy level (Supplementary Table 5 and Supplementary Note 2), implying improved anodic stability of the DMDOHD solvent. Additionally, in contrast to EC (ε = 89.6), DMDOHD (ε = 2.9) will exhibit minimal accumulation on the positive electrode surface during anodic polarization (charging) due to its significantly lower dielectric constant. Consequently, the CEI formation in the DMDOHD-based electrolyte will rely mainly on the anodic decomposition of the DMDOHD/anion complex (e.g., DMDOHD/PF6−) with lower contribution from the anodic decomposition of free solvent molecules compared to the conventional EC-containing commercial electrolytes (refer to Supplementary Table 6 and Supplementary Note 3 for details). This suggests that the CEI formed in the DMDOHD-based system might encompass a higher proportion of inorganic and fluorine-containing compounds compared to the conventional EC-based commercial electrolytes, fostering the development of a denser, more uniform, and robust CEI, which is able to effectively suppress the electrolyte decomposition and maintain the stability of the positive electrode/electrolyte interface,

thus facilitating the long-term cycling of the positive electrode material.

(3) As for the desolvation ability, it is generally considered to be opposite to the solvation ability of the electrolyte solvent, which is related to the interaction between Li ion and the carbonyl group of the carbonate solvent. Since DMDOHD has a similar carbonyl structure to DMC, its interaction with Li ions should be weaker than that of EC, inheriting the good desolvation ability of DMC. (4) The boiling and flash points of DMDOHD are both much higher than that of DMC, EMC and DEC (Supplementary Table 7), thereby the use of DMDOHD enhances the battery safety. Finally, as compared to EC, DMDOHD is liquid at room temperature and can be thus used without any co-solvent.

To verify these hypotheses, the physical properties and anodic stability of the DMDOHD-based electrolyte were first investigated. Figure 2a displays the temperature dependence of ionic conductivity for the 1 M LiPF6/DMC (1 M-LPF-DMC), 1 M LiPF6/EC/DMC (1 M-LPF-EC/DMC) and 1 M LiPF6/DMDOHD (1 M-LPF-DMDOHD) electrolytes in the temperature range from 20 to 70 °C. The data are all fitted by the Arrhenius equation and the corresponding results are shown in Supplementary Table 8. The 1 M-LPF-DMDOHD electrolyte shows a lower ionic conductivity of 0.21 mS cm−1 at 20 °C as compared to that of 1 M-LPF-DMC (2.6 mS cm−1) or 1 M-LPF-EC/DMC (7.8 mS cm−1) electrolytes, which is due to higher viscosity as compared to the short chain carbonates (Supplementary Tables 7 and 9). Certainly, such a low ionic conductivity may render this electrolyte unable to perform effectively at high C-rates under room temperature conditions, which will be discussed in detail in the latter text. However, this value is in the range of, or even higher than of many ionic liquids, solid ceramic and polymer electrolytes used in LMBs or LIBs (Supplementary Table 10), so that it can still meet the requirement of battery applications at low or moderate C-rates[39–42]. The contact angle measurements also indicate good wettability of Celgard® separators with 1 M-LPF-DMDOHD electrolyte (Supplementary Fig. 4), which should allow efficient impregnation and formation cycle the battery assembly step.

To analyze and confirm the anodic stability of the DMDOHD-based electrolyte (according to HOMO energy calculations), linear sweep voltammetry (LSV) measurement was conducted. As shown in Supplementary Figs. 5 and 6, the 1 M-LPF-DMDOHD electrolyte shows higher anodic stability compared to the reference carbonate-based electrolytes, as evidenced by the lower anodic current and higher

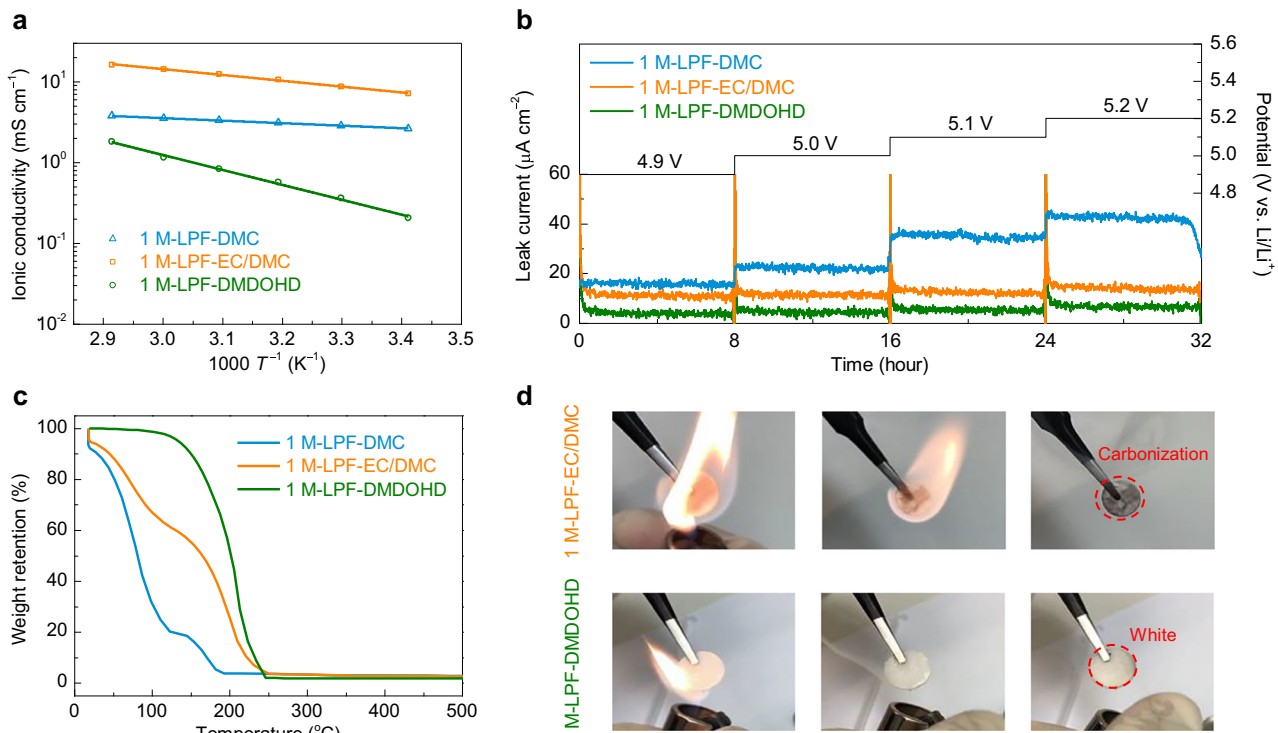

**Fig. 2 | Physical properties and anodic stability. a** Variable temperature ionic conductivity of 1 M-LPF-DMC, 1 M-LPF-EC/DMC and 1 M-LPF-DMDOHD electrolytes. **b** Potentiostatic floating profiles of Li||LiNi$_{0.5}$Mn$_{1.5}$O$_4$ cells maintained at 4.9, 5.0, 5.1 and 5.2 V for a period of 8 h. The test was performed at room temperature. **c** Thermogravimetric analysis of 1 M-LPF-DMC, 1 M-LPF-EC/DMC and 1 M-LPF-DMDOHD electrolytes. **d** Flame tests of glass fiber sheets soaked with 1 M-LPF-EC/DMC (upper panels) and 1 M-LPF-DMDOHD (bottom panels) electrolytes.

onset oxidation potential (See detailed discussion in Supplementary Note 4). The high voltage tolerance of DMDOHD-based electrolyte suggests its potential combination with 5 V-class positive electrode material systems, e.g., LiNi$_{0.5}$Mn$_{1.5}$O$_4$. To confirm this, an electrochemical floating test was performed in a Li||LiNi$_{0.5}$Mn$_{1.5}$O$_4$ cell configuration with potential steps of 0.1 V, ranging from 4.9 to 5.2 V (vs. Li/Li$^+$). As presented in Fig. 2b, the 1 M-LPF-DMDOHD electrolyte remains stable with a minimal leak current at 4.9 V, with further minimal increase as the voltage rises to 5.2 V, whereas the 1 M-LPF-EC/DMC or the 1 M-LPF-DMC formulations show a much larger leak current which gradually increases as the polarization rises.

Considering the safety aspects of most carbonate solvents, the thermal stability of the DMDOHD-based electrolyte was also studied. Figure 2c depicts the thermogravimetric analysis of the three electrolyte formulations. The 1 M-LPF-DMDOHD electrolyte exhibits no significant weight loss up to 100 °C, whereas considerable weight loss is observed for both,1 M-LPF-DMC and 1 M-LPF-EC/DMC electrolytes tested under the same conditions, indicating that DMDOHD-based electrolyte has a better thermal stability. This is in accordance with the boiling point of the constituent solvents (DMC – 91 °C, EC – 248 °C, DMDOHD – 220 °C), while also considering the thermal stability of LiPF$_6$. Furthermore, the differential scanning calorimetry (DSC) results also demonstrate that the thermal stability of DMDOHD-based electrolyte is superior to that of DMC and EC/DMC-based electrolytes (see detailed discussion in Supplementary Fig. 7).

When directly exposed to an open flame, a 1 M-LPF-EC/DMC impregnated glass fiber disk ignited instantly, sustaining the fire and burning rapidly even in the absence of external thermal stimulus (Fig. 2d−upper panel, as well as Supplementary Movie 1). By contrast, the 1 M-LPF-DMDOHD electrolyte did not ignite and self-extinguished after exposure to the direct flame (Fig. 2d – bottom panel, and Supplementary Movie 2), implying improved safety, attributed to its high

flash point (Supplementary Table 6), and high thermal stability (Fig. 2c). However, it is worth mentioning that the flame test experiments in Supplementary Movies 1 and 2 are illustrative of the improved thermal stability of the DMDOHD-based electrolyte as compared to that of the state-of-the-art electrolyte, while still remaining flammable at higher temperatures. The 1 M-LPF-DMDOHD electrolyte will also ignite when the temperature is high enough, such as reaching the flash point of DMDOHD or using a longer exposure to an external flame source (Supplementary Movies 3 and 4), which remains a common problem of most organic solvents-based electrolytes. This can be further improved by introducing the fire-retardant additives (beyond the research focus of this work and was not studied furthermore in details).

Additionally, to more clearly compare the safety differences between the two electrolytes, we conducted tests on the self-extinguishing time (SET) of the two electrolytes under three different conditions. The results, presented in Supplementary Movies 5–10, Supplementary Figs. 8–10 and Supplementary Table 11, consistently showing that the 1 M-LPF-DMDOHD electrolyte has smaller SET values than the 1 M-LPF-EC/DMC electrolyte, regardless of the test conditions, suggesting thus its superior safety properties. Overall, the above analysis confirm that the DMDOHD-based electrolyte possesses considerably better anodic stability and safety than the state-of-the-art commercial carbonate-based electrolytes, albeit lower transport properties.

### Electrochemical stability of the DMDOHD-based electrolyte towards lithium metal
The lithium plating/stripping behavior in the DMDOHD-based electrolyte was next evaluated in Li||Cu cell configuration. As shown in Fig. 3a (as well as Supplementary Figs. 11 and 12), the overpotential measured in the cells with 1 M-LPF-DMDOHD electrolyte is slightly

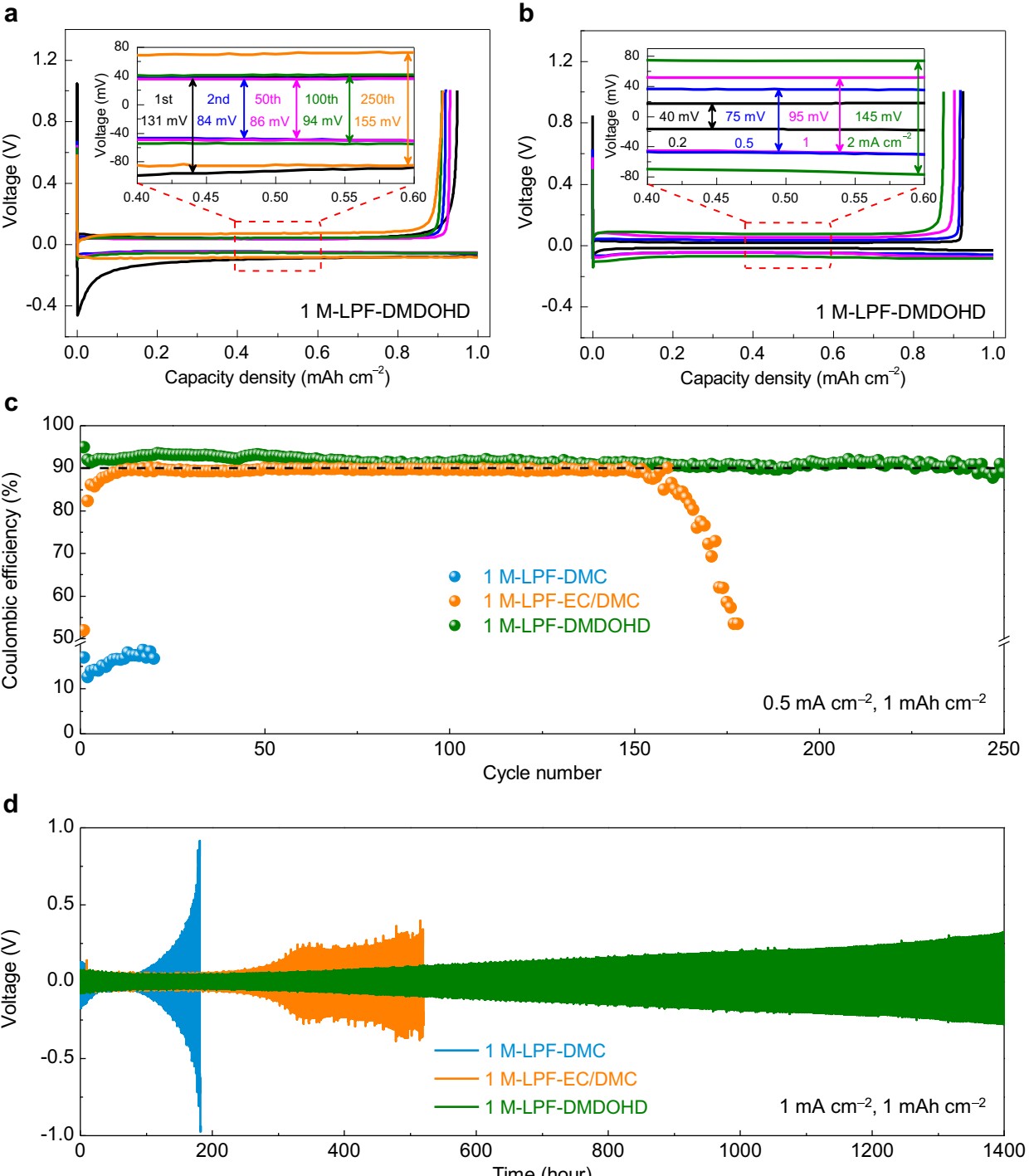

**Fig. 3 | Lithium plating/stripping behavior in different electrolytes.**
**a** Galvanostatic lithium plating/stripping profile of Li||Cu cell configuration cycled in 1 M-LPF-DMDOHD electrolyte at a current density of 0.5 mA cm⁻² (cutoff capacity of 1 mAh cm⁻²). **b** Polarization profiles of the Li||Cu cell configuration cycled in 1 M-LPF-DMDOHD electrolyte at different current densities. **c** Lithium plating/stripping coulombic efficiency of the Li||Cu cell cycled in different electrolytes at a current density of 0.5 mA cm⁻² with a cutoff capacity of 1 mAh cm⁻². **d** Cycling performance and polarization evolution of the Li||Li symmetric cells in different electrolytes at a current density of 1 mA cm⁻² and a cutoff capacity of 1 mAh cm⁻².

higher than for 1 M-LPF-DMC or 1 M-LPF-EC/DMC electrolytes, which is due to the higher viscosity (Supplementary Table 9) and consequently lower conductivity (Supplementary Table 8) of the 1 M-LPF-DMDOHD electrolyte. However, the lithium plating/stripping CE and stability of the 1 M-LPF-DMDOHD electrolyte are both increased as compared to that of 1 M-LPF-DMC and 1 M-LPF-EC/DMC electrolytes (Fig. 3c and Supplementary Fig. 12). Specifically, when tested at a current density of

0.5 mA cm⁻² and a capacity of 1 mAh cm⁻², the Li||Cu cell with 1 M-LPF-DMC was found to exhibit poor lithium metal plating/stripping CE (<20% for the first 20 cycles). Although the addition of EC as a co-solvent (i.e., 1 M-LPF-EC/DMC) improves significantly the plating-stripping efficiency, the results are still unsatisfactory—the average CE for the first 150 cycles remains below 90% (ca. -89%) and decreases sharply after 160 cycles. On the contrary, a longer cycle life of >250

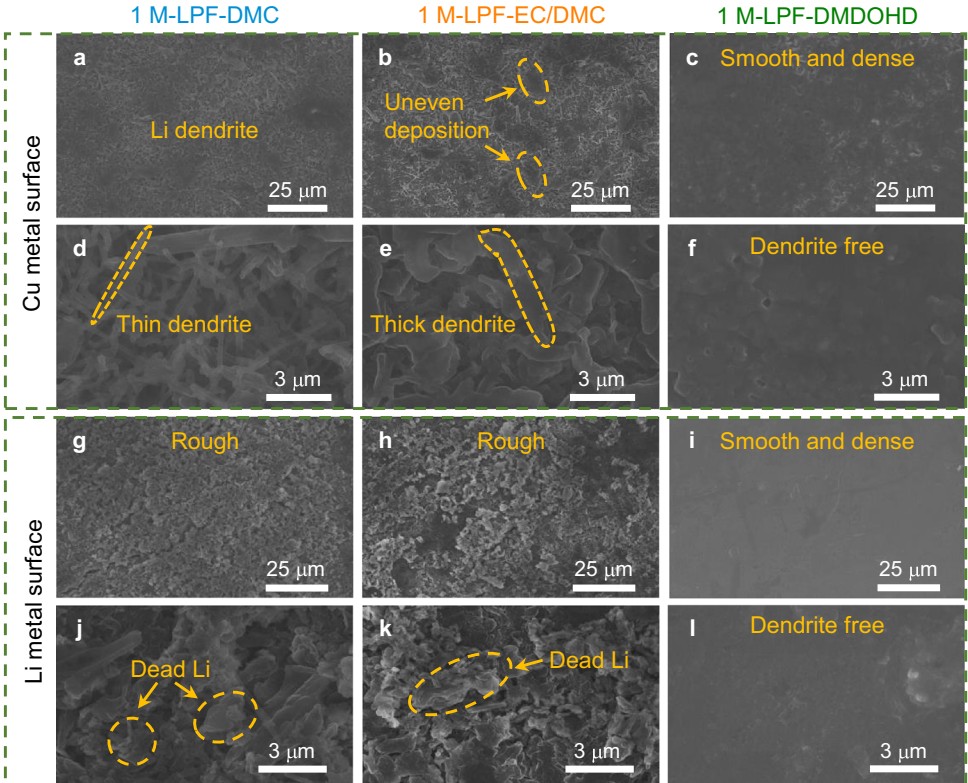

**Fig. 4 | Scanning electron microscopy of Cu electrode and lithium metal surfaces cycled in different electrolytes. a–f** Images of Cu metal surface after plating 1 mAh cm$^{-2}$ lithium metal in (**a**, **d**) 1 M-LPF-DMC, (**b**, **e**) 1 M-LPF-EC/DMC and (**c**, **f**) 1 M-LPF-DMDOHD electrolytes at a current density of 0.2 mA cm$^{-2}$. **g–l** Images of lithium metal surface after 20 cycles (cell//test stopped and disassembled after Li stripping sequence) in (**g**, **j**) 1 M-LPF-DMC, (**h**, **k**) 1 M-LPF-EC/DMC and (**i**, **l**) 1 M-LPF-DMDOHD electrolytes at a current density of 0.5 mA cm$^{-2}$.

cycles, with a higher average lithium plating/stripping CE of 92% can be attained with 1 M-LPF-DMDOHD electrolyte. To the best of our knowledge, this is the highest lithium plating/stripping CE reported so far for the low concentration (1 M) carbonate-based electrolytes without any additives or co-solvents (Supplementary Table 1).

As the current density increases from 0.2 to 2 mA cm$^{-2}$, the cell displayed minor polarization increase of ~20, ~37.5, ~47.5 and ~72.5 mV (Fig. 3b), demonstrating the good rate capability, despite having lower ionic conductivity than the control carbonate-based systems. At high current density of 1 mA cm$^{-2}$, the cell with 1 M-LPF-DMDOHD electrolyte also displayed good stability, sustaining more 150 cycles, with an average CE of ~87% (Supplementary Fig. 12), which remains much better than that of the cell with 1 M-LPF-EC/DMC electrolyte formulation. The long-term stability of the lithium metal in the three electrolyte compositions was also analyzed in a Li||Li symmetric cell configuration. As shown in Fig. 3d, for an applied current density of 1 mA cm$^{-2}$ and a total exchange capacity of 1 mAh cm$^{-2}$, the cell with 1 M-LPF-DMDOHD electrolyte shows the most stable and reversible lithium plating/stripping behavior over 1400 h of testing, whereas performances of the cells with 1 M-LPF-DMC or 1 M-LPF-EC/DMC were found to rapidly degrade. Considering that the viscosity and the ionic conductivity of the bulk electrolyte is not changing significantly with cycling, the observed improvements in lithium plating/stripping CE and cycling stability are assigned to a stable lithium metal interfacial chemistry endowed by the DMDOHD solvent, which will be discussed in the following sections.

**Morphological and structural characterization of lithium metal electrode**

To further confirm the improved lithium interfacial compatibility with the DMDOHD-based electrolyte, the surface morphology of Cu

electrodes (of the cycled Li||Cu cells) and of lithium metal disks (of the cycled Li||Li symmetric cells) for the three studied electrolyte formulations were analyzed by scanning electron microscope (SEM). As shown by Fig. 4a, d, after plating 1 mAh cm$^{-2}$ of Li at a current density of 0.2 mA cm$^{-2}$, the surface of the Cu electrode operated in 1 M-LPF-DMC displays massive, needle-like lithium dendritic growth and a highly porous structure. This morphology is caused by the continuous reaction between the lithium metal and electrolyte[43], due to the poor ability of DMC to form a stable SEI on the negative electrode surface[32], confirmed by the measured low lithium plating/stripping CE (<20%) and poor cycling stability (Fig. 3c). After adding EC co-solvent, the lithium deposition morphology on the Cu surface is improved and lower porosity deposits are observed (Fig. 4b, e). The lithium deposits remain however unevenly distributed, the porous structure and lithium dendrites are overwhelming, so that the lithium plating/stripping CE and cycling stability of the Li||Cu cell with 1 M-LPF-EC/DMC electrolyte remain unsatisfactory (Fig. 3c).

In stark contrast, while using the 1 M-LPF-DMDOHD electrolyte, a smooth, compact and dendrite-free lithium deposit can be achieved on the Cu surface, confirming the improved interfacial chemistry, endowed by the protective SEI formed in DMDOHD-based electrolyte (Fig. 4c, f). Corroborating information is retrieved from the analysis of the lithium metal surface taken from the symmetric lithium cells after 20 cycles at a current density of 0.5 mA cm$^{-2}$ with a cutoff capacity of 0.5 mAh cm$^{-2}$ (cell stopped and disassembled after Li stripping sequence in Fig. 3d). As shown in Fig. 4g–k, the lithium metal surface cycled in 1 M-LPF-DMC (Figs. 4g, j) and 1 M-LPF-EC/DMC (Fig. 4h, k) electrolytes presents a porous, rough surface with huge blocks accumulated in the form of dead lithium. The continuous accumulation of the SEI product and of dead lithium on the surface explains the increasing polarization of the Li||Li cell during cycling in these electrolytes, which will eventually lead to the

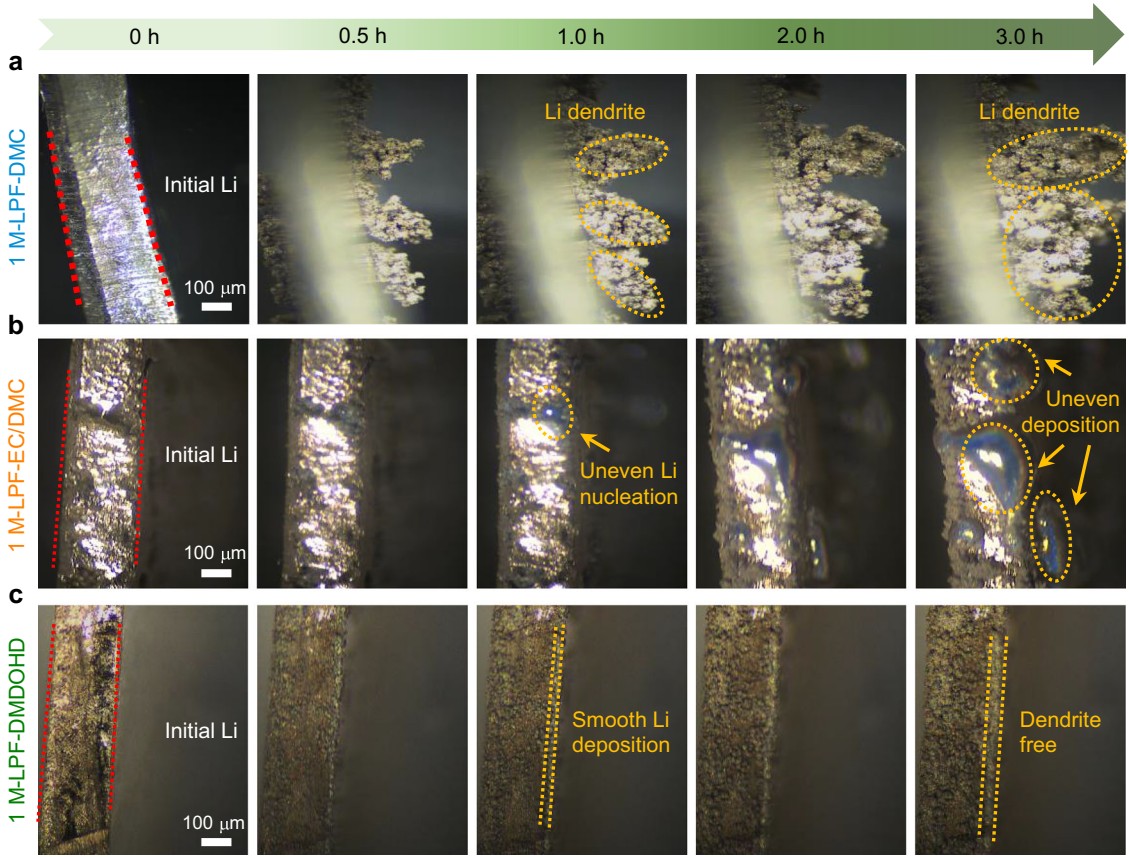

**Fig. 5 | In-situ optical microscopy of dynamic lithium deposition in different electrolytes. a–c** Images of lithium deposition at a current density of 1 mA cm$^{-2}$ for 0, 0.5, 1, 2 and 3 h in (**a**) 1 M-LPF-DMC, (**b**) 1 M-LPF-EC/DMC and (**c**) 1 M-LPF-DMDOHD electrolytes.

cell failure once the electrolyte is completely depleted. In sharp contrast, the lithium metal can maintain a dense, smooth and dendritic-free surface (Fig. 4i, l) while cycled in the 1 M-LPF-DMDOHD electrolyte, demonstrating the improved lithium metal compatibility of the DMDOHD-based electrolyte as well.

The dynamics of the lithium plating process was monitored by in-situ optical microscopy as visual evidence to present the different morphological evolution of the deposited lithium metal in the three electrolyte systems. As shown in Fig. 5a and Supplementary Movie 11, lithium dendrites are generated on the lithium metal surface after deposition for a duration of 0.5 h at a current density of 1 mA cm$^{-2}$ in the 1 M-LPF-DMC electrolyte. The dendrites gradually grow and become massive after 3 h of deposition, corroborating the SEM ex-situ analysis results (Fig. 4a, d). For the 1 M-LPF-EC/DMC electrolyte system, no significant dendrite formation was observed on the lithium metal surface during the initial 0.5 h. However, protrusions are generated on the lithium metal surface after deposition for 1 h, which would serve as preferential nucleation sites for the further lithium deposition and eventually result in uneven lithium deposition with porous structure and the growth of thick lithium dendrites (Fig. 5b and Supplementary Movie 12). The lithium metal surface in the 1 M-LPF-DMDOHD electrolyte exhibits a smooth and highly compact lithium deposit, without dendrite growth and obvious non-uniform volume expansion during the whole plating process (Fig. 5c and Supplementary Movie 13), once again confirming the good interfacial chemistry of the DMDOHD-based electrolyte with the Li-metal.

## Electrochemical performance of the high-voltage LMBs with DMDOHD-based electrolyte

Encouraged by the excellent results attained on the lithium metal interfacial compatibility and the enhanced anodic stability of the DMDOHD-based electrolyte, we applied the developed formulation to a 5 V-class high-voltage LMB using the LiNi$_{0.5}$Mn$_{1.5}$O$_4$ as positive electrode material. Figure 6a shows the first galvanostatic charge-discharge curves of the Li||LiNi$_{0.5}$Mn$_{1.5}$O$_4$ cells (LiNi$_{0.5}$Mn$_{1.5}$O$_4$ active material mass loading of 10 mg cm$^{-2}$) cycled in different electrolytes at a rate of C/10. Compared to the cells with 1 M-LPF-DMC (~89.5%) and 1 M-LPF-EC/DMC (~90.3%) electrolytes, the Li||LiNi$_{0.5}$Mn$_{1.5}$O$_4$ cell with 1 M-LPF-DMDOHD electrolyte exhibits the highest first cycle CE of 94.4%. In addition, the DMDOHD-based electrolyte also displays improved cycling stability, with a capacity retention of ~94% over more than 200 cycles, and an average CE of 99.2% at a rate of C/5, which is considerably better when compared to stability attained with 1 M-LPF-DMC (~48% capacity retention after 100 cycles), or with 1 M-LPF-EC/DMC (~71% capacity retention after 200 cycles) electrolyte systems (Fig. 6b, c and Supplementary Fig. 13). At higher rate of C/3, the Li||LiNi$_{0.5}$Mn$_{1.5}$O$_4$ cell with 1 M-LPF-DMDOHD electrolyte can still maintain a high-capacity retention ratio of ~94% after 250 cycles (Fig. 6d and Supplementary Fig. 14). The rate capability of the Li||LiNi$_{0.5}$Mn$_{1.5}$O$_4$ cell with 1 M-LPF-DMDOHD electrolyte was also studied. As presented in Supplementary Fig. 15, the Li||LiNi$_{0.5}$Mn$_{1.5}$O$_4$ cell operated with the 1 M-LPF-DMDOHD electrolyte retained a low specific capacity at C-rates above 1 C, which can be attributed to the low ionic conductivity of the 1 M-LPF-DMDOHD electrolyte. However, it was able to return to a discharge capacity close to that of the initial capacity when the C-rate reverted back to a small C-rate of 0.2 C, demonstrating its good capacity recovery capability.

It is worth mentioning that many studies typically use oversized thick lithium foils, often around 1 mm in thickness, to assemble LMBs. This results in a substantial negative/positive (N/P) electrode capacity ratio, often exceeding 100, which deviates significantly from the practical application conditions where the N/P ratio should be as close

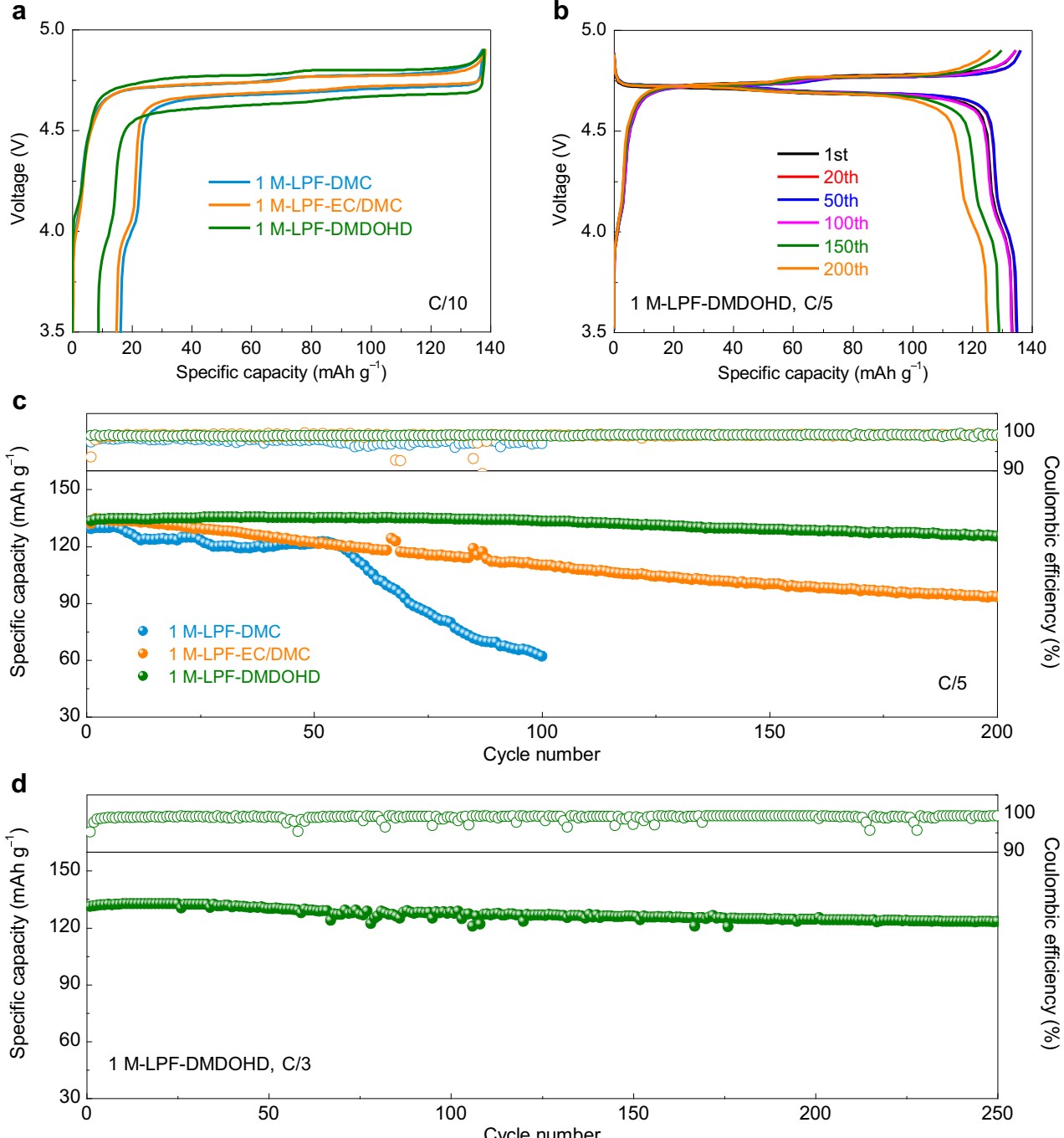

**Fig. 6 | Electrochemical performances of LMBs using LiNi$_{0.5}$Mn$_{1.5}$O$_4$ as positive electrode material cycled in different electrolytes. a** The first galvanostatic charge-discharge curves of the Li||LiNi$_{0.5}$Mn$_{1.5}$O$_4$ cells cycled in different electrolytes at a rate of C/10 (1 C corresponds to 142 mA g$^{-1}$). **b** Galvanostatic charge-discharge plots at different cycle indices for the Li||LiNi$_{0.5}$Mn$_{1.5}$O$_4$ cell cycled with 1 M-LPF-DMDOHD electrolyte at a rate of C/5. **c** Cycling stability of the Li||LiNi$_{0.5}$Mn$_{1.5}$O$_4$ cells in different electrolytes (C/5 cycling rate). **d** Cycling stability of the Li||LiNi$_{0.5}$Mn$_{1.5}$O$_4$ cell with 1 M-LPF-DMDOHD electrolyte cycled at rate of C/3.

as possible to 1. Due to our current limitations in preparing lithium metal pouch cells, to create more rigorous and practical testing conditions, we have focused on maximizing the LiNi$_{0.5}$Mn$_{1.5}$O$_4$ positive electrode material mass loading in the coin cell configuration (e.g., 16 mg cm$^{-2}$, with a theoretical areal capacity of about 2.3 mAh cm$^{-2}$) and reducing the thickness of the lithium metal negative electrode (e.g., 50 μm, with a theoretical areal capacity of about 10.3 mAh cm$^{-2}$) to reduce the N/P ratio. As depicted in Supplementary Fig. 16, when operating at an N/P ratio exceeding 50, the DMDOHD-based electrolyte displays excellent cycling stability with a capacity retention of

~93% over more than 100 cycles, and an average CE of 99.3% at a rate of C/5, which is considerably better when compared to stability attained with 1 M-LPF-EC/DMC (~79% capacity retention after 70 cycles) electrolyte system. Even at a substantially lower N/P ratio of approximately 4.5, the Li||LiNi$_{0.5}$Mn$_{1.5}$O$_4$ cell operated with the 1 M-LPF-DMDOHD electrolyte manages to uphold a high-capacity retention ratio of approximately 85% after 80 cycles (Supplementary Fig. 17). In contrast, the cell employing the 1 M-LPF-EC/DMC electrolyte system retained only about 38% of its initial capacity for the same number of cycles.

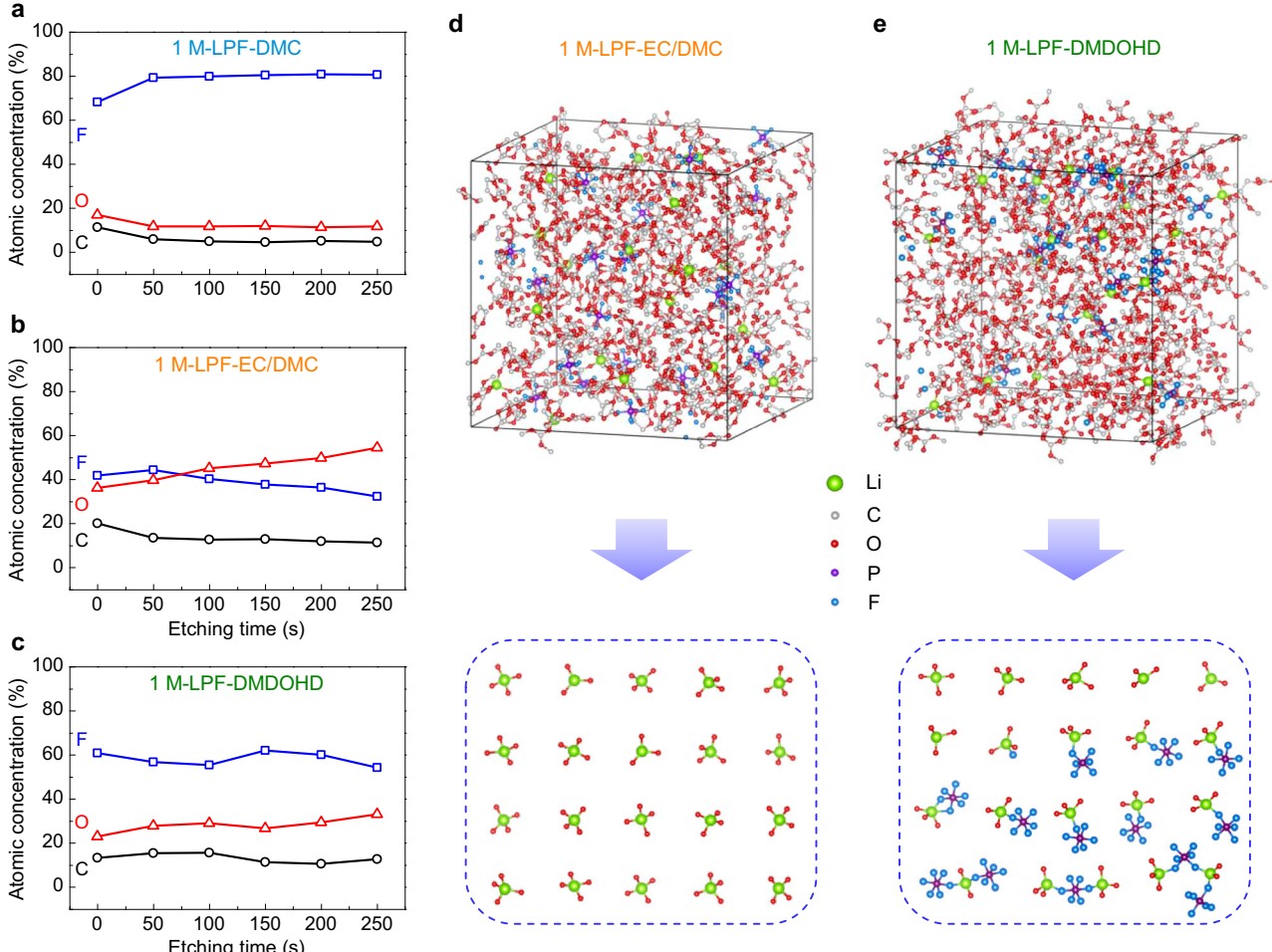

**Fig. 7 | Surface composition analysis of the SEI formed on Cu electrodes and the related solvation structure of the electrolytes. a–c** In-depth composition profiles for various durations of Ar⁺ sputtering of the SEI formed on Cu/Li electrodes cycled in (**a**) 1 M-LPF-DMC, (**b**) 1 M-LPF-EC/DMC and (**c**) 1 M-LPF-DMDOHD electrolytes. Snapshots of the molecular dynamics simulation box and the simplified 20 Li⁺- coordination environment structures for 1 M-LPF-EC/DMC (**d**) and 1 M-LPF-DMDOHD (**e**) electrolytes. Atom colors: Li, green; C, dark gray; O, red; P, purple; F, light blue. The cubic boxes represent the periodic boundaries of the supercells used in the molecular dynamic simulations.

To further highlight the high anodic stability of the DMDOHD-based electrolyte, we also tested the electrochemical performance of the Li||LiNi$_{0.5}$Mn$_{1.5}$O$_4$ cells cycled in different electrolytes with the charge cutoff potential increased to 5 V vs. Li/Li⁺. As shown in Supplementary Fig. 18, the Li||LiNi$_{0.5}$Mn$_{1.5}$O$_4$ cell with 1 M-LPF-DMDOHD electrolyte provides stable cycling over more than 120 cycles, with a capacity retention of ~95%. On the contrary, an irreversible overcharge is observed when 1 M-LPF-EC/DMC electrolyte is used on the first charge (Supplementary Fig. 19), consistent the lower anodic stability of this electrolyte at 5 V. These results consolidate the findings on the high anodic stability of the DMDOHD-based electrolyte and its application potential in actual high-voltage LMB systems.

**Lithium metal negative electrode and 5 V · LiNi$_{0.5}$Mn$_{1.5}$O$_4$ interphase chemistry**

It is widely accepted that the interfacial compatibility and cycling stability of lithium metal are closely related to the properties of the formed SEI[10,16]. To correlate the electrochemical performances and the SEI nature attained in the three electrolyte formulations studied, in-depth X-ray photoelectron spectroscopy (XPS) profiling technique was used on cycled electrodes. Figure 7a–c, Supplementary Figs. 20–22 and Supplementary Table 12 present the elemental content and composition of the SEI formed in the three electrolytes at different depths.

In the 1 M-LPF-DMC electrolyte system, excessive fluorine content is detected, associated with the inorganic species such as LiF, Li$_x$PO$_y$F$_z$ and Li$_x$PF$_y$ originating primarily from the decomposition of LiPF$_6$ (Fig. 7a, Supplementary Fig. 20a and Supplementary Table 12). In contrast, the content of carbon and oxygen that are mainly associated with the organic species such as C − C, C − O, C = O and −OCO$_2$Li is very small (Supplementary Figs. 21a, 22a and Supplementary Table 12), indicating the lower contribution of solvent molecules in the formation of SEI, or the high solubility of the formed organic species. It has been reported that the presence of organic species in the SEI is related to the mechanical flexibility and robustness, while the inorganic species is associated with the rigidity of the SEI[10,44]. Inorganic-excessive, combined with low organic content SEI formed in 1 M-LPF-DMC electrolyte renders the surface films highly rigid, which is prone to fracturing during volume changes associated with Li-metal deposition and stripping. Therefore, such type of SEI cannot effectively prevent the continuous reaction between the lithium metal and electrolyte, which will inevitably result in continuous electrolyte consumption, lithium dendrite growth and low lithium plating/stripping CE (Supplementary Fig. 23), in line with the observed experimental results (Figs. 3c and 4a, d).

Addition of EC (i.e., 1 M-LPF-EC/DMC) reverses the trend: the fluorine content decreases significantly, while the carbon and oxygen

amount increases (Fig. 7b), which suggests that the SEI formed in the 1 M-LPF-EC/DMC composition contains a higher amount of organic species as compared to inorganic ones (Supplementary Figs. 20b, 21b, 22b and Supplementary Table 12). The higher organic content originates from the decomposition of EC due to its strong lithium-ion solvating ability that enables it to enter the solvation shell and thus participate in the formation of SEI as demonstrated in the previous studies[12,32,45]. Obviously, the increase of organic species in the SEI formed in the 1 M-LPF-EC/DMC electrolyte enhances its flexibility and making it robust towards volume changes during cycling, thereby suppressing to some extent the continuous reaction between lithium metal and the electrolyte, as evidenced by the decreased dendritic growth and lower porosity of the deposited lithium (Fig. 4b, e) and the improved lithium plating/stripping CE (Fig. 3c). However, as demonstrated in other studies[46,47], the diffusion of Li ions through the SEI is hindered by the presence of organic species due to their high bonding affinity, resulting in vertical lithium deposition and perforation the SEI. In addition, the organic SEI also possesses a porous structure that is not dense enough to completely block the diffusion of (fully, or partially solvated Li-ions) and thus further reaction between lithium metal and electrolyte solvent(s)[46]. Therefore, the properties of the SEI formed in 1 M-LPF-EC/DMC system are still not satisfactory enough to avoid the lithium dendritic growth and attain high lithium plating/stripping CE (Supplementary Fig. 23).

Different from and as compared to the two previous baseline systems, the composition of the SEI formed in the 1 M-LPF-DMDOHD electrolyte system shows an intermediate situation − with the content of both inorganic and organic species being balanced (Fig. 7c, Supplementary Figs. 20–22 and Supplementary Table 12), similar to the situation of inorganic-rich SEI as discussed in the literature[10,46,48,49]. The increased content of inorganic components in the SEI can facilitate the lateral diffusion of lithium ions along the SEI/lithium metal interface due to the weaker affinity with lithium metal/lithium ion[46,50,51]. The inorganic species also generally possess a higher Young's modulus, which endows the SEI with adequate mechanical properties to suppress the growth of lithium dendrites and the vertical penetration of lithium into SEI[46,51]. The SEI formed in the 1 M-LPF-DMDOHD electrolyte also shows a stable depth composition profile in fluorine, carbon and oxygen (Fig. 7c), implying the uniformity of the formed SEI. Such a uniform SEI with suitable inorganic-organic mixed species is expected to possess both good rigidity and flexibility (Supplementary Fig. 23)[6,10,44], thus stabilizing and protecting the lithium metal from the growth of lithium dendrites and the corrosion of electrolyte (Fig. 4c, f, i, l).

In fact, the SEI is formed by the reduction of the solvents and anions in the solvation shell on the negative electrode surface, which is closely related to the solvation structure of the electrolyte system[52,53]. To further understand the formation of the different SEI compositions in these different electrolyte systems, molecular dynamics (MD) simulations were conducted to explore the solvation structure in the studied electrolyte systems (Fig. 7d, e, Supplementary Figs. 24 and 25). The snapshots shown in Fig. 7d, e simulate the coordination environment of Li$^+$ in 1 M-LPF-EC/DMC and 1 M-LPF-DMDOHD electrolytes, respectively. In the 1 M-LPF-EC/DMC electrolyte, all the Li ions are coordinated by four solvent molecules in the first solvation shell (Fig. 7d), with an average coordination number (CN) of Li$^+$ with EC/DMC at ≈4 (EC: ≈2.2, DMC: ≈1.8), and an average CN of Li$^+$ with PF$_6^-$ at ≈0 (Supplementary Figs. 24 and 25), resulting in a solvent-separated ion pair (SSIP) solvation structure. The strong coordination of Li$^+$−solvent demonstrate the predominant role of carbonate solvent (especially EC, as it has a larger CN of Li$^+$ compared to DMC) in the formation of SEI in 1 M-LPF-EC/DMC electrolyte, corresponding to the results of XPS (Fig. 7b). In contrast, within the first solvation shell of the 1 M-LPF-DMDOHD electrolyte system (Fig. 7e), nearly 30% of Li$^+$ are coordinated by four solvent molecules to form SSIPs, 50% of Li$^+$ forms contact ion pairs (CIPs) with the solvent molecules and PF$_6^-$ anions,

while 20% of Li$^+$ participates in the formation of ion aggregates (AGGs), which results in an average CN value of Li$^+$ with DMDOHD and PF$_6^-$ of ≈2.9 and ≈1.1, respectively (Supplementary Fig. 24). The strong interaction between PF$_6^-$ and Li$^+$ in 1 M-LPF-DMDOHD electrolyte weakens the Li$^+$−solvent interaction, as also evidenced by the decreased CN (about 28%) of oxygen (from solvent) in the first solvation shell of Li$^+$ compared to that of 1 M-LPF-EC/DMC electrolyte (Supplementary Fig. 24a), which leads to the formation of an anion-derived SEI, corresponding to the XPS results again (Fig. 7c). In addition, with the reduced coordination number of solvent in the first solvation shell of Li$^+$, the overall solvation energy of Li$^+$ will be decreased in the 1 M-LPF-DMDOHD electrolyte and the corresponding Li$^+$-desolvation process at the electrode/electrolyte interface will be kinetically favorable[54]. Therefore, taken the above results and analysis into account, it is reasonable to understand why the 1 M-LPF-DMDOHD electrolyte shows better electrochemical performances, although the ionic conductivity is inferior (given also the ion pairing) compared to that of 1 M-LPF-EC/DMC electrolyte.

To further verify the differences in ion pairing between the 1 M-LPF-EC/DMC and 1 M LPF-DMDOHD electrolyte systems, Raman spectroscopy and $^7$Li nuclear magnetic resonance (NMR) spectrum analysis were conducted. As shown in Supplementary Fig. 26, within the 1 M-LPF-EC/DMC electrolyte system, the Raman band corresponding to the PF$_6^-$ anion appears at 741 cm$^{-1}$, which suggests that the Li$^+$ solvation structure predominantly comprises SSIPs, aligning with prior research findings[55,56]. In contrast, the 1 M-LPF-DMDOHD electrolyte system exhibits a notable blue shift in the spectral position of the PF$_6^-$ anion Raman band, specifically at around 745 cm$^{-1}$, implying that the lithium-ion solvation structure in this system may involve a combination of SSIPs, CIPs, and AGGs[55,57]. The formation of CIPs and AGGs within the 1 M-LPF-DMDOHD electrolyte system will intensify the interaction between Li$^+$ and PF$_6^-$, resulting in an increase in electron cloud density around Li$^+$ (known as the shielding effect), which is further supported by the upfield chemical shift observed in the $^7$Li NMR in Supplementary Fig. 27 when comparing the 1 M-LPF-DMDOHD electrolyte system to the 1 M-LPF-EC/DMC electrolyte system. These results corroborate the solvation structure suggested by the MD simulations.

Similar to SEI, the properties of CEI will also determine the stability of the positive electrode/electrolyte interface and the electrochemical performances, especially at high potentials[6,45]. A stable, conformal and uniform CEI will suppress the side reactions between positive electrode material and electrolyte by blocking electron transfer at the positive electrode/electrolyte interface, thereby maintaining the integrity and composition of the electrolyte and interphase. Whereas a thick and porous CEI will not only fail to suppress the irreversible oxidation of the electrolyte at high voltages, but will also hinder the transportat of lithium ion. Several studies revealed that the composition of CEI also plays an essential role on its physicochemical properties[6,12]. For instance, the −CF$_3$ and Li$_x$PO$_y$F$_z$ species are considered to have higher oxidation stability than organic/inorganic oxide species, while being also conducive to the formation of a conformal and dense CEI on the positive electrode surface[6,12], thus effectively suppressing the side reactions between positive electrode material and electrolyte. Based on the above rationale, and to explain the different electrochemical performances of LiNi$_{0.5}$Mn$_{1.5}$O$_4$ in the three electrolyte systems, XPS and transmission electron microscopy techniques were used to analyze the CEI composition and structure.

Figure 8a, Supplementary Figs. 28–30 and Supplementary Table 13 display the elemental content and composition of the CEI formed in the three electrolytes. Qualitatively, the composition of the CEI formed in the three electrolytes is similar, including the −CF$_3$, Li$_x$PO$_y$F$_z$, LiF, C − C, and organic/inorganic oxide species (e.g., Li$_2$CO$_3$, −OCO$_2$Li, C = O, C − O, and M − O). The content of these however is significantly different. For example, the CEI formed in the 1 M-LPF-

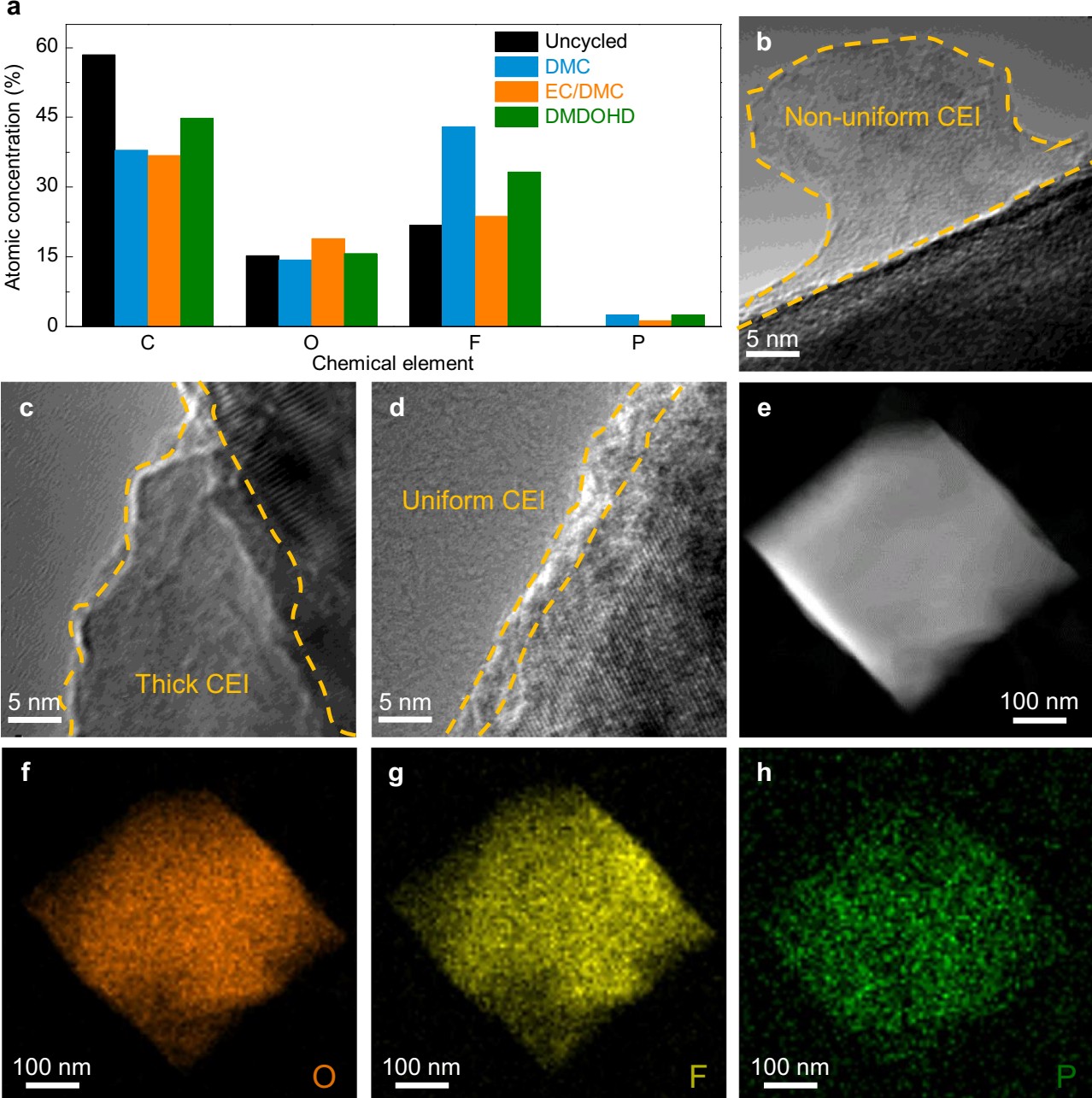

**Fig. 8 | Surface composition analysis and transmission electron microscopy of the LiNi$_{0.5}$Mn$_{1.5}$O$_4$ positive electrode material cycled in different electrolytes. a** Composition of the CEI on the surface of cycled LiNi$_{0.5}$Mn$_{1.5}$O$_4$ positive electrode materials in different electrolytes. High-resolution transmission electron microscopy (HRTEM) images of the LiNi$_{0.5}$Mn$_{1.5}$O$_4$ particles surface cycled in (**b**) 1 M-LPF-DMC, (**c**) 1 M-LPF-EC/DMC and (**d**) 1 M-LPF-DMDOHD electrolytes. High angle annular dark field-scanning transmission electron microscopy (HAADF-STEM) image (**e**) and the corresponding EDX maps for different elements of O (**f**), F (**g**), and P (**h**) of a LiNi$_{0.5}$Mn$_{1.5}$O$_4$ particle cycled in 1 M-LPF-DMDOHD electrolyte.

DMDOHD electrolyte possesses a considerably higher amount of −CF$_3$ and Li$_x$PO$_y$F$_z$ components compared to the one formed in the two baseline electrolytes (Supplementary Fig. 30 and Supplementary Table 13), which, as discussed earlier, display higher anodic stability. The high-resolution TEM images (Fig. 8b–d) and energy-dispersive X-ray mapping (Fig. 8e–h) further reveal that the CEI formed in the 1 M-LPF-DMDOHD electrolyte is also more uniform, dense and conformal (~5 nm), as compared to one formed in 1 M-LPF-DMC and 1 M-LPF-EC/DMC electrolytes. The composition and structural superiority of the CEI formed in the 1 M-LPF-DMDOHD electrolyte (Supplementary Fig. 31) explains thus the improved cycling stability analyses of the LiNi$_{0.5}$Mn$_{1.5}$O$_4$ electrodes (Fig. 6c, d, and Supplementary Fig. 13),

demonstrating again the advantages of the DMDOHD-based electrolyte.

## Further improvement of the high-voltage LM and Li-ion cells with co-solvent and additives

Although the DMDOHD-based electrolyte clearly exhibits improved lithium plating/stripping CE than traditional commercial electrolytes, the results remain beyond the practical reality−the average CE attained of 92% (Fig. 3c) remains modest as compared to technology required realm of 99.95% efficiency per cycle. It has to be nevertheless considered that these results are attained with single electrolyte constituents: pure solvent (DMDOHD), and single salt (LiPF$_6$) composition,

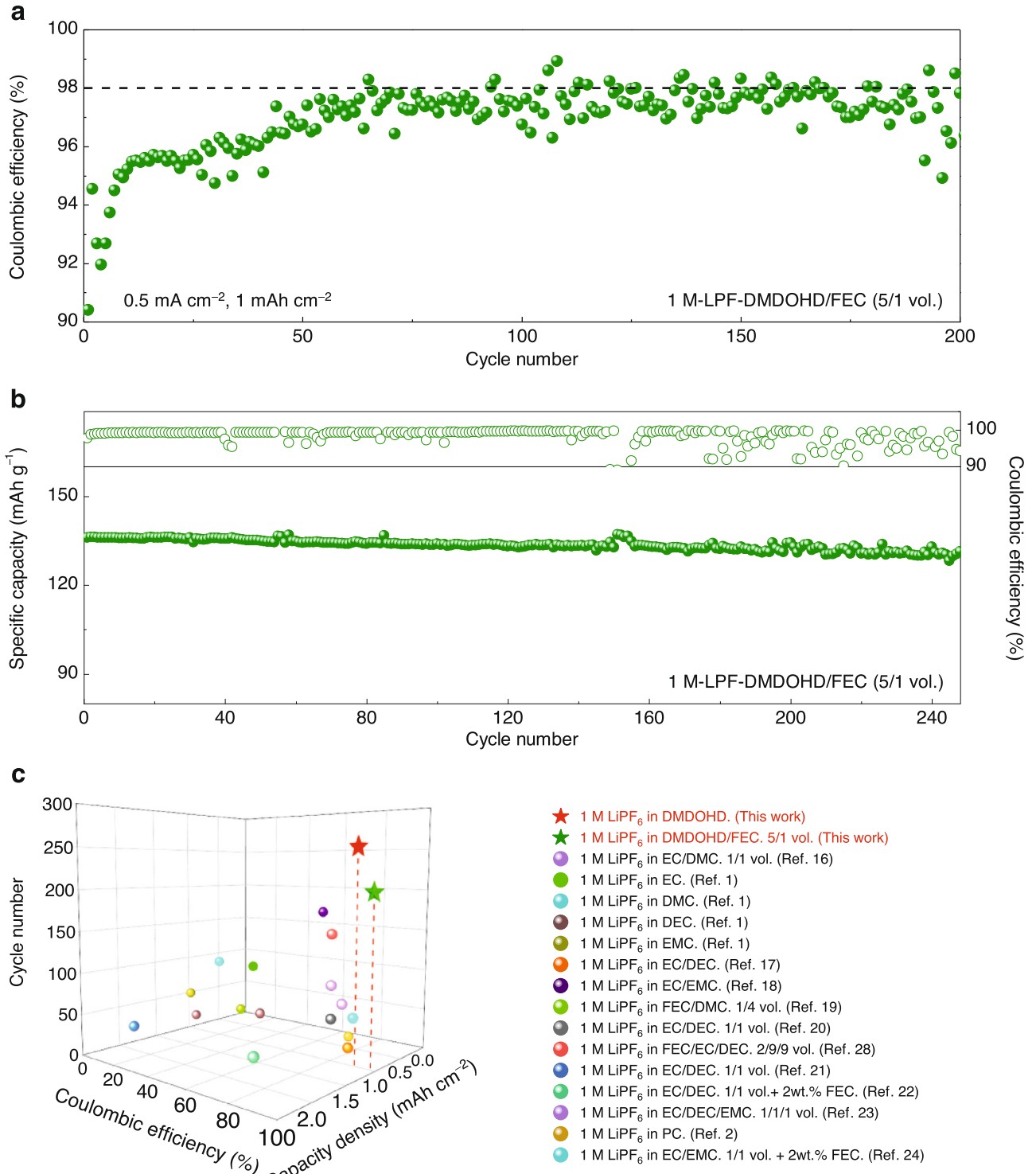

**Fig. 9 | Lithium plating/stripping behavior and electrochemical performances of LMBs using LiNi$_{0.5}$Mn$_{1.5}$O$_4$ as positive electrode material in DMDOHD/FEC-based electrolyte. a** Lithium plating/stripping Coulombic efficiency of the Li‖Cu coin cell using 1 M-LPF-DMDOHD/FEC-5/1 (vol.) electrolyte at a current density of 0.5 mA cm$^{-2}$ and a cutoff capacity of 1 mAh cm$^{-2}$. **b** Cycling stability of the Li‖LiNi$_{0.5}$Mn$_{1.5}$O$_4$ cell in 1 M-LPF-DMDOHD/FEC-5/1 (vol.) electrolyte at a rate of C/3. **c** Comparison of the lithium plating/stripping Coulombic efficiency and cycle life of the Li‖Cu cells with DMDOHD-based electrolytes as compared to previously reported 1 M LiPF$_6$ in carbonate formulated electrolytes. Note: The number of references cited in Fig. 9c pertains specifically to the references listed in the Supplementary Information.

that is known rare to provide high voltage stability coupled to Li-metal compatibility. As widely reported and also applied in commercialized cells[58,59], the introduction of co-solvents or additive can lead to improvements on the cycling stability and safety of Li-ion and also emerging LM cells. FEC is one such key co-solvent that can improve the

lithium plating/stripping CE of the electrolyte systems[58,60]. Therefore, we tested the impact of FEC as additive and as co-solvent for DMDOHD-based electrolyte on the lithium plating/stripping CE. As displayed by Supplementary Fig. 32, the lithium plating/stripping CE of the Li‖Cu cells with DMDOHD-based electrolyte gradually improves

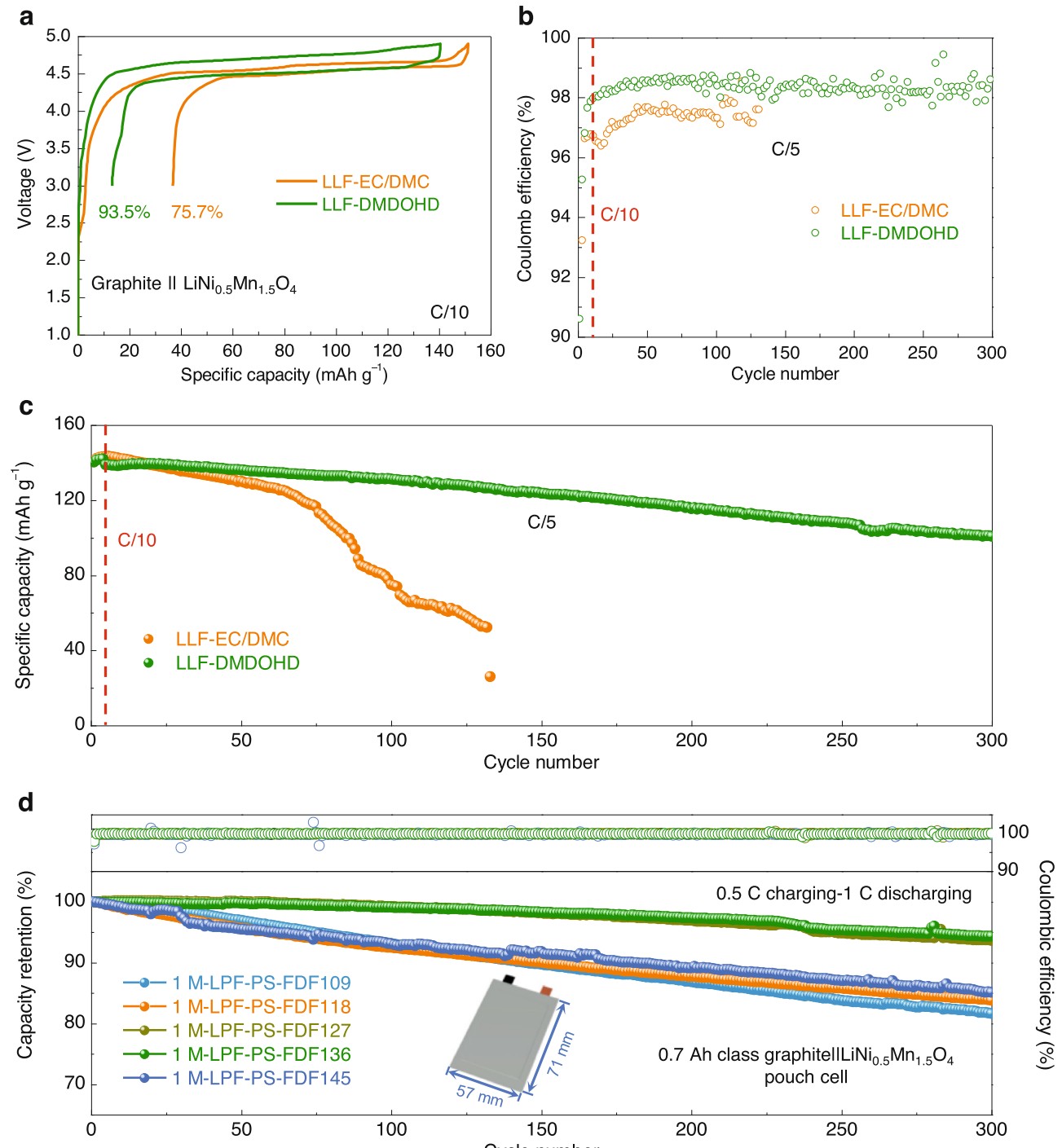

**Fig. 10 | Electrochemical performances of LIBs using LiNi$_{0.5}$Mn$_{1.5}$O$_4$ as positive electrode material and graphite as negative electrode material with DMDOHD-based electrolyte. a** The first galvanostatic charge-discharge curves of the graphite||LiNi$_{0.5}$Mn$_{1.5}$O$_4$ cells cycled in different electrolytes at a rate of C/10 (1 C corresponds to 142 mA g$^{-1}$ based on the mass of LiNi$_{0.5}$Mn$_{1.5}$O$_4$). LLF represents 1 M LiPF$_6$ + 0.5 M LiDFOB + 6wt.% FEC. Comparison of the (**b**) Coulombic efficiency and (**c**) cycle life of the graphite||LiNi$_{0.5}$Mn$_{1.5}$O$_4$ cells in different electrolytes (C/5 cycling rate). **d** Cycling stability of the graphite||LiNi$_{0.5}$Mn$_{1.5}$O$_4$ pouch cells in different electrolytes (0.5 C charging-1 C discharging) at 25 °C. FEMC and PS represent 2,2,2-trifluoroethyl methyl carbonate and 1,3-propanesultone, respectively. 1 M-LPF-PS-FDFxyz (x + y + z = 10) represents 1 M LiPF$_6$ + 4 wt.% PS in FEC/DMDOHD/FEMC (x/y/z, vol.).

with the increased content of FEC. Specifically, the average CE within 100 cycles augments from 92.54% with 1 M LPF-DMDOHD/FEC (15/1, vol.) to 96.41% with 1 M LPF-DMDOHD/FEC (5/1, vol.). However, upon further addition of FEC (e.g., DMDOHD/FEC, 4/1 by vol.) the average CE decreases to 96.05%, with more serious degradation upon further FEC content increase (cells with DMDOHD/FEC, 3/1, 2/1, and 1/1 vol eventually failed at the first cycle). This analysis indicates that the optimal

DMDOHD/FEC-based composition is 1 M-LPF-DMDOHD/FEC (5/1 by vol.).

Figure 9a and Supplementary Fig. 33 display the lithium plating/stripping CE and voltage profiles of the Li||Cu cell cycled in the optimally found 1 M-LPF-DMDOHD/FEC (5/1 by vol.) electrolyte formulation at a current density of 0.5 mA cm$^{-2}$ with a cutoff capacity of 1 mAh cm$^{-2}$. Compared to the baseline DMDOHD electrolyte without

FEC (Fig. 3c), the 1 M-LPF-DMDOHD/FEC-5/1 electrolyte achieves an improved CE of 98% after approximately 60 cycles of activation and remains stable for another 200 cycles, which, to the best of our knowledge, is the best performance reported to date using low concentration carbonate-based electrolyte considered with, and without additives (Fig. 9c, Supplementary Table 3). Besides, the Li‖LiNi$_{0.5}$Mn$_{1.5}$O$_4$ cell with DMDOHD/FEC-5/1-based electrolyte also displays improved cycling stability sustaining more than 250 cycles with a capacity retention ratio of ~97% at a rate of C/3 (Fig. 9b), which is also superior to most of the previously reported carbonate-based electrolytes (Supplementary Table 4).

Complementary to J. Dahn recent report[27], the DMDOHD based electrolyte was also optimized to stabilize graphite negative electrode material interphase and allow assembly of high-voltage Li-ion cells with LiNi$_{0.5}$Mn$_{1.5}$O$_4$ as positive electrode material. Considering that different SEI properties are required at the Li-metal as compared to graphite surface, with the critical one being the ability to avoid solvent co-intercalation, a different electrolyte formulation was found optimal in this case. For the graphite‖LiNi$_{0.5}$Mn$_{1.5}$O$_4$ full cell, the best composition found is − 1 M LiPF$_6$, 0.5 M lithium difluoro(oxalato)borate (LiD-FOB), and 6 wt.% FEC in DMDOHD (marked as LLF-DMDOHD). For comparison purposes, same salt composition was also used in EC/DMC solvent (marked as LLF-EC/DMC). Galvanostatic cycling data show net better performances of DMDOHD-based electrolyte as compared to EC/DMC formulation, including higher CE at first cycle, and average at the following cycles (Fig. 10a, b), as well as better capacity retention (~73%) over more than 300 cycles (Fig. 10c). These results further complement on the versatility of DMDOHD electrolyte solvent for the emerging battery applications.

Given the versatility of DMDOHD in LMBs and LIBs, we also explored its potential as electrolyte solvent for lithium-sulfur batteries. As depicted in Supplementary Fig. 34, when operating at a low C-rate of 0.02 C and discharging to approximately 1 V (vs. Li/Li$^+$), the lithium-sulfur cell with the 1 M-LPF-DMDOHD electrolyte was found able to attain a specific discharge capacity of approximately 1360 mAh g$^{-1}$ during discharge, accompanied by the typical "two-plateau" feature, which corresponds to the successive reduction of elemental sulfur to long-chain polysulfides and subsequently to short-chain lithium sulfide. This behavior stands in stark contrast to the lithium-sulfur battery employing the conventional carbonate electrolyte (1 M-LPF-EC/DMC), which exhibits only one plateau in the high voltage region (~2.4 V) associated to the reduction of elemental sulfur to long-chain polysulfides, and a considerably lower specific capacity of approximately 565 mAh g$^{-1}$. However, similar to the conventional carbonate electrolyte, the DMDOHD-based electrolyte also failed to reverse the electrochemical redox reaction of elemental sulfur, even by polarizing the cell up to 3.5 V (vs. Li/Li$^+$). As authors' perspective, although the DMDOHD electrolyte demonstrates favorable compatibility with lithium metal and has the potential to slow down the dissolution of polysulfides given higher viscosity (thereby reducing the shuttling rate of the polysulfides), it remains ineffective in inhibiting the dissolution of polysulfides and against the nucleophilic attack by polysulfides. Therefore, akin to conventional carbonate-based electrolytes[61,62], DMDOHD can undergo chemical reactions with polysulfides (e.g., polysulfides react with DMDOHD via nucleophilic addition or substitution reactions), resulting in an irreversible sulfur redox reaction within the lithium-sulfur battery system.

Finally, to clarify whether DMDOHD has any potential value for practical application, we assembled 0.7 Ah class graphite‖LiNi$_{0.5}$Mn$_{1.5}$O$_4$ pouch cells (with an N/P ratio of approximately 1.08) and evaluated the electrochemical performances using DMDOHD as the single electrolyte solvent. As illustrated in Supplementary Figs. 35 and 36, the graphite‖LiNi$_{0.5}$Mn$_{1.5}$O$_4$ pouch cell employing 1 M-LPF-DMDOHD was able to maintain >63% of its initial capacity after 300 cycles under the testing conditions of 0.3 C charging-0.5 C

discharging. It's worth noting that the improvement in the electrochemical performance of graphite‖LiNi$_{0.5}$Mn$_{1.5}$O$_4$ full cells when using DMDOHD as the single electrolyte solvent appears to be less significant as compared to Li‖LiNi$_{0.5}$Mn$_{1.5}$O$_4$ cells. As previously mentioned, this difference is attributed to the distinct requirements for SEI properties on lithium metal and graphite surfaces wherein the graphite‖LiNi$_{0.5}$Mn$_{1.5}$O$_4$ system demands in particular an electrolyte system capable of avoiding solvent co-intercalation. Moreover, it's important to recall that the use of DMDOHD as an electrolyte's single solvent results in lower ionic conductivity. This limitation is another discouraging factor for its application as the sole electrolyte solvent in practical scenarios. Hence, we adopted 2,2,2-trifluoroethyl methyl carbonate (FEMC, η = 1 mPa•s) as the main solvent, FEC as the co-solvent and 1,3-propanesultone (PS) as an electrolyte additive to establish a base electrolyte system (1 M-FEC/FEMC (1/9 vol.) + 4 wt.% PS) and then explored the potential of DMDOHD to enhance the electrochemical performance of graphite‖LiNi$_{0.5}$Mn$_{1.5}$O$_4$ pouch cells when employed as an electrolyte co-solvent. Here we abbreviate the electrolyte system, comprising 1 M LPF + 4 wt.% PS in FEC/DMDOHD/FEMC (x/y/z, vol.), as 1 M-LPF-PS-FDFxyz (x + y + z = 10).

As illustrated in Fig. 10d and Supplementary Fig. 37, the cycling stability of the graphite‖LiNi$_{0.5}$Mn$_{1.5}$O$_4$ pouch cells exhibits significant improvement as the volume ratio of DMDOHD increases from 0% to 30% within the FEC/FEMC-based electrolyte system. After 300 cycles under the test condition of 0.5 C charge-1 C discharge, the discharge capacity is retained at more than 94% of the initial capacity. Even when the volume ratio of DMDOHD reached 40%, the cycling stability remained superior to that of the system without DMDOHD, albeit with a reduction compared to the 30% volume ratio system. Additionally, we conducted rate capability tests on the graphite‖LiNi$_{0.5}$Mn$_{1.5}$O$_4$ pouch cell with a 30% volume ratio of DMDOHD in the FEC/FEMC-based electrolyte system, varying the discharge rate from 0.1 C to 2 C. The results, shown in Supplementary Fig. 38, reveal excellent rate capability of the graphite‖LiNi$_{0.5}$Mn$_{1.5}$O$_4$ pouch cell with this electrolyte composition. Specifically, the rate capability ($Q_{2C}/Q_{0.1C}$) reached 93%, and the discharge capacity at 2 C was approximately 0.589 Ah, indicating an impressive utilization rate of ~84% at a high C-rate. Further, we also tested the high temperature electrochemical performance of the graphite‖LiNi$_{0.5}$Mn$_{1.5}$O$_4$ pouch cells with and without DMDOHD as an electrolyte co-solvent. As illustrated in Supplementary Figs. 39 and 40, the pouch cell employing 30% volume ratio of DMDOHD within the FEC/FEMC-based electrolyte system demonstrates excellent cycling stability, with a capacity retention of ~86% over 300 cycles, and an average CE close to 100% at a C-rate of 1 C under 45 °C, which is superior to that observed in the cell without DMDOHD as an electrolyte co-solvent (~80% capacity retention after 243 cycles). Although certainly further developments and validations tests may be required, the above combined results consolidate the practical application value of DMDOHD as an electrolyte solvent or co-solvent, for target specific applications.

## Discussions

A new electrolyte system suitable for high-voltage 5 V-class LMBs is disclosed. The simple formulation of 1 M LiPF$_6$ dissolved in DMDOHD solvent results in an electrolyte displaying key advantages as compared to the conventional carbonate-based electrolytes applied for high-voltage LMBs. Specifically, in: (1) attaining high lithium plating/stripping CE; (2) dendrite-free and dense lithium deposition; (3) high anodic stability, up to 5.2 V (vs. Li/Li$^+$); (4) enhanced thermal stability; and (5) improved interphase chemistry. Benefiting from these combined effects, a Li‖LiNi$_{0.5}$Mn$_{1.5}$O$_4$ cell operated with this electrolyte shows a high-capacity retention of ~97% after 250 cycles, which represent a considerable advance as compared to commercial, as well as most of the previously reported carbonate-based electrolyte formulations. These improvements are the result of the particular

chemical structure of the DMDOHD solvent, the interaction with Li cation and the solvation environment, resulting in efficient interphase formation at both, positive and negative electrode surfaces, and thus limited degradation. Upon further fine optimization, the use of DMDOHD as sole solvent, with additives, or as co-solvent, is found to enable high performance metrics of 5 V-class LMBs but also Li-ion cells, confirming the high versatility of this particular solvent chemistry. We believe that the performances of this electrolyte system for 5 V-class next generation batteries, can be further improved by for example increasing the concentration of lithium salt or use of multi-salts formulations, use of specific additives, and electrolyte thinners by exploiting the full potential of DMDOHD solvent, or by exploring other versatile dicarbonate chemistries.

## Methods

### Chemicals and materials
The Li metal chips (diameter: 15.6 mm; thicknesses: 1 mm), coin cell spares (stainless steel, SS-316), current collector (Al foil and Cu foil), conductive carbon and poly (vinylidene difluoride) (PVDF) were purchased from TOB new energy technology Co., Ltd. (Xiamen, China). Glass fiber separator (GF/D) was purchased from Whatman. Dimethyl carbonate (anhydrous, >99%, Sigma-Aldrich), DMDOHD (>98%, Tokyo Chemical Industry), FEC (>98%, Tokyo Chemical Industry), and 2,2,2-trifluoroethyl methyl carbonate (>98%, Tokyo Chemical Industry) were chosen as electrolyte solvents and used after drying over molecular sieves (4 Å, Sigma-Aldrich) to ensure a water content of less than 5 ppm (determined by Karl Fischer titration, Metrohm 899 Coulometer). 1,3-propanesultone (>99%, Tokyo Chemical Industry) was chosen as a film-forming additive in pouch-type full-cells. The $LiPF_6$ salt (battery grade, DoDochem) was dried under vacuum at 100 °C for 12 h in an argon-filled glovebox before use. 1 M $LiPF_6$ in EC/DMC (battery grade) was purchased from DoDochem.

### Coin-cell assembly and electrochemical measurements
The $LiNi_{0.5}Mn_{1.5}O_4$ positive electrode was prepared by mixing $LiNi_{0.5}Mn_{1.5}O_4$ powder, conductive carbon and PVDF binder in a weight ratio of 85:8:7 in N-methyl-2-pyrrolidinone (NMP) to form a homogeneous slurry. The slurry was coated on an Al foil and dried at 120 °C under vacuum for 12 h. Afterwards, circular electrodes of 12 mm in diameter were cut, pressed, dried for additional 12 h at 120 °C in a vacuum oven and weighted. The mass loading of the tested $LiNi_{0.5}Mn_{1.5}O_4$ electrode was in the range of 10–16 mg cm$^{-2}$. CR2032-type coin cells were assembled with a Li metal foil as counter and pseudo-reference electrode, one sheet of glass fiber separator soaked with the selected electrolyte, and one $LiNi_{0.5}Mn_{1.5}O_4$ sheet as the working positive electrode in an argon-filled glovebox (MBraun, Inc., $H_2O < 1$ ppm, $O_2 < 1$ ppm). The galvanostatic charge−discharge tests were performed on a Neware battery testing system (Shenzhen Neware Electron. Co. Ltd., China) within the potential range of 3.0−4.9 V or 3.0−5.0 V (vs. Li/Li$^+$). The cycling performance of the Li||$LiNi_{0.5}Mn_{1.5}O_4$ cells was recorded after several cycles of formation at a C-rate of C/10. The specific capacity and current density are calculated and reported based on the mass of $LiNi_{0.5}Mn_{1.5}O_4$ active material component.

The ionic conductivity of the electrolytes was measured using AC impedance spectroscopy (SP300 potentiostat, Biologic). A polypropylene gasket with an internal diameter of 11.3 mm and an outer diameter of 16.4 mm was prepared using a 3D printer (Ultimaker 3) and placed between two stainless steel disks in a CR2032 coin cell setup. The cell was filled with the studied electrolyte, and the pressure between the disks and the gasket was applied by a metallic spring. The ionic conductivity was calculated using the equation: $C = L/(R \cdot A)$; wherein R is the resistance of the electrolyte (estimated from the EIS plot), L is the separation between the two metal plate electrodes (0.6 mm), and A is the contact area (1 cm$^2$).

Symmetric Li||Li coin cells were used to explore the lithium deposition behavior on lithium substrate and the cycling performance. A gasket with an internal area of 0.5 cm$^2$ was cut from a 45 μm thick high-density polyethylene (HDPE) foil (Scancell Folien-Vertriebs GmbH) and added on top of the lithium metal electrode on the positive side. HDPE was used since it is electrochemically stable in all tested electrolytes, as well as in contact with Li metal. A stainless steel (SS-316) disc and spring were added to the lithium chip to guarantee a uniformly distributed pressure. The cell components were then crimped inside a CR2032 coin cell case (SS-316). The galvanostatic charge−discharge tests were performed on a Neware battery testing system (Shenzhen Neware Electron. Co. Ltd., China) after a rest time of 12 h to allow complete wetting of the electrodes and the separator.

The CE of Li||Cu cell was evaluated using CR2032 coin-type cells. The same gasket as we used in the Li||Li cell with an internal area of 0.5 cm$^2$ was also added to the Cu electrode on the positive side. After that, a Celgard separator, a lithium metal chip, stainless steel (SS-316) disc, and spring were added sequentially.

LSV experiments were conducted using an electrochemical workstation (CHI760E, CH Instruments, Shanghai, China) with a three-electrode electrochemical cell setup. A glassy carbon (GC) disk electrode with 3 mm diameter was used as the working electrode, while two lithium metal foils were used as the counter and reference electrodes, respectively. The GC electrode was polished with 0.5 μm and 50 nm alumina powder successively before the tests.

### Graphite||LNMO pouch cell assembly and electrochemical test
The pouch-type full cells were used to evaluate the practical application value of DMDOHD as an electrolyte co-solvent. The positive electrode comprises LNMO, PVDF (Kynar® HSV 1800), and carbon black (Super P) in a mass ratio of 97.5:1:1.5. Slurries of positive electrode were coated on both side of Al foils, corresponding to a mass loading of LNMO of 28 mg cm$^{-2}$, providing an areal capacity of approximately 4 mAh cm$^{-2}$. The negative electrode was formulated with graphite (GCP-80), PVDF (Kynar® HSV 1800), and carbon black (Super P) in a mass ratio of 90:5:5. The slurry was coated on both side of Cu foils, the mass loading of graphite being around 11.5 mg cm$^{-2}$, corresponding to an areal capacity of 4.3 mAh cm$^{-2}$. The negative/positive (N/P) electrode areal capacity ratio stands at approximately 1.08. The dimensions of the positive and negative are 53 mm * 67 mm and 55 mm * 69 mm, respectively. Both positive and negative electrodes were dried overnight under vacuum at 80 °C before cell assembly. We employed microporous polyethylene (PE) (Celgard 2400) as the separator. The amount of electrolyte added into the pouch cell was 4.2 g Ah$^{-1}$. For the 0.7 Ah class pouch-type Graphite||LNMO full-cells assembly, five positives and six negative electtode sheets were utilized. The galvanostatic charge−discharge tests were performed on a Neware battery testing system (Shenzhen Neware Electron. Co. Ltd., China) within the voltage range of 2.8−4.85 V at 25 °C or 45 °C, the real capacity ($C_0$) of 0.7 Ah class pouch-type Graphite||LNMO full-cells is calculated based on the capacity calibration in the first cycle at a C-rate of C/3, and the cycling current rate is 0.5 C charging and 1 C discharging at 25 °C or 1 C charging and 1 C discharging at 45 °C, the current rate is calculated based on $C_0$.

### Physico-chemical characterization
XPS measurements were performed on a Thermo Scientific Escalab 250Xi+ spectrometer with a monochromatized and micro-focused Al Kα X-ray source. The samples were transferred using a sealed chamber with no exposure to ambient air. The surface morphology of the Li metal negative electrode was investigated by using field emission scanning electron microscopy (FESEM, HITACHI S-4800), with a short exposure of samples to ambient air during transfer (of less than 10 s). The surface morphology of cycled $LiNi_{0.5}Mn_{1.5}O_4$ was investigated by

using and a TECNAI HRTEM (F30). The in-situ dynamic lithium deposition behavior was monitored using a ZOOM-0850C optical microscope (SHANGHAI PUQIAN OPTICAL INSTRUMENT CO., LTD.). Raman spectra were recorded on a XploRA confocal Raman microscope (Jobin YvonHoriba, France) with an excitation wavelength of 785 nm. The $^7Li$ NMR spectra were recorded at room temperature (295 K) on a Bruker Ascend 600 MHz spectrometer. For NMR measurements, 400 μL of electrolyte samples were placed in the outer tube of a 5-mm NMR Wilmad coaxial cannula, while the inner tube contained the reference solution of 1 M LiCl in $D_2O$.

## Computational details
Density functional theory (DFT) calculation. The energies of the HOMO and the LUMO were calculated using the DFT implemented in Gaussian 09 software. The calculations were performed using the B3LYP, M05-2X, wB97XD and G4MP2 functionals with a basis set of 6-311 + G(d, p). Here we chose the polarizable continuum model (IEFPCM) and the solvation model density (SMD) to take into account the solvent effect, where the SMD is considered more suitable for organic solvents used in batteries due to the fact that more accurate results can be attained by taking into account not only the electrostatic interactions between the solute and the solvent in the polarizable continuum model, but also the inclusion of the non-electrostatic interactions (cavity-dispersion-solvent-structure term)[63]. The selection of the implicit solvent model was determined based on the dielectric constants of the studied carbonate molecules. For linear carbonate molecules such as DMC (ε = 3.12), EMC (ε = 2.93), DEC (ε = 2.82), and DMDOHD (ε = 2.9), their dielectric constants exhibit minimal variation and are closely aligned (around 3), thus ethyl acetate which has a similar structure and a similar dielectric constant (ε = 5.99) was used as the implicit solvent model. While the cyclic carbonate molecule EC possesses a significantly higher dielectric constant of 89.6, prompting the adoption of water as an implicit solvent model, given its similarly large dielectric constant of 78.5. Additionally, the effect of other implicit solvent models, such as acetonitrile (ε = 37.5), methanol (ε = 33.7), acetone (ε = 20.5) and diethylether (ε = 4.22), with varying dielectric constants on the energy levels of the studied carbonate molecules has been also explored.

The reduction potential of DMDOHD with $Li^+$ and oxidation potential of DMDOHD with $PF_6^-$ were calculated by using the Nernst equation, referring to the methodology of previously published work as follows:

Electrochemical stability of the complex M was defined using the thermodynamic energy cycles relative to the $Li^+/Li$ scale according to Eqs. (1) and (2)[64–66].

$$E_{ox}(M) = [\triangle G_i + \triangle G^0_S(M^+) - \triangle G^0_S(M)]/F - 1.4 \quad (1)$$

$$E_{red}(M) = -[\triangle G_a + \triangle G^0_S(M^-) - \triangle G^0_S(M)]/F - 1.4 \quad (2)$$

where in $\triangle G_i$ is the ionization free energy and $\triangle G_a$ is the electron affinity free energy in gas-phase at 298.15 K; $\triangle G_S(M^+)$, $\triangle G_S(M^-)$ and $\triangle G_S(M)$ are the solvation free energies of the oxidized, reduced and initial complexes, respectively; F is the Faraday constant. A shift factor of 1.4 was used to account for the difference between the absolute redox potential scale and redox potential of $Li^+/Li$ couple. The shift factor depends on the nature of solvent and salt, as well as salt concentration, and can vary by 0.1–0.3 V given the variation of the lithium free energy of solvation in various solvents[64,66]. For the calculations, B3LYP/6-311 + G(d, p) or M05-2X/6-31 + G(d,p) was used for structure optimization, while M05-2X/6-31 G(d) was employed for solvation energy calculations with SMD (acetone, ε = 20.5).

Classic MD simulations: The ground state molecular and ions geometries were optimized using the DFT method at the B3LYP/6-

311 G + (d, p) level. All DFT calculations were performed using the Gaussian 09 software package. The MD simulations were conducted using the GROMACS 2018 program[67]. The molecules and ions were described using the optimized potentials for liquid simulations all-atom (OPLS-AA) force field[68]. The partial charges on the atoms of the solvent were computed by fitting the molecular RESP at the atomic centers using the B3LYP/aug-cc-pVDZ basis set[69]. The simulation boxes were cubic with a side length of approximately 4 nm and contained 1 M $LiPF_6$ solvated in EC/DMC and DMDOHD solvents. During the simulations, the temperature was maintained at 300 K using a Nosé-Hoover thermostat with a relaxation time of 0.2 ps, and the pressure was controlled at 1 bar using a Parrinello-Rahman barostat with a relaxation time of 2.0 ps. For a duration of 40 ns, and the last 20 ns of the simulations were utilized for analysis.

## Data availability
The data that support the findings of this study are available from the corresponding author upon reasonable request.

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

## Acknowledgements

The authors acknowledge China Scholarship Council [X.Z., H.X., J.W., Y.Z., Z.C.], F.R.S.-FNRS through grant N° - F.4552.21 – DEMIST [A.V., J.F.G.], CF-ARC grant (18/23-093) MICROBAT [A.V.], European Research Council (ERC) under the European Union's Horizon 2020 research and innovation program (grant agreement No. 770870) [A.V.], Marie Skłodowska-Curie Actions Postdoctoral Fellowships (grant agreement No. 101064286) [X.L.], LC-BAT-5-2019 HYDRA (grant agreement No. 875527) [A.V.], and the Projects of National Natural Science Foundation of China (U1805254 [Q.D.], 21773192 [M.Z.], 22072117 [R.Y.], 22201209 [W.S.]) for financial support.

## Author contributions

A.V., Q.D., X.L. and X.Z. conceived the project. X.Z., P.X., J.D. and X.L. designed experiments and analyzed results. X.Z., P.X., J.D., X.L., H.X., W.D., Q.Z. and S.Y. performed experiments. J.S., W.S. and R.Y. designed and performed the computational studies. J.W., Y.Z., Z.C., M.Z. and J.F.G. helped with discussion. A.V., J.F.G. and Q.D. supervised the project implementation. X.Z., X.L. and A.V. wrote the manuscript. All authors contributed to finalizing the manuscript.

## Competing interests

The authors declare no competing interests.
