## [Peer Review File · Nature Communications]

A Dicarbonate Solvent Electrolyte for High Performance 5 V-Class Lithium-based BatteriesEditorial Note: Parts of this Peer Review File have been redacted as indicated to remove third-party material where no permission to publish could be obtained.

REVIEWER COMMENTS

Reviewer #1 (Remarks to the Author):

This paper reports the use of DMDOHD as an electrolyte solvent for 5 V class Li metal cells. The work on the stability of the Li metal anode is decent. But, the paper has many flaws and over selling to try to get the paper into Nature Commun.

Let's talk about some good points first. I like Figure 1 of the manuscript and also Figure S2. These may help to give a clue as to why DMDOHD is performing well. The XPS measurements are extensive and appear to be well-carried out.

Now let's talk about some bad points.

First - authors say the conductivity of electrolytes with DMDOHD is "slightly" lower than that of conventional electrolytes. In fact the numbers they present show it is basically 50 times lower than an electrolyte commonly used in LiBs. This means battery rate capability will be abysmal at room temperature. Authors simply cannot present a paper like this without acknowledging the truth.

Then, Authors state: As shown in Supplementary Fig. 4, the 1 M-LPF-DMDOHD electrolyte shows significantly higher anodic stability compared to the reference carbonate-based electrolytes, as evidenced by the much lower anodic current, even when polarized up to 5.5 V (vs. Li/Li+)

Well in S4 - The data for EC/DMC and DMC look very strange. It would be nice if these results had been repeated or if a second anodic sweep had been included. Also, authors need to consider the incredible low ionic conductivity of the electrolyte and how it will influence the CV results. This cannot be ignored. Currents will be small due to poor transport properties, not only because of better anodic stability. The same considerations need to be made about the leakage currents later in the same paragraph.

Authors state: Meanwhile, the DMDOHD-based electrolyte also exhibits a wide liquid range down to -70°C , which may have the potential for low temperature applications.

Authors have no connection to reality. The conductivity will be so incredibly low that cells simply will not function.

Figure S21 is quite impressive and could be moved to the main text. One can clearly see in this Figure the large charge discharge polarization for the DMDOMD cells compared to the standard electrolyte cells. This is due to the poor conductivity of the DMDOMD. This data is collected at room temperature. If authors want to publish in a high impact journal, they MUST show test results at elevated temperature. It is easy to make things work at room temperature.

Figure S20 also shows the large polarization in the Li/Li cells due to poor ionic conductivity.

Overall, this paper has value, but it is oversold in order to be published in Nature Comm. Authors should make the paper more realistic and more truthful, Maybe then it could be suitable.

Reviewer #2 (Remarks to the Author):

Comments on the DFT calculations:

Model chemistry is PCM(???) / B3LYP/6-311+G(d,p), what is the implicit solvent model used here? The default is water which I do not think is appropriate - at least by itself. What is the justification for using B3LYP over functionals like M05-2X or wB97XD or even the G4MP2 composite method?

Since permittivity of the electrolyte is not known, it would be reasonable to consider a few options like diethylether, acetone, acetonitrile/methanol, and water. How does the potential vary with permittivity?

HOMO/LUMO is not necessarily reflective of the actual redox stability of the materials at battery-relevant potentials due to deformation of the molecule in the reduced or oxidized state (or bond breaking). A more insightful approach the authors should consider is the explicit calculation of reduction and oxidation potentials using the Nernst equation. Because the DMDOHD molecule is assumed to contribute to SEI, the reduction potential should be computed with Li⁺ present as well.

The authors should conduct conformer analysis for solvents with rotatable bonds, it looks like the authors have shown only the linear/extended conformer of DMDOHD in Supp Table 5 which is not the lowest energy conformer I get from a simple search in RDKit. Since this structure is used as an input to the BOMD simulations, used to justify the increase in Li-PF₆ ion pairing, and to justify the participation of DMDOHD in the SEI, this needs to be addressed. Some of the low energy conformers look like they could potentially coordinate Li⁺ with both C=O groups (cis-like conformers). This may or may not change the extent of ion pairing relative to that claimed by the authors but will certainly influence the reduction potential of the solvent and its subsequent participation in the SEI. As an example, the coordinates of the lowest energy conformer + Li from a not-exhaustive set of gas phase calculations with M05-2X/6-31+G(d,p) are below. The binding free energy between Li⁺ and the solvent here is ~25 kcal/mol lower than the linear/extended conformer.

In case of formatting issues, given as [atom symbol, X, Y, Z] like Xmol/XYZ format.

O 0.4294470000 -2.7958520000 -1.3194130000

O -0.2846170000 0.0457890000 -3.1548250000

O 0.5156800000 -1.8928660000 0.7619330000

O 1.1131100000 -0.0546540000 -1.4853170000
O -0.9009860000 -3.4615450000 0.3613350000
O -0.2266930000 1.7237540000 -1.6348830000
C 1.4462310000 0.1947660000 -0.0936820000
C 1.7309310000 -1.1576270000 0.5151430000
C 0.0430290000 -2.7255760000 -0.1509620000
C -1.5357560000 -4.4237040000 -0.5188570000
C 0.1468170000 0.5904540000 -2.1469910000
C -1.2811900000 2.4362610000 -2.3552230000
Li 0.5842880000 -1.6733040000 -2.7804090000
H 0.6130260000 0.6936890000 0.3964860000
H 2.3950220000 -1.7443580000 -0.1186840000
H 2.3342570000 0.8240650000 -0.0680220000
H -0.7854420000 -5.1037870000 -0.9130400000
H -2.0450440000 -3.8990160000 -1.3237660000
H 2.1765110000 -1.0295600000 1.4989550000
H -2.2448990000 -4.9454110000 0.1127390000
H -0.9297140000 2.6597850000 -3.3581930000
H -2.1708370000 1.8137900000 -2.3847020000
H -1.4431610000 3.3385170000 -1.7783430000

Comments on the Born-Oppenheimer MD simulation:

Firstly, there is critical information to reproduce these simulations missing from the methods section.

a.) What version of the VASP code used?

b.) If different from the version of the VASP code, give the version of the pseudopotentials used, e.g., 5.2 or 5.4, etc.

c.) What variants of the pseudopotentials were used? `_sv` for Li? etc.

d.) What is the functional used here? Is dispersion included, if so which model?

e.) What is the cutoff used? 400 eV? 520 eV?

f.) Were simulations performed using the Gamma point only? Some other kpoint scheme?

g.) What mass of hydrogen is used here? 2 fs timestep isn't appropriate for BOMD unless you only have heavy atoms (>He mass), hydrogen masses should be 0.5 fs or less, deuterium and tritium masses are safer for 1 fs timestep. You can accumulate significant errors in the energy and forces using too long a timestep.

h.) What procedure was used to prepare BOMD simulations after randomly packing the box and before the 60 ps run? Do you equilibrate the structure with a force field based simulation (if so, give details), only perform DFT relaxation (if so, give details), or just run without any further processing (if so, mention this)?

The authors can strengthen their claims about differences in ion pairing between the base carbonate electrolyte and the designed one using Raman spectroscopy (740 cm^{-1} for solvent separated ion pairs – 750 cm^{-1} for contact ion pairs) or Li/F NMR. Alternatively, the authors could perform a simulation with

many LiPF₆ pairs using an appropriate force field for 10s of nanoseconds. BOMD on these timescales (60-70 ps) and from a single trajectory for a single ion pair cannot provide a strong statistical justification for the average coordination environment around Li in the electrolyte – especially when attempting to correlate the formation of contact ion pairs and aggregates to LiF generation.

General comments about novelty:

The single salt, single solvent concept is great and appears to match or improve reversibility of LNMO|Li full cells among near peers from the last 5-6 years. I appreciate the authors' pursuit of practical electrolyte design, though unfortunately the authors have inherited some of the bad practices common to battery research which I think detract from the novelty and impact of this work. Mass loading of LNMO is 10 mg/cm² in full cells, while practical mass loading is closer to ~25 mg/cm². I don't know the lithium foil thickness in the full cells. However, the Li|Cu cells use 100-200 micron thick Li reservoirs with a very low per cycle depth of discharge. If the authors saw comparable performance under more rigorous and practical conditions, this would be a higher impact paper.

Reviewer #3 (Remarks to the Author):

In this work the authors have introduced the use of an electrolyte system compatible with high-voltage 5 V-class Lithium metal batteries. The use of dimethyl 2,5-dioxahexanedioate (DMDOHD) solvent with a LiPF₆ has been shown to have

several advantages as compared to the traditional carbonate-based electrolytes in terms of superior lithium plating/stripping characteristics, Coulombic efficiency; suppression of dendrites and dense lithium deposition with an anodic stability, up to 5.5 V vs. Li/Li⁺. Enhanced performance of cells with Li metal anode and LNMO cathodes supports the claim owing to the electrolyte's enhanced interfacial forming capability and solvation structure. The authors have also acknowledged that the solvent has been previously tested as an additive in previous works. However it is the first time DMDOHD has been used as a single solvent for a 5v class cathode.

1. Since the authors have managed to bring back the use of carbonate based solvents for use in Li metal anode batteries, I would insist that the authors take this opportunity to test these electrolytes in the much vaunted Li/S batteries and the stability of such electrolytes towards polysulphides. I therefore suggest the authors to add some comments or run sum preliminary test and add their views in the manuscript.

2. CEI composition schematic needs to be provided for better understanding the difference of the CEI composition with the different electrolytes involved.

3. The flame test is not convincing to conclude that these electrolytes are safe and non flammable as the video shows a different scenario.

4. I do not find a c-rate capability test for the electrolytes under study. This needs to be incorporated in the manuscript or supplementary info to have an idea about the recovery of specific capacity.

Reviewer #1:

Comments:

This paper reports the use of DMDOHD as an electrolyte solvent for 5 V class Li metal cells. The work on the stability of the Li metal anode is decent. But, the paper has many flaws and over selling to try to get the paper into Nature Commun.

Response: We thank the reviewer for the objective assessment and insightful comment on our manuscript. We have improved our work based on the following detailed comments proposed by the reviewer, and are confident that the revised version will meet the reviewer's requirements.

1. Let's talk about some good points first. I like Figure 1 of the manuscript and also Figure S2. These may help to give a clue as to why DMDOHD is performing well. The XPS measurements are extensive and appear to be well-carried out.

Response: We appreciate the reviewer for his/her positive feedback regarding Figure 1, Figure S2 and the XPS measurements.

2. Now let's talk about some bad points.

First - authors say the conductivity of electrolytes with DMDOHD is "slightly" lower than that of conventional electrolytes. In fact the numbers they present show it is basically 50 times lower than an electrolyte commonly used in LiBs. This means battery rate capability will be abysmal at room temperature. Authors simply cannot present a paper like this without acknowledging the truth.

Response: We thank the reviewer for the critical and valuable comments on our manuscript and we apologize for our less rigorous statement. We do indeed agree that we over-hyped the results, maybe because of being too excited about overall results attained in this work. And indeed, we also agree that the low ionic conductivity may be probably the weakest point of this electrolyte systems, yet still not being so low. In fact, the conductivity is comparable to ionic liquids, that have been also used and studied for battery applications, so from this point of view the system is actually quite interesting.

But agree, we have tempered the claims, make it more realistic and less confusing, and the relevant text has been revised as follows:

“The 1 M-LPF-DMDOHD electrolyte shows lower ionic conductivity of 0.21 mS cm^{-1} at $20 \text{ }^\circ\text{C}$ than that of the 1 M-LPF-DMC (2.6 mS cm^{-1}) and 1 M-LPF-EC/DMC (7.8 mS cm^{-1}) electrolytes, which is due to higher viscosity and ion pairing as compared to the short chain carbonates (Supplementary Tables 6 and 8). Such a low ionic conductivity may render this electrolyte unable to perform effectively at high C-rates under room temperature conditions, which will be discussed in detail in the latter text. However, this value is similar to or even outperforming the conductivity of ionic liquids, solid ceramic and polymer electrolytes so that it can still meet the requirement of battery applications at low/moderate C-rates.^{51–54}”

Besides, we have also included the rate performance of the 1 M-LPF-DMDOHD electrolyte in the supplementary information. As depicted in Supplementary Fig. 12, the Li//LiNi_{0.5}Mn_{1.5}O₄ cell operated with the 1 M-LPF-DMDOHD electrolyte exhibits poor electrochemical performance at large C-rates ($> 1 \text{ C}$), which can be attributed primarily to the low ionic conductivity of the 1 M-LPF-DMDOHD electrolyte, corresponding to what the reviewer mentioned. However, we also found that it was able to return to a discharge capacity close to that of the initial capacity when the C-rate reverted back to a small C-rate of 0.2 C , which demonstrated its good capacity recovery capability.

Given that electrolyte systems using DMDOHD as a single solvent exhibit low ionic conductivity and unsatisfactory rate performance, the practicality of employing DMDOHD as the single electrolyte solvent appears doubtful. To further clarify whether DMDOHD has any value for practical application, we explore its potential to enhance the electrochemical performance of LNMO-based full cells when employed as an electrolyte co-solvent under more rigorous and practical conditions. Here, we adopted the pouch cell configuration for our investigations. Regrettably, since we do not have the technical conditions to prepare lithium metal pouch cell, we opted for graphite//LNMO pouch cell as an alternative for our studies. The cathode was made of LNMO, PVDF (Kynar® HSV 1800), and carbon black (Super P) in a mass ratio of 97.5:1:1.5. The mass loading of LNMO is about 28 mg/cm^2 , providing an areal capacity of 4 mAh/cm^2 . The anode was made of graphite (GCP-80), PVDF (Kynar® HSV 1800), and carbon black (Super P) in a mass ratio of 90:5:5. The mass loading of graphite is around 11.5 mg/cm^2 , yielding an areal capacity of 4.3 mAh/cm^2 . The dimensions of the cathode and

anode are 53 mm * 67 mm and 55 mm * 69 mm, respectively. We employed microporous polyethylene (PE) (Celgard 2400) as the separator. The amount of electrolyte added into the pouch cell was 4.2 g/Ah. For assembly, we utilized five cathodes and six anodes to construct the pouch cell. The pouch cell's theoretical capacity is approximately 0.7 Ah, based on the cathode's capacity, and the N/P ratio stands at approximately 1.08.

As depicted in Figure 10d and Supplementary Fig. 34, the cycling stability of graphite//LNMO pouch cells exhibited significant improvement as the volume ratio of DMDOHD increased from 0% to 30% within the FEC/FEMC-based electrolyte system. After 300 cycles under the test condition of 0.5 C charging-1 C discharging, the discharge capacity can still be maintained at more than 94% of the initial capacity. Even when the volume ratio of DMDOHD reached 40%, the cycling stability remained superior to that of the system without DMDOHD, albeit with a reduction compared to the 30% volume ratio system. Additionally, we conducted rate capability tests on the graphite//LNMO pouch cell with a 30% volume ratio of DMDOHD in the FEC/FEMC-based electrolyte system, varying the discharge rate from 0.1 C to 2.0 C. The results, shown in Supplementary Fig. 35, reveal the outstanding rate capability of the graphite//LNMO pouch cell with this electrolyte composition. Specifically, the rate capability ($Q_{2C}/Q_{0.1C}$) reached 93%, and the discharge capacity at 2 C was approximately 0.589 Ah, indicating a high utilization rate of 84% at a high current rate. The above results fully demonstrate the practical application value of DMDOHD as an electrolyte co-solvent in Li-ion cells.

Supplementary Fig. 12 (a) Galvanostatic charge-discharge plots at different current rates and (b) rate performance of the Li//LiNi_{0.5}Mn_{1.5}O₄ cell cycled with 1 M-LPF-DMDOHD electrolyte.

Figure 10d. Cycling stability of the graphite//LNMO pouch cell in different electrolytes (0.5 C charge-1 C discharge) at 25 °C. A 0.5 C constant current and 4.85 V constant voltage (CC-CV) mode was adopted for all the charging sequences while a 1 C constant current (CC) mode was adopted for all the discharging sequences. FEC and PS represent 2,2,2-trifluoroethyl methyl carbonate and 1,3-propanesultone, respectively.

Supplementary Fig. 34 Selected voltage profiles of the graphite//LNMO pouch cells (a) without and (b) with a 30% volume ratio DMDOHD in the FEC/FEMC-based electrolyte system (0.5 C charging-1 C discharge) at 25 °C. A 0.5 C constant current and 4.85 V constant voltage (CC-CV) mode was adopted for all the charging sequences while a 1 C constant current (CC) mode was adopted for all the discharging sequences. FEC and PS represent 2,2,2-trifluoroethyl methyl carbonate and 1,3-propanesultone, respectively.

Supplementary Fig. 35 Rate capability of graphite//LNMO pouch cell with a 30% volume ratio DMDOHD in the FEC/FEMC-based electrolyte system at 25 °C. A 0.3 C constant current and 4.85 V constant voltage (CC-CV) mode was adopted for all the charging sequences. Voltage profiles are plotted with respect to the cell capacity. FEMC and PS represent 2,2,2-trifluoroethyl methyl carbonate and 1,3-propanesultone, respectively.

3. Then, Authors state: As shown in Supplementary Fig. 4, the 1 M-LPF-DMDOHD electrolyte shows significantly higher anodic stability compared to the reference carbonate-based electrolytes, as evidenced by the much lower anodic current, even when polarized up to 5.5 V (vs. Li/Li⁺).

Well in S4 - The data for EC/DMC and DMC look very strange. It would be nice if these results had been repeated or if a second anodic sweep had been included. Also, authors need to consider the incredible low ionic conductivity of the electrolyte and how it will influence the CV results. This cannot be ignored. Currents will be small due to poor transport properties, not only because of better anodic stability. The same considerations need to be made about the leakage currents later in the same paragraph.

Response: We thank the reviewer for the insightful comments and valuable suggestions, which we respond to below.

(1) The linear scan voltammetry (LSV) plots previously shown in Supplementary Fig. 4 (now replaced by Supplementary Fig. 5) were obtained in a two-electrode coin-cell system. This system utilized a stainless-steel disc as the working electrode, and a piece of lithium foil

...serving as both the counter electrode and quasi-reference electrode. However, it's important to note that in such a coin-cell setup, the solution impedance is relatively high, and the extent to which the lithium foil participates in the electrochemical reaction can impact the electrode potential, given its dual role as both the counter electrode and reference electrode. Due to these considerations, when measuring the electrochemical stability voltage window of the electrolyte in the two-electrode coin-cell system, it's generally advisable to use an exceptionally low scan rate, such as 0.1 mV/s. This is done to minimize potential inaccuracies caused by excessive polarization within the cell. However, conducting measurements at such a low scan rate can lead to LSV curve fluctuations during the process. These fluctuations may result from various factors, including instrument measurement precision, instability within the coin-cell's microenvironment, or external environmental changes such as temperature fluctuations or micro-vibrations. Consequently, the LSV curves displayed previously were smoothed using Origin software, which may account for their strange shape.

In fact, the three-electrode electrochemical cell system is more suitable for evaluating the electrochemical stability voltage window of the electrolyte system as compared to the two-electrode coin-cell setup, because of its lower impedance and the more stable electrode potential of the reference electrode (since it does not participate in the electrochemical reaction). Many studies have adopted this method for testing the electrochemical stability voltage window of electrolyte systems.^[1-5] Therefore, to enhance the reliability and repeatability of our data, we re-evaluated the three studied electrolyte systems using a three-electrode electrochemical cell setup. This setup featured a glassy carbon (GC) disk electrode (3 mm in diameter) as the working electrode, two lithium metal foils as the counter and reference electrodes, respectively, and 10 mL of 1 M-LPF-DMC, 1 M-LPF-EC/DMC, or 1 M-LPF-DMDOHD as the electrolyte. In the electrochemical stability voltage window measurement of the three studied electrolytes using the three-electrode electrochemical cell system, we opted for a scan rate of 1 mV/s. This choice can effectively mitigate concerns associated with inadequate mass transport properties within the electrolyte system, which will be discussed in detail below.

Following the reviewer's suggestions, we have also provided the second anodic scan curves for the three electrolytes. As depicted in Supplementary Fig. 5, during the first anodic scan, the onset oxidation potentials for 1 M-LPF-DMC, 1 M-LPF-EC/DMC, and 1 M-LPF-DMDOHD electrolytes were 4.3, 4.6, and 4.8 V, respectively. Additionally, the anodic currents

when polarized to 6.5 V were 1.7, 1.3, and 0.19 mA/cm², respectively. However, in the second anodic scan, all three electrolytes exhibited delayed onset oxidation potentials, measuring 4.8, 5.0, and 5.2 V, respectively. Furthermore, the anodic currents at polarization to 6.5 V decreased to 1.1, 0.6, and 0.15 mA/cm², respectively. This phenomenon suggests that the passivation film formed by the electrolyte decomposition on the GC electrode surface during the first anodic scan can partially inhibit further electrolyte decomposition. Given that the 1 M-LPF-DMDOHD electrolyte demonstrates higher onset oxidation potentials and lower anodic currents compared to the two carbonate-based reference electrolytes, it can be concluded that the 1 M-LPF-DMDOHD electrolyte offers better anodic stability.

Further, to explore whether the poor transport properties caused by the low ionic conductivity may have an effect on the LSV results, we conducted LSV experiments on the 1 M-LPF-DMDOHD electrolyte system at various scan rates, ranging from 1 to 100 mV/s. Before each measurement at different scan rates, we polished, cleaned and dried the GC electrodes to eliminate any influence from the passivation film that might have formed on the GC electrode surface due to electrolyte decomposition in previous experiments. As displayed in Supplementary Fig. 6, our findings reveal that the onset oxidation potentials of the first and second anodic scans of the 1 M-LPF-DMDOHD electrolyte system stabilized at around 4.8 and 5.2 V, respectively, while the anodic current gradually increased as the scan rate was gradually increased from 1 mV/s to 100 mV/s. This suggests that the LSV tests on the 1 M-LPF-DMDOHD electrolyte are not significantly affected by mass transport limitations. Moreover, the small anodic oxidation current of the 1 M-LPF-DMDOHD electrolyte can indeed be attributed to its better anodic stability. These conclusions are bolstered by the following analysis: as the scan rate progressively increased, the reaction rate at the electrode surface accelerated. If the mass transport properties of the 1 M-LPF-DMDOHD electrolyte were inadequate, it would have been increasingly challenging for mass transport to match the reaction rate. In such a scenario, it would have been unlikely to observe the phenomenon in which the anodic oxidation current gradually increased with the rising scan rate.

(2) For the leakage current test, if the small leakage currents observed in the 1 M-LPF-DMDOHD electrolyte system were caused by poor mass transport, it would be unlikely to see a significant increase in transient currents at each step to a higher potential. As illustrated in Figure R1, the occurrence of transient currents increasing from minimal levels to as high as 0.45 mA/cm² at 4.9 V to 5.0 V, 5.0 V to 5.1 V, and 5.1 V to 5.2 V suggests that the 1 M-LPF-

DMDOHD electrolyte system does not face mass transport limitations in the leakage current test. Otherwise, the leakage currents at 5.0, 5.1, and 5.2 V would always remain at the same minimal level as observed at 4.9 V. Additionally, the peak transient current is approximately 0.45 mA/cm², corresponding to a current density of around 45 mA/g (considering LNMO mass loading is approximately 10 mg/cm²), which equals a C-rate of C/3. As demonstrated in Fig. 6d, the Li//LNMO cell can achieve a specific capacity of about 131 mAh/g at a C-rate of C/3 in the 1 M-LPF-DMDOHD electrolyte system, and it maintains a specific capacity of >123 mAh/g even after 250 cycles. This indicates that mass transport is not an issue at this C-rate (i.e., C/3) when testing in the 1 M-LPF-DMDOHD electrolyte system. Hence, given that even the highest transient current of 0.45 mA/cm² (about C/3) would not lead to insufficient mass transport issues during the leakage current test, we can confidently conclude that the smaller leakage current observed in the 1 M-LPF-DMDOHD electrolyte system can indeed be attributed to its superior anodic stability.

The corresponding texts and results have been inserted in the suitable positions in our revised manuscript and Supporting Information.

Supplementary Fig. 5 (a) Linear scan voltammetry (LSV) plots of 1 M-LPF-DMC, 1 M-LPF-EC/DMC and 1 M-LPF-DMDOHD electrolytes from open-circuit voltage (OCV) to 6.5 V. The measurement was conducted by using a three-electrode electrochemical cell with a 3 mm

diameter glassy carbon (GC) disk as the working electrode and two lithium metal foils as the counter and reference electrodes, respectively. The scan rate is 1 mV s^{-1} . (b, c) Enlarged LSV plots of (b) the first scan and (c) the second scan of 1 M-LPF-DMC, 1 M-LPF-EC/DMC and 1 M-LPF-DMDOHD electrolytes shown in (a). (d–f) Enlarged LSV plots of (d) 1 M-LPF-DMC, (e) 1 M-LPF-EC/DMC and (f) 1 M-LPF-DMDOHD electrolytes shown in (a).

Supplementary Fig. 6 Linear scan voltammetry (LSV) plots of 1 M-LPF-DMDOHD electrolytes from open-circuit voltage (OCV) to 6.5 V at scan rates varying from 1 to 100 mV s^{-1} . The measurement was conducted by using a three-electrode electrochemical cell with a 3 mm diameter glassy carbon (GC) disk as the working electrode and two lithium metal foils as the counter and reference electrodes, respectively. The GC electrode was polished, cleaned, and dried before each measurement at different scan rates. This step aimed to minimize the impact of the passivation film formed on the GC surface by the electrolyte decomposition in the previous experiments.

Figure R1. Potentiostatic floating profiles of $\text{Li}||\text{LiNi}_{0.5}\text{Mn}_{1.5}\text{O}_4$ cell with 1 M-LPF-DMDOHD electrolyte maintained at 4.9, 5.0, 5.1 and 5.2 V for 8 hours at room temperature.

Figure 6d. *Cycling stability of the Li//LiNi_{0.5}Mn_{1.5}O₄ cell with 1 M-LPF-DMDOHD electrolyte cycled at rate of C/3.*

References

[1] Zheng, Q., et al. (2020). A cyclic phosphate-based battery electrolyte for high voltage and safe operation. *Nature Energy*, 5(4), 291–298.

[2] Xu, J., et al. (2022). Aqueous electrolyte design for super-stable 2.5 V LiMn₂O₄|| Li₄Ti₅O₁₂ pouch cells. *Nature Energy*, 7(2), 186–193.

[3] Wang, J., et al. (2016). Superconcentrated electrolytes for a high-voltage lithium-ion battery. *Nature Communications*, 7(1), 12032.

[4] Azcarate, I., et al. (2020). Assessing the oxidation behavior of EC: DMC based electrolyte on non-catalytically active surface. *Journal of The Electrochemical Society*, 167(8), 080530.

[5] Xu, K., Ding, S. P., & Jow, T. R. (1999). Toward reliable values of electrochemical stability limits for electrolytes. *Journal of The Electrochemical Society*, 146(11), 4172.

4. Authors state: Meanwhile, the DMDOHD-based electrolyte also exhibits a wide liquid range down to -70°C , which may have the potential for low temperature applications. Authors have no connection to reality. The conductivity will be so incredibly low that cells simply will not function.

Response: We thank the reviewer for this critical comment and we apologize for our less rigorous statement. We have deleted this statement.

5. Figure S21 is quite impressive and could be moved to the main text. One can clearly see in this Figure the large charge discharge polarization for the DMDOHD cells compared to the standard electrolyte cells. This is due to the poor conductivity of the DMDOHD. This data is collected at room temperature. If authors want to publish in a high impact journal, they MUST show test results at elevated temperature. It is easy to make things work at room temperature.

Figure S20 also shows the large polarization in the Li/Li cells due to poor ionic conductivity.

Response: We thank the reviewer for the insightful comments and valuable suggestions.

(1) Following the reviewer's suggestions, we have moved Figure S21 to the main text (now Figure 10a–c). We also agree with the reviewers that the low ionic conductivity of the electrolyte system with DMDOHD as the single/major solvent causes the large polarization problem in the LNMO-Li cell in Figure S21 and the Li/Li symmetric cell in Figure S20 (now Supplementary Fig. 30).

*As discussed in the response of comment 2, given that electrolyte systems using DMDOHD as a single solvent exhibit extremely low ionic conductivity and poor rate performance, the practicality of employing DMDOHD as the single electrolyte solvent appears doubtful. To further clarify whether DMDOHD has any value for practical application, we explore its potential to enhance the electrochemical performance of LNMO-based full cells when employed as an electrolyte co-solvent under more rigorous and practical conditions. Here, we adopted the pouch cell configuration for our investigations. Regrettably, since we do not have the conditions to prepare lithium metal pouch cell, we opted for graphite//LNMO pouch cell as an alternative for our studies. The cathode comprises LNMO, PVDF (Kynar® HSV 1800), and carbon black (Super P) in a mass ratio of 97.5:1:1.5. The mass loading of LNMO is about 28 mg/cm², providing an areal capacity of 4 mAh/cm². Meanwhile, the anode is composed of graphite (GCP-80), PVDF (Kynar® HSV 1800), and carbon black (Super P) in a mass ratio of 90:5:5. The mass loading of graphite is around 11.5 mg/cm², yielding an areal capacity of 4.3 mAh/cm². The dimensions of the cathode and anode are 53 mm * 67 mm and 55 mm * 69 mm, respectively. We employed microporous polyethylene (PE) (Celgard 2400) as the separator. The amount of electrolyte added into the pouch cell was 4.2 g/Ah. For assembly, we utilized five cathodes and six anodes to construct the pouch cell. The pouch cell's theoretical capacity is approximately 0.7 Ah, based on the cathode's capacity, and the N/P ratio stands at approximately 1.08.*

As depicted in Figure 10d and Supplementary Fig. 34, the cycling stability of graphite//LNMO pouch cells exhibited significant improvement as the volume ratio of DMDOHD increased from 0% to 30% within the FEC/FEMC-based electrolyte system. After 300 cycles under the test condition of 0.5 C charging-1 C discharging, the discharge capacity

can still be maintained at more than 94% of the initial capacity. Even when the volume ratio of DMDOHD reached 40%, the cycling stability remained superior to that of the system without DMDOHD, albeit with a reduction compared to the 30% volume ratio system. Additionally, we conducted rate capability tests on the graphite//LNMO pouch cell with a 30% volume ratio of DMDOHD in the FEC/FEMC-based electrolyte system, varying the discharge rate from 0.1 C to 2.0 C. The results, shown in Supplementary Fig. 35, reveal the outstanding rate capability of the graphite//LNMO pouch cell with this electrolyte composition. Specifically, the rate capability ($Q_{2C}/Q_{0.1C}$) reached 93%, and the discharge capacity at 2 C was approximately 0.589 Ah, indicating an impressive utilization rate of 84% at a high current rate. The above results fully demonstrate the practical application value of DMDOHD as an electrolyte co-solvent.

(2) Further, we also tested the high temperature electrochemical performance of the graphite//LNMO pouch cells with and without DMDOHD as an electrolyte co-solvent. As illustrated in Supplementary Figs. 36 and 37, the graphite//LNMO pouch cell employing a 30% volume ratio of DMDOHD within the FEC/FEMC-based electrolyte system demonstrates excellent cycling stability, with a capacity retention of ~86% over 300 cycles, and an average CE close to 100% at a C-rate of 1 C under 45 °C, which is superior to that observed in the cell without DMDOHD as an electrolyte co-solvent (<80% capacity retention after 240 cycles). This result once again demonstrates the practical application value of DMDOHD as an electrolyte co-solvent.

The corresponding texts and results have been inserted in the suitable positions in our revised manuscript and Supporting Information.

Figure 10d. Cycling stability of the graphite//LNMO pouch cell in different electrolytes (0.5 C charging-1 C discharging) at 25 °C. A 0.5 C constant current and 4.85 V constant voltage (CC-CV) mode was adopted for all the charging sequences while a 1 C constant current (CC) mode was adopted for all the discharging sequences. FEMC and PS represent 2,2,2-trifluoroethyl methyl carbonate and 1,3-propanesultone, respectively.

Supplementary Fig. 34 Selected voltage profiles of the graphite//LNMO pouch cells (a) without and (b) with a 30% volume ratio of DMDOHD in the FEC/FEMC-based electrolyte system (0.5 C charging-1 C discharging) at 25 °C. A 0.5 C constant current and 4.85 V constant voltage (CC-CV) mode was adopted for all the charging sequences while a 1 C constant current (CC) mode was adopted for all the discharging sequences. FEMC and PS represent 2,2,2-trifluoroethyl methyl carbonate and 1,3-propanesultone, respectively.

Supplementary Fig. 35 Rate capability of graphite//LNMO pouch cell with a 30% volume ratio of DMDOHD in the FEC/FEMC-based electrolyte system at 25 °C. A 0.3 C constant current and 4.85 V constant voltage (CC-CV) mode was adopted for all the charging sequences. Voltage profiles were plotted with cell capacity. FEMC and PS represent 2,2,2-trifluoroethyl methyl carbonate and 1,3-propanesultone, respectively.

Supplementary Fig. 36 Cycling stability of the graphite//LNMO pouch cells in different electrolytes (1 C charging-1 C discharging) at 45 °C. A 1 C constant current and 4.85 V constant voltage (CC-CV) mode was adopted for all the charging sequences while a 1 C constant current (CC) mode was adopted for all the discharging sequences. FEMC and PS represent 2,2,2-trifluoroethyl methyl carbonate and 1,3-propanesultone, respectively.

Supplementary Fig. 37 Selected voltage profiles of the graphite//LNMO pouch cell with a 30% volume ratio of DMDOHD in the FEC/FEMC-based electrolyte system (1 C charging-1 C discharging) at 45 °C. A 1 C constant current and 4.85 V constant voltage (CC-CV) mode was adopted for all the charging sequences while a 1 C constant current (CC) mode was adopted for all the discharging sequences. FEMC and PS represent 2,2,2-trifluoroethyl methyl carbonate and 1,3-propanesultone, respectively.

6. Overall, this paper has value, but it is oversold in order to be published in Nature Comm. Authors should make the paper more realistic and more truthful, maybe then it could be suitable.

Response: We thank the reviewer for the critical and valuable comments on our manuscript. We have improved our work based on the above comments proposed by the reviewer. We hope this revision has well addressed the reviewer's concern and the manuscript may now be considered for publication in Nature Communications.

Reviewer #2:

1. Comments on the DFT calculations:

Model chemistry is PCM(???) / B3LYP/6-311+G(d,p), what is the implicit solvent model used here? The default is water which I do not think is appropriate - at least by itself. What is the justification for using B3LYP over functionals like M05-2X or wB97XD or even the G4MP2 composite method?

Since permittivity of the electrolyte is not known, it would be reasonable to consider a few options like diethylether, acetone, acetonitrile/methanol, and water. How does the potential vary with permittivity?

Response: We thank the reviewer for the insightful questions and constructive comments, which we respond to below.

(1) Model chemistry is PCM(???) / B3LYP/6-311+G(d,p), what is the implicit solvent model used here? The default is water which I do not think is appropriate - at least by itself. Since permittivity of the electrolyte is not known, it would be reasonable to consider a few options like diethylether, acetone, acetonitrile/methanol, and water. How does the potential vary with permittivity?

Previously, we employed the IEFPCM/B3LYP/6-311+G(d,p) model chemistry to calculate the LUMO and HOMO energy levels of the studied carbonate molecules. However, to provide a more comprehensive comparison of the LUMO and HOMO energy level trends for these, we have expanded our approach to include several additional model chemistries, such as SMD/B3LYP/6-311+G(d,p), SMD/M05-2X/6-311+G(d,p), SMD/wB97XD/6-311+G(d,p), and SMD/G4MP2/6-311+G(d,p), which will be discussed in the response of next comment.

The selection of the implicit solvent model was determined based on the dielectric constants of the studied carbonate molecules. As shown in Table R1, for linear carbonate molecules such as DMC ($\epsilon=3.12$), EMC ($\epsilon=2.93$), DEC ($\epsilon=2.82$), and DMDOHD ($\epsilon=2.9$, calculated value), their dielectric constants exhibit minimal variation and are closely aligned (around 3), thus we use ethyl acetate, which has a similar molecule structure and a close dielectric constant ($\epsilon=5.99$), as the implicit solvent model. While the cyclic carbonate molecule EC possesses a significantly higher dielectric constant of 89.6, prompting the adoption of water

as an implicit solvent model, given its similarly large dielectric constant of 78.5.

Although the dielectric constant of DMDOHD is not readily available in the database, we were able to calculate it, and our result indicates a dielectric constant of 2.9. This value is reasonable considering that DMDOHD shares a linear carbonate structure similar to DMC, DEC, and EMC, which also exhibit dielectric constants in proximity to that of DMDOHD. Consequently, in line with the approach used for DMC, DEC, and EMC, we selected ethyl acetate, which has a similar molecule structure and a close dielectric constant ($\epsilon=5.99$), as the implicit solvent model for DMDOHD to calculate its HOMO and LUMO energy levels.

Additionally, we also used EC with known dielectric constant ($\epsilon=89.6$) as the research object to explore the effect of different implicit solvent models with varying dielectric constants, such as diethylether ($\epsilon=4.22$), acetone ($\epsilon=20.5$), methanol ($\epsilon=33.7$), acetonitrile ($\epsilon=37.5$), and water ($\epsilon=78.5$), on the results of HOMO and LUMO energy level calculations. As shown in Table R2, we found that the implicit solvent models with different dielectric constants had minimal impact on the LUMO calculation results of EC. However, they did significantly affect the HOMO calculation results, with implicit solvents possessing higher dielectric constants (except for methanol) leading to more negative calculated HOMO results for EC. This underscores the importance of selecting an appropriate implicit solvent model. In general, it is advisable to choose solvents with similar structures and close dielectric constants as implicit solvent models when calculating LUMO and HOMO energy levels. Interestingly, it is worth mentioning that the HOMO energy levels of the EC calculated using the five different implicit solvent models mentioned above are all lower than those of the other studied carbonate molecules (DEC: -8.02 eV, EMC: -8.08 eV, DMC: -8.14 eV, and DMDOHD: -8.25 eV) calculated by the same model chemistry (SMD/B3LYP/6-311+G(d,p)), which does not affect the results of the comparison of the energy level trends.

In fact, our original intention of calculating the HOMO and LUMO energy levels of carbonate solvent molecules was to make a rough comparison. We wanted to assess the trend of DMDOHD's oxidation and reduction stability/priority concerning other carbonate solvent molecules, particularly EC and DMC. Based on the above analysis, we believe that our prior calculations of HOMO and LUMO should not pose much problem in evaluating these stability/priority trends between DMDOHD and carbonate solvent molecules like EC and DMC.

Table R1. Dielectric constants of linear and cyclic carbonate molecules.

Carbonate molecules	Dielectric constants (ϵ)
EC	89.6
DMC	3.12
EMC	2.93
DEC	2.82
DMDOHD	2.9 (calculated value)

Table R2. Impact of implicit solvent models on EC's energy levels by using SMD/B3LYP/6-311+G(d,p) as model chemistry.

Implicit solvent models (ϵ)	HOMO (eV)	LUMO (eV)
Water (78.5)	-8.65	0.09
Acetonitrile (37.5)	-8.41	0.09
Methanol (33.7)	-8.66	0.09
Acetone (20.5)	-8.38	0.09
Diethylether (4.22)	-8.36	0.08

(2) What is the justification for using B3LYP over functionals like M05-2X or wB97XD or even the G4MP2 composite method?

To the best of our knowledge, the B3LYP functional is one of the most classic methods for calculating molecular orbitals. Therefore, we chose it previously to calculate the LUMO and HOMO energy levels of the studied carbonate molecules. However, it's worth noting that different functional methods have their unique strengths and weaknesses. The reviewer's insightful question prompted us to enhance our analysis by employing various functional methods to compare the trends in LUMO and HOMO energy levels for the studied carbonate

molecules more comprehensively.

In Supplementary Table 5, we provide a summary of the LUMO and HOMO energy levels calculated for DEC, EMC, DMC, EC, and DMDOHD using different functional methods. Below are the orders of LUMO and HOMO energy levels for the five carbonate molecules calculated with various functional methods:

LUMO energy order:

IEFPCM/B3LYP: DEC>EMC>DMC>DMDOHD>EC

SMD/B3LYP: DEC>EMC>DMC>DMDOHD>EC

SMD/M05-2X: DEC>EMC>DMC>EC>DMDOHD

SMD/wB97XD: DEC>EMC>DMC=DMDOHD>EC

SMD/G4MP2: EC>DEC>EMC>DMC>DMDOHD

HOMO energy order:

IEFPCM/B3LYP: DEC>EMC>DMC>DMDOHD>EC

SMD/B3LYP: DEC>EMC>DMC>DMDOHD>EC

SMD/M05-2X: DEC>EMC>DMC>DMDOHD>EC

SMD/wB97XD: DEC>EMC>DMC=DMDOHD>EC

SMD/G4MP2: DEC>EMC>DMDOHD>EC>DMC

It's important to note that the LUMO and HOMO energy levels calculated by SMD/G4MP2 appear to contradict the conventional belief that EC should be reduced before DMC. According to the conventional belief, the LUMO energy level of EC should be lower than that of DMC. Additionally, it has been demonstrated that the oxidation potential of high-purity EC should be higher than that of high-purity DMC (as seen in Figure R2),^[1] suggesting that the HOMO energy level of EC should also be lower than that of DMC. However, the results attained from SMD/G4MP2 indicate that both the LUMO and HOMO energy levels of EC is higher than that of DMC.

Except for SMD/G4MP2, the energy level trends from other four functional methods largely support our previous conclusions — “(1) For the SEI film forming ability, our theoretical calculation results show that the energy of the lowest unoccupied molecular orbital

(LUMO) in DMDOHD is lower than that of conventional linear carbonates, (e.g., diethyl carbonate - DMC, ethyl methyl carbonate – EMC, or diethyl carbonate - DEC), and close to that of ethylene carbonate - EC (Supplementary Table 5). This indicates that DMDOHD is preferentially reduced to form the SEI under cathodic polarization with an essential role to play in the modulation of the SEI composition and interfacial chemistry, like EC. And (2) For the anodic stability, DMDOHD also exhibits a lower than DMC, EMC and DEC highest occupied molecular orbital (HOMO) energy level (Supplementary Table 5), implying improved anodic stability of the DMDOHD solvent.”

The corresponding texts and results have been inserted in the suitable positions in our revised manuscript and Supporting Information.

[REDACTED]

Figure R2. *Relation between the oxidation potential and purity of several solvents.^[1] (Fig. 19.2 on Page 356 of Ref. [1])*

Supplementary Table 5. Chemical structure and molecular orbital energy of conventional (DEC, EMC, DMC, and EC) as well as of DMDOHD carbonate organic solvents used in battery electrolytes.

Molecule	Structure	Model chemistry	LUMO (eV)	HOMO (eV)
DEC		IEFPCM/B3LYP/6-311+G(d,p)	0.20	-8.25
		SMD/B3LYP/6-311+G(d,p)	0.30	-8.02
		SMD/M05-2X/6-311+G(d,p)	1.46	-10.14
		SMD/wB97XD/6-311+G(d,p)	2.13	-10.28
		SMD/G4MP2/6-311+G(d,p)	1.32	-12.43
EMC		IEFPCM/B3LYP/6-311+G(d,p)	0.18	-8.31
		SMD/B3LYP/6-311+G(d,p)	0.27	-8.08
		SMD/M05-2X/6-311+G(d,p)	1.44	-10.20
		SMD/wB97XD/6-311+G(d,p)	2.09	-10.34
		SMD/G4MP2/6-311+G(d,p)	1.29	-12.56
DMC		IEFPCM/B3LYP/6-311+G(d,p)	0.15	-8.37
		SMD/B3LYP/6-311+G(d,p)	0.24	-8.14
		SMD/M05-2X/6-311+G(d,p)	1.43	-10.25
		SMD/wB97XD/6-311+G(d,p)	1.96	-10.51
		SMD/G4MP2/6-311+G(d,p)	1.28	-12.80
EC		IEFPCM/B3LYP/6-311+G(d,p)	-0.03	-8.63
		SMD/B3LYP/6-311+G(d,p)	0.09	-8.65
		SMD/M05-2X/6-311+G(d,p)	1.32	-10.59
		SMD/wB97XD/6-311+G(d,p)	1.92	-10.83
		SMD/G4MP2/6-311+G(d,p)	1.47	-12.79
DMDOHD		IEFPCM/B3LYP/6-311+G(d,p)	0.06	-8.47
		SMD/B3LYP/6-311+G(d,p)	0.15	-8.25
		SMD/M05-2X/6-311+G(d,p)	1.31	-10.37
		SMD/wB97XD/6-311+G(d,p)	1.96	-10.51

References

[1] Yoshitake, H. (2009). *Functional electrolytes specially designed for lithium-ion batteries. Lithium-Ion Batteries: Science and Technologies*, 343–366.

2. HOMO/LUMO is not necessarily reflective of the actual redox stability of the materials at battery-relevant potentials due to deformation of the molecule in the reduced or oxidized state (or bond breaking). A more insightful approach the authors should consider is the explicit calculation of reduction and oxidation potentials using the Nernst equation. Because the DMDOHD molecule is assumed to contribute to SEI, the reduction potential should be computed with Li⁺ present as well.

The authors should conduct conformer analysis for solvents with rotatable bonds, it looks like the authors have shown only the linear/extended conformer of DMDOHD in Supp Table 5 which is not the lowest energy conformer I get from a simple search in RDKit. Since this structure is used as an input to the BOMD simulations, used to justify the increase in Li-PF₆ ion pairing, and to justify the participation of DMDOHD in the SEI, this needs to be addressed. Some of the low energy conformers look like they could potentially coordinate Li⁺ with both C=O groups (cis-like conformers). This may or may not change the extent of ion pairing relative to that claimed by the authors but will certainly influence the reduction potential of the solvent and its subsequent participation in the SEI. As an example, the coordinates of the lowest energy conformer + Li from a not-exhaustive set of gas phase calculations with M05-2X/6-31+G(d,p) are below. The binding free energy between Li⁺ and the solvent here is ~25 kcal/mol lower than the linear/extended conformer.

In case of formatting issues, given as [atom symbol, X, Y, Z] like Xmol/XYZ format.

O 0.4294470000 -2.7958520000 -1.3194130000

O -0.2846170000 0.0457890000 -3.1548250000

O 0.5156800000 -1.8928660000 0.7619330000
O 1.1131100000 -0.0546540000 -1.4853170000
O -0.9009860000 -3.4615450000 0.3613350000
O -0.2266930000 1.7237540000 -1.6348830000
C 1.4462310000 0.1947660000 -0.0936820000
C 1.7309310000 -1.1576270000 0.5151430000
C 0.0430290000 -2.7255760000 -0.1509620000
C -1.5357560000 -4.4237040000 -0.5188570000
C 0.1468170000 0.5904540000 -2.1469910000
C -1.2811900000 2.4362610000 -2.3552230000
Li 0.5842880000 -1.6733040000 -2.7804090000
H 0.6130260000 0.6936890000 0.3964860000
H 2.3950220000 -1.7443580000 -0.1186840000
H 2.3342570000 0.8240650000 -0.0680220000
H -0.7854420000 -5.1037870000 -0.9130400000
H -2.0450440000 -3.8990160000 -1.3237660000
H 2.1765110000 -1.0295600000 1.4989550000
H -2.2448990000 -4.9454110000 0.1127390000
H -0.9297140000 2.6597850000 -3.3581930000
H -2.1708370000 1.8137900000 -2.3847020000
H -1.4431610000 3.3385170000 -1.7783430000

Response: We totally agree with the reviewer's viewpoints and thank the reviewer for his/her helpful suggestions. We have conducted conformer analysis for solvents with rotatable bonds and adopted the coordinates provided by the reviewer and used the Nernst equation to explicitly calculate the reduction potential of DMDOHD with Li⁺ present as well. The relevant

calculations refer to the methodology of previously published work as follows:

Electrochemical stability of the complex M is defined using the thermodynamic energy cycles relative to the Li/Li^+ scale as given by Eqs. (1) and (2).^[2-4]

$$E_{\text{ox}}(M) = [\Delta G_i + \Delta G_s^0(M^+) - \Delta G_s^0(M)]/F - 1.4 \quad (1)$$

$$E_{\text{red}}(M) = -[\Delta G_a + \Delta G_s^0(M^-) - \Delta G_s^0(M)]/F - 1.4 \quad (2)$$

where ΔG_i is the ionization free energy and ΔG_a is the electron attachment free energy in gas-phase at 298.15 K; $\Delta G_s(M^+)$, $\Delta G_s(M^-)$ and $\Delta G_s(M)$ are the free energies of solvation of the oxidized, reduced and initial complexes respectively; and F is the Faraday constant. A shift factor of 1.4 accounts for the difference between the absolute potential scale and Li/Li^+ . The shift factor depends on the nature of solvent, salt and concentration and might vary by 0.1–0.3 V because the variation of the lithium free energy of solvation in various solvents.^[2,4,5] For our calculations, B3LYP/6-311+G(d, p) or M05-2X/6-31+G(d,p) is used for structure optimization, while M05-2X/6-31G(d) is employed for solvation energy calculations with SMD (acetone, $\epsilon=20.5$).

As illustrated in Supplementary Fig. 2, our calculations reveal that the reduction potential of $\text{Li}^+(\text{EC})$ stands at 0.43 or 0.58 V vs. Li/Li^+ , as determined through B3LYP/6-311+G(d,p) and M05-2X/6-31+G(d,p) methods, respectively. These findings closely align with prior results obtained by Oleg Borodin and his coworkers,^[6-8] demonstrating the reliability of our computational methodology. As for the DMDOHD system, our calculations indicate that the lowest energy $\text{Li}^+(\text{DMDOHD})$ conformer exhibits a reduction potential of 0.3 V vs. Li/Li^+ (computed via B3LYP/6-311+G(d,p)) or 0.23 V vs. Li/Li^+ (calculated using M05-2X/6-31+G(d,p)). Although these values are slightly lower than that of $\text{Li}^+(\text{EC})$, they are still higher than the lithium electroplating potential (theoretical value: 0 V vs. Li/Li^+), implying that DMDOHD has the ability to participate in SEI formation.

Supplementary Fig. 2 Optimized geometry and reduction potential (vs. Li/Li^+) of (a, c) DMDOHD/Li^+ and (b, d) EC/Li^+ complexes attained from (a, b) B3LYP/6-311+G(d, p) and (c, d) M05-2X/6-31+G(d, p) for structure optimization and M05-2X/6-31G(d) for solvation energy calculation with SMD (acetone). Atom colors: Li, purple; C, dark grey; O, red; H, white.

References

- [2] Borodin, O., Behl, W., & Jow, T. R. (2013). Oxidative stability and initial decomposition reactions of carbonate, sulfone, and alkyl phosphate-based electrolytes. *The Journal of Physical Chemistry C*, 117(17), 8661–8682.
- [3] Fry, A. J. (2017). Computational applications in organic electrochemistry. *Current Opinion in Electrochemistry*, 2(1), 67–75.
- [4] Borodin, O. (2019). Challenges with prediction of battery electrolyte electrochemical stability window and guiding the electrode–electrolyte stabilization. *Current Opinion in Electrochemistry*, 13, 86–93.
- [5] Borodin, O. (2014). Molecular modeling of electrolytes. In *Electrolytes for Lithium and Lithium-Ion Batteries* (pp. 371–401). New York, NY: Springer New York.

[6] Borodin, O., et al. (2015). Challenges with quantum chemistry-based screening of electrochemical stability of lithium battery electrolytes. *ECS Transactions*, 69(1), 113.

[7] Borodin, O., et al. (2015). Towards high throughput screening of electrochemical stability of battery electrolytes. *Nanotechnology*, 26(35), 354003.

[8] Delp, S. A., Borodin, O., Olguin, M., Eisner, C. G., Allen, J. L., & Jow, T. R. (2016). Importance of reduction and oxidation stability of high voltage electrolytes and additives. *Electrochimica Acta*, 209, 498–510.

3. Comments on the Born-Oppenheimer MD simulation:

Firstly, there is critical information to reproduce these simulations missing from the methods section.

a.) What version of the VASP code used?

b.) If different from the version of the VASP code, give the version of the pseudopotentials used, e.g., 5.2 or 5.4, etc.

c.) What variants of the pseudopotentials were used? `_sv` for Li? etc.

d.) What is the functional used here? Is dispersion included, if so which model?

e.) What is the cutoff used? 400 eV? 520 eV?

f.) Were simulations performed using the Gamma point only? Some other kpoint scheme?

g.) What mass of hydrogen is used here? 2 fs timestep isn't appropriate for BOMD unless you only have heavy atoms (>He mass), hydrogen masses should be 0.5 fs or less, deuterium and tritium masses are safer for 1 fs timestep. You can accumulate significant errors in the energy and forces using too long a timestep.

h.) What procedure was used to prepare BOMD simulations after randomly packing the box and before the 60 ps run? Do you equilibrate the structure with a force field based simulation (if so, give details), only perform DFT relaxation (if so, give details), or just run without any further processing (if so, mention this)?

Response: We thank the reviewer for the insightful questions and helpful comments. In the

revised manuscript, we conducted classical molecular dynamics simulations using the GROMACS 2018 program to investigate the solvation structure of the studied electrolyte system, which includes many LiPF₆ pairs. Additionally, we also provided point-by-point answers to the questions raised by the reviewer upon the previous calculations, as described below.

a.) What version of the VASP code used?

VASP 5.4 was used in our calculations.

b.) If different from the version of the VASP code, give the version of the pseudopotentials used, e.g., 5.2 or 5.4, etc.

The version of the pseudopotentials is 2003 version.

*c.) What variants of the pseudopotentials were used? *_sv* for Li? etc.*

Standard pseudopotentials were used for all the elements.

d.) What is the functional used here? Is dispersion included, if so which model?

The Perdew-Burke-Ernzerhof generalized gradient approximation (GGA-PBE) was utilized to represent the exchange-correlation functional.

e.) What is the cutoff used? 400 eV? 520 eV?

The energy cutoff is 400 eV.

f.) Were simulations performed using the Gamma point only? Some other kpoint scheme?

A Monkhorst-Pack k-point mesh grid scheme with a 2×2×2 grid was used.

g.) What mass of hydrogen is used here? 2 fs timestep isn't appropriate for BOMD unless you only have heavy atoms (>He mass), hydrogen masses should be 0.5 fs or less, deuterium

and tritium masses are safer for 1 fs timestep. You can accumulate significant errors in the energy and forces using too long a timestep.

The mass of hydrogen used here is the standard hydrogen mass. According to the reviewer's suggestions, we recalculated the solvation structures using AIMD simulations with a time step of 0.5 fs. In order to obtain relatively reasonable solvation structures, the systems were equilibrated for at least 50 ps. The computational details and corresponding results are shown below.

AIMD Simulation Details: Ab initio Molecular Dynamics (AIMD) simulations were performed using the Vienna Ab initio Simulation Package (VASP 5.4) based on the pseudopotential plane-wave approach. The Perdew-Burke-Ernzerhof generalized gradient approximation (GGA-PBE) was utilized to represent the exchange-correlation functional, with a cutoff energy of 400 eV. The initial configurations of LiPF₆, EC, DMC, and DMDOHC were optimized using Gaussian09 program with B3LYP/6-311+G(d,p). The LiPF₆ salt/solvent mixtures were prepared by randomly placing the molecules in the simulation box based on their experimental densities and molar ratios. A cubic-shaped simulation box of length 1.186 nm was used for all dimensions. Then AIMD simulations were performed at 300 K using the NVT ensemble with a time step of 0.5 fs. Temperature oscillations were controlled using a Nose thermostat with a Nose-mass parameter of 1.0. A Monkhorst-Pack k-point mesh grid scheme with a 2×2×2 grid was used. The systems were equilibrated for at least 50 ps before the production run of 10 ps. The VESTA program was used to sample the most probable solvation shells from the simulation trajectory.

As depicted in Figures R3 and R4, the recalculated results from Ab initio molecular dynamics (AIMD) simulations align well with our previous calculation results. Specifically, in the 1 M-LPF-EC/DMC electrolyte system, Li⁺ predominantly coordinates with four solvent molecules in the first solvation shell (with a coordination number (CN) of Li⁺ with EC/DMC at ≈4, and CN of Li⁺ with PF₆⁻ at ≈0), resulting in a solvent-separated ion pair. In contrast, within the 1 M-LPF-DMDOHD electrolyte system, Li⁺ primarily coordinates with three solvent molecules and one anion in the first solvation shell (with a CN of Li⁺ with DMDOHD at ≈3, and CN of Li⁺ with PF₆⁻ at ≈1), forming a contact ion pair.

However, as pointed out by the reviewer in the subsequent comment, BOMD on these timescales (60-70 ps) and from a single trajectory for a single ion pair cannot provide a strong

statistical justification for the average coordination environment around Li in the electrolyte – especially when attempting to correlate the formation of contact ion pairs and aggregates to LiF generation. Consequently, we have also conducted simulations to explore the solvation structures of the studied electrolyte systems containing 20 LiPF₆ pairs using classic molecular dynamics simulations with the GROMACS 2018 program. The related results will be discussed in detail in our response to the next comment.

Figure R3. Snapshots of the molecular dynamics simulation box for (a) 1 M-LPF-EC/DMC and (b) 1 M-LPF-DMDOHD electrolytes. Atom colors: Li, green; C, dark grey; O, red; P, purple; F, light blue. The cubic boxes represent the periodic boundaries of the supercells used in the molecular dynamic simulations.

Figure R4. Radial distribution functions ($g(r)$, solid lines) and coordination numbers (CN, dashed lines) of Li^+ with (a) solvent (Li-O) and (b) anion (Li-F) in 1 M-LPF-EC/DMC and 1 M-LPF-DMDOHD electrolyte formulations.

h.) What procedure was used to prepare BOMD simulations after randomly packing the box and before the 60 ps run? Do you equilibrate the structure with a force field based simulation (if so, give details), only perform DFT relaxation (if so, give details), or just run without any further processing (if so, mention this)?

We performed DFT relaxation to equilibrate the structure after randomly packing the box and before the 60 ps run. The DFT relaxation was carried out by using Perdew-Burke-Ernzerhof (PBE) gradient-corrected exchange-corrected functional with the projector augmented plane wave (PAW) method as implemented in the Vienna ab-initio simulation package (VASP 5.4). The energy cutoff was set to 400 eV. A Monkhorst-Pack k -point mesh grid scheme with a $2 \times 2 \times 2$ grid was used. During structural optimization, all atoms were fully relaxed until the residual forces were less than 0.03 eV/\AA .

4. The authors can strengthen their claims about differences in ion pairing between the base carbonate electrolyte and the designed one using Raman spectroscopy (740 cm^{-1} for solvent

separated ion pairs – 750 cm^{-1} for contact ion pairs) or Li/F NMR. Alternatively, the authors could perform a simulation with many LiPF₆ pairs using an appropriate force field for 10s of nanoseconds. BOMD on these timescales (60-70 ps) and from a single trajectory for a single ion pair cannot provide a strong statistical justification for the average coordination environment around Li in the electrolyte – especially when attempting to correlate the formation of contact ion pairs and aggregates to LiF generation.

Response: We totally agree with the reviewer's viewpoints and thank the reviewer for his/her helpful suggestions, which we respond to below.

(1) According to reviewer's suggestions, we conducted the additional analysis of the differences in ion pairing between the base carbonate electrolyte and the designed one using Raman spectroscopy and ⁷Li NMR. As shown in Supplementary Fig. 23, within the 1 M-LPF-EC/DMC electrolyte system, the Raman band corresponding to the PF₅⁻ anion appears at 741 cm^{-1} , which suggests that the lithium-ion solvation structure predominantly comprises solvent-separated ion pairs (SSIPs), aligning with prior research findings.^[6-9] In contrast, the 1 M-LPF-DMDOHD electrolyte system exhibits a notable blue shift in the spectral position of the PF₅⁻ anion Raman band, specifically at around 745 cm^{-1} , implying that the lithium-ion solvation structure in this system may involve a combination of SSIPs, contact ion pairs (CIPs), and ion aggregates (AGGs).^[9-11] The formation of CIPs and AGGs within the 1 M-LPF-DMDOHD electrolyte system will intensify the interaction between Li⁺ and PF₅⁻, resulting in an increase in electron cloud density around Li⁺ (known as the shielding effect), which is further supported by the upfield chemical shift observed in the ⁷Li NMR spectrum in Supplementary Fig. 24 when comparing the 1 M-LPF-DMDOHD electrolyte system to the 1 M-LPF-EC/DMC electrolyte system. These results are consistent with the conclusions of our initial manuscript version.

Supplementary Fig. 23 Raman spectra of 1 M-LPF-EC/DMC and 1 M-LPF-DMDOHD electrolytes.

Supplementary Fig. 24 ${}^7\text{Li}$ NMR spectra of 1 M-LPF-EC/DMC and 1 M-LPF-DMDOHD electrolytes.

References

[6] Kim, Yong Min, et al. (2023). Flattening of Lithium Plating in Carbonate Electrolytes Enabled by All-In-One Separator. *Small* 19(28): 2301754.

[7] Woo, Sang-Gil, et al. (2021). High transference number enabled by sulfated zirconia

superacid for lithium metal batteries with carbonate electrolytes. Energy & Environmental Science, 14(3), 1420–1428.

[8] Miele, Ermanno, et al. (2022). *Hollow-core optical fibre sensors for operando Raman spectroscopy investigation of Li-ion battery liquid electrolytes. Nature Communications, 13(1), 1651.*

[9] Aoki, Yasuhito, et al. (2022). *Predictive Characterization of SEI Formed on Graphite Negative Electrodes for Efficiently Designing Effective Electrolyte Solutions. ACS Applied Energy Materials, 5(1), 1085–1094.*

[10] Wang, Jianhui, et al. (2016). *Superconcentrated electrolytes for a high-voltage lithium-ion battery. Nature Communications, 7(1), 12032.*

[11] Chaurasia, S. K., Singh, R. K., & Chandra, S. (2011). *Ion–polymer and ion–ion interaction in PEO-based polymer electrolytes having complexing salt LiClO₄ and/or ionic liquid, [BMIM][PF₆]. Journal of Raman Spectroscopy, 42(12), 2168–2172.*

(2) *We also conducted simulations to explore the solvation structures of the studied electrolyte systems containing 20 LiPF₆ pairs using classic molecular dynamics simulations with the GROMACS 2018 program. The computational details and corresponding results are detailed below.*

Classic molecular dynamics (MD) simulations: The ground state molecular and ions geometries were optimized using the DFT method at the B3LYP/6-311G+(d, p) level. All DFT calculations were performed using the Gaussian 09 software package. The MD simulations were conducted using the GROMACS 2018 program.^[12] The molecules and ions were described using the optimized potentials for liquid simulations all-atom (OPLS-AA) force field.^[13] The partial charges on the atoms of the solvent were computed by fitting the molecular RESP at the atomic centers using the B3LYP/aug-cc-pVDZ basis set.^[14] The simulation boxes were cubic with a side length of approximately 4 nm and contained 1 M LiPF₆ solvated in EC/DMC and DMDOHD solvents. During the simulations, the temperature was maintained at 300 K using a Nosé-Hoover thermostat with a relaxation time of 0.2 ps, and the pressure was

controlled at 1 bar using a Parrinello-Rahman barostat with a relaxation time of 2.0 ps. For a duration of 40 ns, and the last 20 ns of the simulations were utilized for analysis.

As illustrated in Figures 7d, e and Supplementary Figs. 21 and 22, the newly calculated results from classic molecular dynamics (MD) simulations also align with our previous results. Specifically, in the 1 M-LPF-EC/DMC electrolyte system, all 20 Li ions coordinate with four solvent molecules in the first solvation shell (with an average coordination number (CN) of Li^+ with EC/DMC at ≈ 4 , and an average CN of Li^+ with PF_6^- at ≈ 0), resulting in a solvent-separated ion pair. In contrast, within the first solvation shell of the 1 M-LPF-DMDOHD electrolyte system, 30% of Li^+ coordinates with four solvent molecules to form solvent-separated ion pairs (SSIPs), 50% of Li^+ forms contact ion pairs (CIPs) with the solvent molecules and PF_6^- anions, while 20% of Li^+ participates in the formation of ion aggregates (AGGs), which ultimately results in an average CN value of Li^+ with DMDOHD and PF_6^- of ≈ 2.9 and ≈ 1.1 , respectively. These findings strongly align with the outcomes of our Raman and ^7Li NMR analyses. Additionally, Figure R5 illustrates two representative Li^+ solvation structures, SSIP (Figure R5a) and CIP (Figure R5b), within the 1M-LPF-DMDOHD electrolyte system. The presence of these two structures corroborates the reviewer's viewpoint that DMDOHD could potentially form solvation structures with lithium ions in low-energy distorted conformers, where Li^+ is coordinated by two C=O groups (cis-like conformers).

The corresponding texts and results have been inserted in the suitable positions in our revised manuscript and Supporting Information.

Figure 7. Snapshots of the molecular dynamics simulation box and the simplified 20 Li^+ -coordination environment structures for (d) 1 M-LPF-EC/DMC and (e) 1 M-LPF-DMDOHD electrolytes. Atom colors: Li, green; C, dark grey; O, red; P, purple; F, light blue. The cubic boxes represent the periodic boundaries of the supercells used in the molecular dynamic simulations.

Supplementary Fig. 21 Radial distribution functions ($g(r)$, solid lines) and coordination numbers (CN, dashed lines) of Li^+ with (a) solvent (Li-O) and (b) anion (Li-F) in 1 M-LPF-EC/DMC and 1 M-LPF-DMDOHD electrolyte formulations.

Supplementary Fig. 22 Comparison of radial distribution functions ($g(r)$, solid lines) and coordination numbers (CN, dashed lines) of Li^+ with DMC and EC in 1 M-LPF-EC/DMC electrolyte formulation. Within the 1 M-LPF-EC/DMC electrolyte system, the average CN values of Li^+ with EC and DMC are ≈ 2.2 and ≈ 1.8 , respectively.

Figure R5. The selected Li^+ -coordination environment structures for 1 M-LPF-DMDOHD electrolyte system: (a) solvent-separated ion pair (SSIP) and (b) contact-ion pair (CIP). Atom colors: Li, green; C, dark grey; O, red; P, purple; F, light blue.

References

- [12] Abraham, M. J., Murtola, T., Schulz, R., Páll, S., Smith, J. C., Hess, B., & Lindahl, E. (2015). GROMACS: High performance molecular simulations through multi-level parallelism from laptops to supercomputers. *SoftwareX*, 1, 19–25.
- [13] Kaminski, G. A., Friesner, R. A., Tirado-Rives, J., & Jorgensen, W. L. (2001). Evaluation and reparametrization of the OPLS-AA force field for proteins via comparison with accurate quantum chemical calculations on peptides. *The Journal of Physical Chemistry B*, 105(28), 6474–6487.
- [14] Sambasivarao, S. V., & Acevedo, O. (2009). Development of OPLS-AA force field parameters for 68 unique ionic liquids. *Journal of chemical theory and computation*, 5(4), 1038–1050.

5. General comments about novelty:

The single salt, single solvent concept is great and appears to match or improve reversibility of LNMO|Li full cells among near peers from the last 5-6 years. I appreciate the authors' pursuit of practical electrolyte design, though unfortunately the authors have inherited some of the bad

practices common to battery research which I think detract from the novelty and impact of this work. Mass loading of LNMO is 10 mg/cm² in full cells, while practical mass loading is closer to ~25 mg/cm². I don't know the lithium foil thickness in the full cells. However, the Li|Cu cells use 100-200 micron thick Li reservoirs with a very low per cycle depth of discharge. If the authors saw comparable performance under more rigorous and practical conditions, this would be a higher impact paper.

Response: We thank the reviewer for his/her objective evaluation and valuable comments on our manuscript. We have improved our work based on the comments proposed by the reviewer, and we hope this revision have well addressed the reviewer's concerns.

*(1) To assess the electrochemical performance of LNMO-based full cells using our designed electrolyte under more rigorous and practical conditions, we adopted the pouch cell configuration for our investigations. Regrettably, since we do not have the capacity to assemble lithium metal pouch cell, we opted for graphite//LNMO pouch cell as an alternative for our studies. The cathode comprises LNMO, PVDF (Kynar® HSV 1800), and carbon black (Super P) in a mass ratio of 97.5:1:1.5. The mass loading of LNMO is about 28 mg/cm², providing an areal capacity of 4 mAh/cm². Meanwhile, the anode is composed of graphite (GCP-80), PVDF (Kynar® HSV 1800), and carbon black (Super P) in a mass ratio of 90:5:5. The mass loading of graphite is around 11.5 mg/cm², yielding an areal capacity of 4.3 mAh/cm². The dimensions of the cathode and anode are 53 mm * 67 mm and 55 mm * 69 mm, respectively. We employed microporous polyethylene (PE) (Celgard 2400) as the separator. The amount of electrolyte added into the pouch cell was 4.2 g/Ah. For assembly, we utilized five cathodes and six anodes to construct the pouch cell. The pouch cell's theoretical capacity is approximately 0.7 Ah, based on the cathode's capacity, and the N/P ratio stands at approximately 1.08.*

First, we conducted an assessment of the electrochemical performance of graphite//LNMO pouch cell using 1 M-LPF-DMDOHD as an electrolyte. As illustrated in Supplementary Figs. 32 and 33, the graphite//LNMO pouch cell employing 1 M-LPF-DMDOHD was able to maintain >63% of its initial capacity after 300 cycles under the testing conditions of 0.3 C charging-0.5 C discharging. It's worth noting that the improvement in the electrochemical performance of graphite//LiNi_{0.5}Mn_{1.5}O₄ full cells when using DMDOHD as the single electrolyte solvent appears to be less significant compared to Li//LiNi_{0.5}Mn_{1.5}O₄ cells.

As previously mentioned in the manuscript, this difference may be attributed to the distinct requirements for SEI properties on lithium metal and graphite surfaces. The graphite//LiNi_{0.5}Mn_{1.5}O₄ battery system also demands an electrolyte system capable of avoiding solvent co-intercalation. Moreover, it's important to mention that the use of DMDOHD as a single solvent in the electrolyte system results in low ionic conductivity, as also discussed earlier. This limitation serves as a discouraging factor for its application as the sole electrolyte solvent in practical scenarios. Hence, we adopted 2,2,2-trifluoroethyl methyl carbonate (FEMC, $\eta=1$ mPa·s) as the main solvent, FEC as the co-solvent and 1,3-propanesultone (PS) as an electrolyte additive to establish a base electrolyte system (1 M-FEC/FEMC (1/9 vol) + 4 wt.% PS) and then explored the potential of DMDOHD to enhance the electrochemical performance of graphite//LiNi_{0.5}Mn_{1.5}O₄ pouch cells when employed as an electrolyte co-solvent. As depicted in Figure 10d and Supplementary Fig. 34, the cycling stability of graphite//LNMO pouch cells exhibited significant improvement as the volume ratio of DMDOHD increased from 0% to 30% within the FEC/FEMC-based electrolyte system. After 300 cycles under the test condition of 0.5 C charging-1 C discharging, the discharge capacity can still be maintained at more than 94% of the initial capacity. Even when the volume ratio of DMDOHD reached 40%, the cycling stability remained superior to that of the system without DMDOHD, albeit with a reduction compared to the 30% volume ratio system. Additionally, we conducted rate capability tests on the graphite//LNMO pouch cell with a 30% volume ratio of DMDOHD in the FEC/FEMC-based electrolyte system, varying the discharge rate from 0.1 C to 2.0 C. The results, shown in Supplementary Fig. 35, reveal the outstanding rate capability of the graphite//LNMO pouch cell with this electrolyte composition. Specifically, the rate capability ($Q_{2C}/Q_{0.1C}$) reached 93%, and the discharge capacity at 2 C was approximately 0.589 Ah, indicating an impressive utilization rate of 84% at a high current rate. The above results fully demonstrate the practical application value of DMDOHD as an electrolyte co-solvent.

Supplementary Fig. 32 Cycling stability of the graphite//LNMO pouch cell with 1 M-LPF-DMDOHD electrolyte (0.3 C charging-0.5 C discharging) at 25 °C. A 0.3 C constant current and 4.85 V constant voltage (CC-CV) mode was adopted for the charging sequence while a 0.5 C constant current (CC) mode was adopted for the discharging sequence.

Supplementary Fig. 33 Selected voltage profiles of the graphite//LNMO pouch cell with 1 M-LPF-DMDOHD electrolyte (0.3 C charging-0.5 C discharging) at 25 °C. A 0.3 C constant current and 4.85 V constant voltage (CC-CV) mode was adopted for the charging sequence while a 0.5 C constant current (CC) mode was adopted for the discharging sequence.

Figure 10d. Cycling stability of the graphite//LNMO pouch cell in different electrolytes (0.5 C charging-1 C discharging) at 25 °C. A 0.5 C constant current and 4.85 V constant voltage (CC-CV) mode was adopted for all the charging sequences while a 1 C constant current (CC) mode was adopted for all the discharging sequences. FEMC and PS represent 2,2,2-trifluoroethyl methyl carbonate and 1,3-propanesultone, respectively.

Supplementary Fig. 34 Selected voltage profiles of the graphite//LNMO pouch cells (a) without and (b) with a 30% volume ratio of DMDOHD in the FEC/FEMC-based electrolyte system (0.5 C charging-1 C discharging) at 25 °C. A 0.5 C constant current and 4.85 V constant voltage (CC-CV) mode was adopted for all the charging sequences while a 1 C constant current (CC) mode was adopted for all the discharging sequences. FEMC and PS represent 2,2,2-trifluoroethyl methyl carbonate and 1,3-propanesultone, respectively.

Supplementary Fig. 35 Rate capability of graphite//LNMO pouch cell with a 30% volume ratio of DMDOHD in the FEC/FEMC-based electrolyte system at 25 °C. A 0.3 C constant current and 4.85 V constant voltage (CC-CV) mode was adopted for all the charging sequences. Voltage profiles were plotted with cell capacity. FEMC and PS represent 2,2,2-trifluoroethyl methyl carbonate and 1,3-propanesultone, respectively.

(2) In fact, the reason pouch cells can achieve an active material mass loading close to or greater than 25 mg/cm² is that we can use aluminum foil coated with active material on both sides as the cathode, with each side having an active material mass loading of around 12.5 mg/cm². Therefore, our single-sided active material mass loading of 10 mg/cm² is quite comparable to the single-sided active material mass loading of a pouch cell. However, in our previous full cells, the lithium foil thickness was around 1 mm, resulting in an N/P ratio exceeding 100, which deviates significantly from the practical application conditions (where the N/P ratio is approximately 1). Due to our current limitations in preparing lithium metal pouch cells, to create more rigorous and practical testing conditions, we have focused on maximizing the single-sided active material mass loading of the cathode in the coin cell configuration (e.g., 16 mg/cm², with a theoretical areal capacity of about 2.3 mAh/cm²) and reducing the thickness of the lithium metal anode (e.g., 50 μm, with a theoretical areal capacity of about 10.3 mAh/cm²) to reduce the N/P ratio. In the revised manuscript, we have conducted an exploration of the electrochemical performance of LNMO-based lithium metal batteries utilizing our designed electrolyte under different N/P ratios, including values greater than 50 and approximately 4.5.

As depicted in Supplementary Fig. 13, when operating at an N/P ratio exceeding 50, the DMDOHD-based electrolyte displays excellent cycling stability with a capacity retention of ~93% over more than 100 cycles, and an average CE close to 100% at a rate of C/5, which is considerably better when compared to stability attained with 1 M-LPF-EC/DMC (~79% capacity retention after 70 cycles) electrolyte system. Even at a substantially lower N/P ratio of approximately 4.5, the Li||LiNi_{0.5}Mn_{1.5}O₄ cell operated with the 1 M-LPF-DMDOHD electrolyte manages to uphold a high-capacity retention ratio of approximately 85% after 78 cycles (Supplementary Fig. 14). In contrast, the cell employing the 1 M-LPF-EC/DMC electrolyte system retains only about 38% of its initial capacity after the same number of cycles. These results consolidate the findings on the better stability of the DMDOHD-based electrolyte and its great application potential in actual high-voltage LMB systems.

The corresponding texts and results have been inserted in the suitable positions in our revised manuscript and Supporting Information.

Supplementary Fig. 13 (a, b) Galvanostatic charge-discharge plots at different cycle indices for the Li||LiNi_{0.5}Mn_{1.5}O₄ cell cycled with (a) 1 M-LPF-EC/DMC and (b) 1 M-LPF-DMDOHD electrolytes at a rate of C/5. (c) Cycling stability of the Li||LiNi_{0.5}Mn_{1.5}O₄ cells in different

electrolytes (C/5 cycling rate). The mass loading of the $\text{LiNi}_{0.5}\text{Mn}_{1.5}\text{O}_4$ cathode material is around 16 mg/cm^2 , with an N/P ratio exceeding 50.

Supplementary Fig. 14 (a, b) Galvanostatic charge-discharge plots at different cycle indices for the $\text{Li}||\text{LiNi}_{0.5}\text{Mn}_{1.5}\text{O}_4$ cells cycled with (a) 1 M-LPF-EC/DMC and (b) 1 M-LPF-DMDOHD electrolytes at a rate of C/5. (c) Cycling stability of the $\text{Li}||\text{LiNi}_{0.5}\text{Mn}_{1.5}\text{O}_4$ cells in different electrolytes (C/5 cycling rate). The mass loading of the $\text{LiNi}_{0.5}\text{Mn}_{1.5}\text{O}_4$ cathode material is around 16 mg/cm^2 , with an N/P ratio of approximately 4.5.

Reviewer #3:

Comments:

In this work the authors have introduced the use of an electrolyte system compatible with high-voltage 5 V-class Lithium metal batteries. The use of dimethyl 2,5-dioxahexanedioate (DMDOHD) solvent with a LiPF₆ has been shown to have several advantages as compared to the traditional carbonate-based electrolytes in terms of superior lithium plating/stripping characteristics, Coulombic efficiency; suppression of dendrites and dense lithium deposition with an anodic stability, up to 5.5 V vs.Li/Li⁺.

Enhanced performance of cells with Li metal anode and LNMO cathodes supports the claim owing to the electrolyte's enhanced interfacial forming capability and solvation structure. The authors have also acknowledged that the solvent has been previously tested as an additive in previous works. However it is the first time DMDOHD has been used as a single solvent for a 5V class cathode.

Response: We thank the reviewer for the objective assessment and insightful comments on our manuscript, which we respond to below. We have improved our manuscript based on the following detailed comments proposed by the reviewer. We hope this revision will meet the reviewer's requirements.

1. Since the authors have managed to bring back the use of carbonate based solvents for use in Li metal anode batteries, I would insist that the authors take this opportunity to test these electrolytes in the much vaunted Li/S batteries and the stability of such electrolytes towards polysulphides. I therefore suggest the authors to add some comments or run some preliminary test and add their views in the manuscript.

Response: We thank the reviewer for his/her helpful suggestions. Following the reviewer's suggestion, we evaluated the electrochemical performance of the DMDOHD electrolyte in lithium-sulfur battery system.

Interestingly, as depicted in Supplementary Fig. 31, when operating at an exceptionally low C-rate of 0.02 C and discharging to approximately 1 V (vs. Li/Li⁺), the lithium-sulfur cell with the 1 M-LPF-DMDOHD electrolyte was found able to attain a specific discharge capacity of approximately 1360 mAh/g during discharge, accompanied by a typical "two-plateau"

feature, which corresponds to the successive reduction of elemental sulfur to long-chain polysulfides and subsequently to short-chain lithium sulfide. This behavior stands in stark contrast to the lithium-sulfur battery employing the conventional carbonate electrolyte (1 M-LPF-EC/DMC), which exhibits only one plateau in the high voltage region (~2.4 V vs. Li/Li⁺) associated to the reduction of elemental sulfur to long-chain polysulfides, and a considerably lower specific capacity of approximately 565 mAh/g. However, similar to the conventional carbonate electrolyte, our designed carbonate-based electrolytes also failed to facilitate a reversible electrochemical redox reaction of elemental sulfur, even after charging up to 3.5 V (vs. Li/Li⁺).

In the authors' perspective, although the DMDOHD-based electrolyte system demonstrates favorable compatibility with lithium metal and has the potential to slow down the dissolution of polysulfides given higher viscosity (thereby reducing the shuttling rate of the polysulfides), it remains ineffective in inhibiting the dissolution of polysulfides and defending against nucleophilic attacks by polysulfides. Therefore, akin to conventional carbonate-based electrolytes,^[1,2] DMDOHD can undergo chemical reactions with polysulfides (e.g., polysulfides react with DMDOHD via nucleophilic addition or substitution reactions), resulting in an irreversible sulfur redox reaction within the lithium-sulfur battery system.

The corresponding texts and results have been inserted in the suitable positions in our revised manuscript and Supporting Information.

Supplementary Fig. 31 Galvanostatic charge-discharge plots at different cycle indices for the Li||S/rGO cells cycled with (a, c) 1 M-LPF-EC/DMC and (b, d) 1 M-LPF-DMDOHD electrolytes at a rate of (a, b) C/50 and (c, d) C/10 in different voltage windows (1 C corresponds to 1675 mA g^{-1}).

References

- [1] Gao, J., Lowe, M. A., Kiya, Y., & Abruna, H. D. (2011). Effects of liquid electrolytes on the charge-discharge performance of rechargeable lithium/sulfur batteries: electrochemical and in-situ X-ray absorption spectroscopic studies. *The Journal of Physical Chemistry C*, 115(50), 25132–25137.
- [2] Yim, T., Park, et al. (2013). Effect of chemical reactivity of polysulfide toward carbonate-based electrolyte on the electrochemical performance of Li-S batteries. *Electrochimica Acta*, 107, 454–460.

2. CEI composition schematic needs to be provided for better understanding the difference of the CEI composition with the different electrolytes involved.

Response: We thank the reviewer for his/her good suggestion. According to the reviewer's suggestion, we have provided schematic diagrams illustrating the differences in chemical composition and morphological structure of the cathode-electrolyte interphases (CEIs) generated on the cycled $\text{LiNi}_{0.5}\text{Mn}_{1.5}\text{O}_4$ cathode materials in the three studied electrolyte systems (See Supplementary Fig. 28).

As displayed in Figure 8a, Supplementary Figs. 25–27 and Supplementary Table 10, qualitatively, the composition of the CEI formed in the three electrolytes is similar, including the $-\text{CF}_3$, $\text{Li}_x\text{PO}_y\text{F}_z$, LiF , $\text{C}-\text{C}$, and organic/inorganic oxide species (e.g., Li_2CO_3 , $-\text{OCO}_2\text{Li}$, $\text{C}=\text{O}$, $\text{C}-\text{O}$, and $\text{M}-\text{O}$). The content of these however is significantly different. For example, the CEI formed in the 1 M-LPF-DMDOHD electrolyte possesses a considerably higher amount of $-\text{CF}_3$ and $\text{Li}_x\text{PO}_y\text{F}_z$ components compared to the one formed in the two baseline electrolytes (Supplementary Fig. 27 and Supplementary Table 10), which, as discussed earlier, displays higher anodic stability. The high-resolution TEM images (Figs. 8b–d) and energy-dispersive X-ray (EDX) mapping (Figs. 8e–h) further reveal that the CEI formed in the 1 M-LPF-DMDOHD electrolyte is also more uniform, dense and conformal (~ 5 nm), as compared to one formed in 1 M-LPF-DMC and 1 M-LPF-EC/DMC electrolytes (probably formed through continuous formation, and partial dissolution of CEI constituents, leading to the porous structure). The compositional and structural superiority of the CEI formed in the 1 M-LPF-DMDOHD electrolyte (Supplementary Fig. 28) explains thus the excellent cycling stability analyses of the $\text{LiNi}_{0.5}\text{Mn}_{1.5}\text{O}_4$ electrodes (Figs. 6c, d, and Supplementary Fig. 10), demonstrating again the advantages of the DMDOHD-based electrolyte.

Supplementary Fig. 28 Schematics showing the chemical composition and morphological structure of the cathode-electrolyte interphase (CEI) generated on the cycled $\text{LiNi}_{0.5}\text{Mn}_{1.5}\text{O}_4$ cathode materials in different electrolytes.

3. The flame test is not convincing to conclude that these electrolytes are safe and non flammable as the video shows a different scenario.

Response: We totally agree with the reviewer's viewpoints and thank the reviewer for his/her insightful comment. Based on the results of thermogravimetric analysis (TGA, Fig. 2c) and differential scanning calorimetry (DSC, Supplementary Fig. 7), it is evident that the DMDOHD-based electrolyte exhibits superior thermal stability (higher boiling point, lower vapor pressure) compared to DMC and EC/DMC-based electrolytes. Furthermore, Supplementary Videos 1 and 2 distinctly demonstrate that the DMDOHD-based electrolyte is notably more resistant to combustion than the EC/DMC-based electrolyte when subjected to the same flame exposure conditions. Consequently, these results allow us to conclude that the safety of the DMDOHD-based electrolyte is relatively better than that of the commercial EC/DMC-based electrolyte.

Subsequently, we conducted a more rigorous flame test – utilizing a more violent flame and exposing the electrolyte to the flame for an extended duration (Supplementary Videos 3 and 4). This extended evaluation revealed that the DMDOHD-based electrolyte system does exhibit flammability under these intensified conditions, thus highlighting the need for further safety measures. Therefore, in our initial manuscript, we refrained from asserting that the

DMDOHD-based electrolyte is safe and non-flammable, and instead, our safety assessment was grounded in a comparative analysis with the EC/DMC-based electrolyte. However, the ambiguity may have arisen due to the lack of clear statement in the abstract in our initial manuscript. To avoid this, we have made necessary modifications to the abstract text. We hope that our modification can meet the reviewer's requirement.

4. I do not find a c-rate capability test for the electrolytes under study. This needs to be incorporated in the manuscript or supplementary info to have an idea about the recovery of specific capacity.

Response: We thank the reviewer for his/her good suggestion. Based on the reviewer's suggestion, we have included the rate performance of the studied electrolyte in the supplementary information. As depicted in Supplementary Fig. 12, the Li//LiNi_{0.5}Mn_{1.5}O₄ cell equipped with the 1 M-LPF-DMDOHD electrolyte, although exhibiting poor electrochemical performance at large C-rates (> 1 C), was still able to return to a discharge capacity close to that of the initial capacity when the C-rate reverted back to a small C-rate of 0.2 C, which demonstrated its good capacity recovery capability.

*Given that electrolyte systems using DMDOHD as a single solvent exhibit low ionic conductivity and moderate rate performance, the practicality of employing DMDOHD as the single electrolyte solvent may appear doubtful. To further clarify whether DMDOHD has any value for practical application, we explore its potential to enhance the electrochemical performance of LNMO-based full cells when employed as an electrolyte co-solvent under more rigorous and practical conditions. Here, we adopted the pouch cell configuration for our investigations. Regrettably, since we do not have the conditions to prepare lithium metal pouch cell, we opted for graphite//LNMO pouch cell as an alternative for our studies. The cathode comprises LNMO, PVDF (Kynar® HSV 1800), and carbon black (Super P) in a mass ratio of 97.5:1:1.5. The mass loading of LNMO is about 28 mg/cm², providing an areal capacity of 4 mAh/cm². Meanwhile, the anode is composed of graphite (GCP-80), PVDF (Kynar® HSV 1800), and carbon black (Super P) in a mass ratio of 90:5:5. The mass loading of graphite is around 11.5 mg/cm², yielding an areal capacity of 4.3 mAh/cm². The dimensions of the cathode and anode are 53 mm * 67 mm and 55 mm * 69 mm, respectively. We employed microporous*

polyethylene (PE) (Celgard 2400) as the separator. The amount of electrolyte added into the pouch cell was 4.2 g/Ah. For assembly, we utilized five cathodes and six anodes to construct the pouch cell. The pouch cell's theoretical capacity is approximately 0.7 Ah, based on the cathode's capacity, and the N/P ratio stands at approximately 1.08.

As depicted in Figure 10d and Supplementary Fig. 34, the cycling stability of graphite//LNMO pouch cells exhibited significant improvement as the volume ratio of DMDOHD increased from 0% to 30% within the FEC/FEMC-based electrolyte system. After 300 cycles under the test condition of 0.5 C charging-1 C discharging, the discharge capacity can still be maintained at more than 94% of the initial capacity. Even when the volume ratio of DMDOHD reached 40%, the cycling stability remained superior to that of the system without DMDOHD, albeit with a reduction compared to the 30% volume ratio system. Additionally, we conducted rate capability tests on the graphite//LNMO pouch cell with a 30% volume ratio of DMDOHD in the FEC/FEMC-based electrolyte system, varying the discharge rate from 0.1 C to 2.0 C. The results, shown in Supplementary Fig. 35, reveal the outstanding rate capability of the graphite//LNMO pouch cell with this electrolyte composition. Specifically, the rate capability ($Q_{2C}/Q_{0.1C}$) reached 93%, and the discharge capacity at 2 C was approximately 0.589 Ah, indicating an impressive utilization rate of 84% at a high current rate. The above results fully demonstrate the practical application value of DMDOHD as an electrolyte co-solvent.

Supplementary Fig. 12 (a) Galvanostatic charge-discharge plots at different current rates and (b) rate performance of the Li//LiNi_{0.5}Mn_{1.5}O₄ cell cycled with 1 M-LPF-DMDOHD electrolyte.

Figure 10d. Cycling stability of the graphite//LNMO pouch cells in different electrolytes (0.5 C charging-1 C discharging) at 25 °C. A 0.5 C constant current and 4.85 V constant voltage (CC-CV) mode was adopted for all the charging sequences while a 1 C constant current (CC) mode was adopted for all the discharging sequences. FECM and PS represent 2,2,2-trifluoroethyl methyl carbonate and 1,3-propanesultone, respectively.

Supplementary Fig. 34 Selected voltage profiles of the graphite//LNMO pouch cells (a) without and (b) with a 30% volume ratio of DMDOHD in the FEC/FEMC-based electrolyte system (0.5 C charging-1 C discharging) at 25 °C. A 0.5 C constant current and 4.85 V constant voltage (CC-CV) mode was adopted for all the charging sequences while a 1 C constant current (CC) mode was adopted for all the discharging sequences. FECM and PS represent 2,2,2-trifluoroethyl methyl carbonate and 1,3-propanesultone, respectively.

Supplementary Fig. 35 Rate capability of graphite//LNMO pouch cell with a 30% volume ratio of DMDOHD in the FEC/FEMC-based electrolyte system at 25 °C. A 0.3 C constant current and 4.85 V constant voltage (CC-CV) mode was adopted for all the charging sequences. Voltage profiles were plotted with cell capacity. FEMC and PS represent 2,2,2-trifluoroethyl methyl carbonate and 1,3-propanesultone, respectively.

REVIEWER COMMENTS

Reviewer #1 (Remarks to the Author):

Authors have responded nicely to my comments. The paper is acceptable now.

Reviewer #3 (Remarks to the Author):

The authors have successfully made necessary amendments in the manuscript and SI according to the suggestions provided and the effort taken is very well appreciated.

The interesting part/novelty of this work lies in the results on DMDOHD as a single solvent while additive based approach has already been made by Dahn and co workers. Therefore I would suggest that the authors bear this in mind while reporting their work as some of the modifications made in response to reviewers comments is vice versa.

Reviewer #4 (Remarks to the Author):

The pursuit of high energy density lithium metal batteries requires electrolytic liquid systems that can be stable and match the positive electrode of high operating voltage. However, the current electrolytic liquid system and the common high pressure cathode materials will always have unwanted side reactions, which will affect the battery performance. An effective way to solve the above problems is through electrolyte engineering, especially the development of new electrolytes. In this work, the authors demonstrate a new electrolytic liquid system, i.e. 1M LiPF₆ in DMDOHD. The excellent electrochemical performance of the electrolytic liquid system at high voltage was confirmed by systematic study and characterization. Although the ionic conductivity at room temperature is not satisfactory, it still has great reference value in exploring more practical electrolytes for lithium metal batteries. Consequently, I recommend it to be accepted for publication in Nature Communications after addressing some major points.

1. The authors believe that DMDOHD has the same good film forming potential as EC only from the perspective of reduction potential and structure, which may not be rigorous enough. In previous reports, commercial electrolytic liquid systems (EC-DMC mixtures), interface components are mainly generated by EC decomposition. However, the oxidation potential of EC is actually greater than that of DMC. The more common explanation is that because EC has a high dielectric constant (high polarity), it will rapidly enrich to the electrode surface at high motor potential and thus preferentially decompose. According to the calculation data provided in the author's SI, DMDOHD has a very low dielectric constant, which is the

opposite of EC. Therefore, although the single salt single solvent system in this paper, according to the above viewpoint, it is not rigorous enough to infer that DMDOHD also has good film forming characteristics at the beginning of this paper only based on the structure and similar reduction potential.

2.The DMDOHD has been reported to be used in combination with LiFSI to improve lithium-ion battery performance (voltage < 4V vs. Li+/Li). However, replacing the lithium salt with LiPF₆ was able to boost the battery voltage to 5V vs. Li+/Li. The authors did not characterize whether the oxidation stability of the DMDOHD solvent itself was the dominant factor.

3.The selection of the implicit solvent model is well described in a recent article on the calculation of the redox potential of solvent molecules in electrolytes using Nernst equation theory, which may be very helpful for the author's explanation in the method section [Adv. Funct. Mater. 2023, 33(11)].

4.As for the electrolyte safety test, I suggest that the author draw a graph of the self-extinction time (SET) of the electrolyte, which will more clearly compare the safety differences of different electrolytic liquid systems.

5.In the study of further improvement of high-pressure LM and Li-ion batteries containing cosolvents and additives, the coulomb efficiency of 1M-LPF-DMDOHD/FEC (4:1, v:v) electrolyte decreased instead, what is the possible reason for this?

6.In the Introduction, "Finally, in the final preparation stages of this manuscript, J. Dahn and his team reported on the use of DMDOHD electrolyte solvent with LiFSI salt for 4 V class Li-ion cells such as graphite | LiFePO₄, graphite | Li[Ni_{0.5}Mn_{0.3}Co_{0.2}]O₂, and graphite | Li[Ni_{0.83}Mn_{0.06}Co_{0.11}]O₂.", please check "graphite | Li[Ni_{0.83}Mn_{0.06}Co_{0.11}]O₂."

7.The author is too vague when comparing the ionic conductivity of electrolyte with ionic liquid, solid ceramic and polymer electrolytes used in LMB or LIBs. I suggest that the author list specific substances and their properties to clarify this point.

Reviewer #1:

Authors have responded nicely to my comments. The paper is acceptable now.

Response: We thank the reviewer for his/her appreciation of our revised manuscript and the conclusion that our manuscript is now acceptable for publication in Nature Communications.

Reviewer #3:

The authors have successfully made necessary amendments in the manuscript and SI according to the suggestions provided and the effort taken is very well appreciated.

The interesting part/novelty of this work lies in the results on DMDOHD as a single solvent while additive based approach has already been made by Dahn and co-workers. Therefore, I would suggest that the authors bear this in mind while reporting their work as some of the modifications made in response to reviewers comments is vice versa.

Response: We thank the reviewer for his/her appreciation of our revised manuscript. We totally agree with the reviewer's suggestions, and we recognize that the interesting and novel aspect of this work is the excellent results attained by using DMDOHD as a single solvent while additive based approach has already been made by Dahn and co-workers. We will bear this in mind while reporting our work, and will continue to dedicate our efforts toward the future development of single-solvent, single-salt electrolyte systems.

Reviewer #4:

The pursuit of high energy density lithium metal batteries requires electrolytic liquid systems that can be stable and match the positive electrode of high operating voltage. However, the current electrolytic liquid system and the common high pressure cathode materials will always have unwanted side reactions, which will affect the battery performance. An effective way to solve the above problems is through electrolyte engineering, especially the development of new electrolytes. In this work, the authors demonstrate a new electrolytic liquid system, i.e. 1M LiPF₆ in DMDOHD. The excellent electrochemical performance of the electrolytic liquid system at high voltage was confirmed by systematic study and characterization. Although the ionic conductivity at room temperature is not satisfactory, it still has great reference value in exploring more practical electrolytes for lithium metal batteries. Consequently, I recommend it to be accepted for publication in Nature Communications after addressing some major points.

Response: We thank the reviewer for the positive assessment and insightful comments on our manuscript, which we respond to below. We have improved our manuscript based on the following detailed comments proposed by the reviewer. We hope this revision will meet the reviewer's requirements.

1. The authors believe that DMDOHD has the same good film forming potential as EC only from the perspective of reduction potential and structure, which may not be rigorous enough. In previous reports, commercial electrolytic liquid systems (EC-DMC mixtures), interface components are mainly generated by EC decomposition. However, the oxidation potential of EC is actually greater than that of DMC. The more common explanation is that because EC has a high dielectric constant (high polarity), it will rapidly enrich to the electrode surface at high motor potential and thus preferently decompose. According to the calculation data provided in the author's SI, DMDOHD has a very low dielectric constant, which is the opposite of EC. Therefore, although the single salt single solvent system in this paper, according to the above viewpoint, it is not rigorous enough to infer that DMDOHD also has good film forming characteristics at the beginning of this paper only based on the structure and similar reduction potential.

Response: We thank the reviewer for his/her insightful comments and we totally agree with the reviewer's viewpoints. The film forming ability actually encompasses both the solid-electrolyte interphase (SEI) film forming ability and cathode-electrolyte interphase (CEI) film forming ability. Our prior focus centered on predicting and elucidating the SEI film forming capability of DMDOHD, while inadvertently overlooked the prediction and discussion of its CEI film forming ability at the beginning of our paper. In response, we provide additional insights in this context.

Prior studies reveal that the composition of the CEI is primarily governed by the prevalence of species within the inner Helmholtz layer.^[1] Typically, during the charging phase (i.e., when the voltage rises), anions such as PF_6^- and solvent molecules with high dielectric constant (i.e., high polarity) such as EC, tend to accumulate on the positive electrode surface propelled by elevated electric fields.^[2] Researches by Borodin and Xu et al. indicates that, at low salt concentrations, only approximately 30% of the inner Helmholtz layer is occupied by anions.^[3,4] Consequently, for the 1 M $\text{LiPF}_6/\text{EC-DMC}$ system, the species on the positive electrode surface primarily include free EC solvent molecules, free PF_6^- anions, and EC/PF_6^- complexes. Numerous investigations establish that the anodic stability of free PF_6^- anion and free EC solvent molecule exceeds 6 V (vs. Li/Li^+).^[3,5-7] However, the coordination of EC with PF_6^- anions to form an EC/PF_6^- complex significantly reduces the oxidation potential.^[3,5-7] Additionally, when EC undergoes dehydrogenation and decomposition into CO_2 and OC_2H_3 through ring opening, the oxidation potential decreases to approximately 4.5 V (vs. Li/Li^+).^[3,5-7] Hence, for the 1 M $\text{LiPF}_6/\text{EC-DMC}$ system, CEI primarily results from the ring-opening dehydrogenation decomposition of free EC molecules and the oxidation of the EC/PF_6^- complex, encompassing the decomposition of a majority of EC and a minor fraction of PF_6^- .^[1,2] The constituents of the resulting CEI predominantly comprise organic species generated through the decomposition of EC.^[1,2] However, organic SEI is typically characterized as less dense (thick and porous) and less robust.^[1,8] This characteristic not only fails to effectively suppress the irreversible oxidation of the electrolyte under high voltage but also impedes the transport of lithium ions, posing challenges to the stable cycling of high-voltage positive electrodes.^[1,8]

However, in contrast to EC ($\epsilon=89.6$), DMDOHD ($\epsilon=2.9$) will not enrich on the positive electrode surface during charging due to its significantly lower dielectric constant. Consequently, during the charging process in the DMDOHD-based electrolyte system,

primarily anions such PF_6^- accumulate on the positive electrode surface. However, researches by Borodin and Xu et al. suggests that PF_6^- anions can only partially replace solvent molecules in the inner Helmholtz layer,^[3,4] leaving some DMDOHD solvent molecules near the positive electrode surface. As previously discussed, the oxidation potential of free PF_6^- anions and free solvent molecules are generally higher, while solvent/ PF_6^- complexes have lower oxidation potentials.^[3,5-7] This observation is supported by our calculation results, as indicated in Supplementary Table 6. The oxidation stability of free DMDOHD and PF_6^- anions is higher than 6 V (vs. Li/Li^+), specifically 6.40 and 8.97 V (vs. Li/Li^+), respectively, while the oxidation potential of the DMDOHD/ PF_6^- complex is significantly reduced to about 5.05 V, a value closely aligned with the experimental measurement (4.8 V vs. Li/Li^+) conducted in our experiments (linear scan voltammetry in Supplementary Figs. 5 and 6). Considering the possible existing discrepancy between the theoretically calculated potential value and the practical observation, we attribute the onset potential of 4.8 V (vs. Li/Li^+) in the linear scan voltammetry plots (Supplementary Figs. 5 and 6) to the oxidative decomposition of the DMDOHD/ PF_6^- complex. Therefore, in the 1 M $\text{LiPF}_6/\text{DMDOHD}$ system, the CEI predominantly forms through the oxidative decomposition of the DMDOHD/ PF_6^- complex. In comparison with the 1 M $\text{LiPF}_6/\text{EC-DMC}$ system, the CEI in the 1 M $\text{LiPF}_6/\text{DMDOHD}$ system features a lower anodic decomposition contribution from free DMDOHD solvent molecules. Consequently, the content of inorganic species and fluorine species in the CEI formed by the 1 M $\text{LiPF}_6/\text{DMDOHD}$ system is higher than that in the 1 M $\text{LiPF}_6/\text{EC-DMC}$ system. This will promote the formation of a denser, more uniform, and robust CEI, which is able to effectively suppress the electrolyte decomposition and maintain the stability of the positive electrode/electrolyte interface.^[1,8-10] This characteristic will endow the positive electrode material with the ability to cycle stably over an extended period, consistent with subsequent experimental findings.^[1,8-10]

In summary, the film forming ability of DMDOHD can be effectively categorized into two aspects. Firstly, based on its structural similarity to LEDC – the primary component of the SEI in EC-based electrolytes, and its comparable reduction potential to EC, we can infer its SEI film forming ability. Secondly, considering its low dielectric constant and the oxidation potential of its complex with PF_6^- anions, we can deduce its capability to participate in CEI film formation and to increase the content of inorganic species and fluorine species in the CEI.

The corresponding discussions have been inserted in the suitable positions in our revised

Supplementary Table 6. The corresponding oxidation potential of the studied DMDOHD, PF_6^- and DMDOHD/ PF_6^- complex calculated by using the Nernst equation.

Species	Oxidation potential (vs. Li/Li^+)
DMDOHD/ PF_6^-	5.05
DMDOHD	6.40
PF_6^-	8.97

References

- [1] Liu, J., et al. (2022). Tuning Interphase Chemistry to Stabilize High-Voltage LiCoO_2 Cathode Material via Spinel Coating. *Angew. Chem. Int. Ed.*, 61(35), e202207000.
- [2] Yu, L., et al. (2013). Preferential adsorption of solvents on the cathode surface of lithium ion batteries. *Angew. Chem. Int. Ed.*, 22(52), 5753–5756.
- [3] Borodin, O., et al. (2017). Modeling insight into battery electrolyte electrochemical stability and interfacial structure. *Acc. Chem. Res.*, 50(12), 2886–2894.
- [4] Vatamanu, J., Borodin, O., & Smith, G. D. (2012). Molecular dynamics simulation studies of the structure of a mixed carbonate/ LiPF_6 electrolyte near graphite surface as a function of electrode potential. *J. Phys. Chem. C*, 116(1), 1114–1121.
- [5] Jow, T. R., Xu, K., Borodin, O., & Ue, M. (Eds.). (2014). *Electrolytes for lithium and lithium-ion batteries* (Vol. 58, p. 476). New York: Springer.
- [6] Wu, F., Borodin, O., & Yushin, G. (2017). In situ surface protection for enhancing stability and performance of conversion-type cathodes. *MRS Energy & Sustainability*, 4, E9.
- [7] Borodin, O., Behl, W., & Jow, T. R. (2013). Oxidative stability and initial decomposition reactions of carbonate, sulfone, and alkyl phosphate-based electrolytes. *J. Phys. Chem. C*,

117(17), 8661–8682.

[8] Fan, X., & Wang, C. (2021). High-voltage liquid electrolytes for Li batteries: progress and perspectives. *Chem. Soc. Rev.*, 50(18), 10486–10566.

[9] Fan, X., et al. (2018). Non-flammable electrolyte enables Li-metal batteries with aggressive cathode chemistries. *Nat. Nanotech.*, 13(8), 715–722.

[10] Xu, K. (2014). Electrolytes and interphases in Li-ion batteries and beyond. *Chem. Rev.*, 114(23), 11503–11618.

2. The DMDOHD has been reported to be used in combination with LiFSI to improve lithium-ion battery performance (voltage < 4V vs. Li⁺/Li). However, replacing the lithium salt with LiPF₆ was able to boost the battery voltage to 5V vs. Li⁺/Li. The authors did not characterize whether the oxidation stability of the DMDOHD solvent itself was the dominant factor.

Response: We thank the reviewer for his/her insightful comments.

Firstly, for Jeff Dahn's paper,^[11] they aimed to enhance the cycle life of positive electrode materials (LiFePO₄, Li[Ni_{0.5}Mn_{0.3}Co_{0.2}]O₂, Li[Ni_{0.83}Mn_{0.06}Co_{0.11}]O₂) under elevated temperatures, reaching up to 85 °C. Consequently, they opted for a more thermally stable LiFSI over LiPF₆ lithium salt. Nevertheless, it is widely acknowledged that LiFSI can cause severe corrosion to the aluminum collector at high voltage in a low-concentration electrolyte system. To prevent side reactions, such as aluminum corrosion, they opted to restrict the operating voltage to below 4 V (vs. Li/Li⁺), aiming for an extended cycle life. Therefore, in their study, the operational voltage below 4 V (vs. Li/Li⁺) is not relevant to the oxidative stability of DMDOHD itself.

Secondly, as discussed in our response to Comment 1, the oxidation stability of free DMDOHD and PF₆⁻ anions is higher than 6 V (vs. Li/Li⁺), specifically 6.40 and 8.97 V (vs. Li/Li⁺), respectively, while the oxidation potential of the DMDOHD/PF₆⁻ complex is significantly reduced to about 5.05 V (vs. Li/Li⁺), a value closely aligned with the experimental measurement (4.8 V vs. Li/Li⁺) conducted in our experiments (linear scan voltammetry in

Supplementary Figs. 5 and 6). Considering the possible existing discrepancy between the theoretically calculated potential value and the practical observation, we attribute the onset potential of 4.8 V (vs. Li/Li⁺) in the linear scan voltammetry plots (Supplementary Figs. 5 and 6) to the oxidative decomposition of the DMDOHD/PF₆⁻ complex. Therefore, following the principle of the buckets effect, the anodic stability of the 1 M LiPF₆/DMDOHD system heavily relies on the quality of the CEI formed by the oxidative decomposition of the DMDOHD/PF₆⁻ complex with the lowest oxidation potential. In comparison with the 1 M LiPF₆/EC-DMC system, the CEI in the 1 M LiPF₆/DMDOHD system features a lower oxidative decomposition contribution from free DMDOHD solvent molecules. Consequently, the content of inorganic species and fluorine-containing species in the CEI formed by the 1 M LiPF₆/DMDOHD system is higher than that in the 1 M LiPF₆/EC-DMC system, promoting the formation of a denser, more uniform, and robust CEI, which is able to effectively suppress the electrolyte decomposition and maintain the stability of the positive electrode/electrolyte interface.^[1,8-10] This characteristic will endow the positive electrode material with the ability to cycle stably over an extended period.^[1,8-10]

Thus, the oxidative stability of DMDOHD solvent itself is a contributing factor but not the sole determinant for the 1 M LiPF₆/DMDOHD system's ability to elevate the battery's operating voltage to 5 V (vs. Li/Li⁺). The paramount factor lies in its capacity to form a dense, homogeneous, and robust CEI, which is able to effectively suppress the electrolyte decomposition and maintain the stability of the positive electrode/electrolyte interface, thereby endowing the positive electrode material with the ability to cycle stably over an extended period.

References

[11] Taskovic, T., Eldesoky, A., Aiken, C. P., & Dahn, J. R. (2022). Low-Voltage Operation and Lithium Bis (fluorosulfonyl) imide Electrolyte Salt Enable Long Li-Ion Cell Lifetimes at 85° C. *J. Electrochem. Soc.*, 169(10), 100547.

3. The selection of the implicit solvent model is well described in a recent article on the calculation of the redox potential of solvent molecules in electrolytes using Nernst equation theory, which may be very helpful for the author's explanation in the method section [Adv. Funct. Mater. 2023, 33(11)].

Response: We thank the reviewer for his/her good suggestion. The literature (Adv. Funct. Mater., 2023, 33(11): 2212342) suggested by the reviewer was indeed very helpful in explaining our selection of implicit solvent model in the Methods section. Consequently, we have included the necessary discussion and citation in the Methods section, which has been highlighted in yellow in our revised manuscript.

"Here we chose the polarizable continuum model (IEFPCM) and the solvation model density (SMD) to take into account the solvent effect, where the SMD is considered more suitable for organic solvents used in batteries due to the fact that more accurate results can be attained by taking into account not only the electrostatic interactions between the solute and the solvent in the polarizable continuum model, but also the inclusion of the non-electrostatic interactions (cavity-dispersion-solvent-structure term).⁶³"

4. As for the electrolyte safety test, I suggest that the author draw a graph of the self-extinction time (SET) of the electrolyte, which will more clearly compare the safety differences of different electrolytic liquid systems.

Response: We thank the reviewer for his/her good suggestion. To more clearly compare the safety differences between the two electrolytes, we conducted tests on the self-extinguishing time (SET) of the two electrolytes under three different conditions: (1) Ignition of a suspended separator impregnated with electrolyte (refer to Supplementary Fig. 8 and Supplementary Videos 5 and 6); (2) Ignition of an electrolyte in a stainless-steel coin cell case (refer to Supplementary Fig. 9 and Supplementary Videos 7 and 8); and (3) Ignition of a separator immersed in electrolyte in a stainless-steel coin cell case (refer to Supplementary Fig. 10 and Supplementary Videos 9 and 10). The results, as presented in Supplementary Videos 5–10, Supplementary Figs. 8–10 and Supplementary Table 11, consistently show that the 1 M-LPF-

DMDOHD electrolyte has smaller SET values than the 1 M-LPF-EC/DMC electrolyte, regardless of the test conditions, suggesting its superior safety properties.

The corresponding texts and results have been inserted in the suitable positions in our revised manuscript and Supporting Information.

Supplementary Fig. 8 Flammability tests for the conventional 1 M-LPF-EC/DMC and our 1 M-LPF-DMDOHD electrolytes in the condition of ignition of a suspended separator impregnated with electrolyte. The detailed parameters of the two electrolytes are shown in Supplementary Table 11. The resulting SET of the conventional 1 M-LPF-EC/DMC and our 1 M-LPF-DMDOHD electrolytes is 58.8 and 53.8 s g⁻¹, respectively.

After several ignition attempts within a minute, it still cannot catch fire.

Supplementary Fig. 9 Flammability tests for the conventional 1 M-LPF-EC/DMC and the 1 M-LPF-DMDOHD electrolytes in the condition of ignition of an electrolyte in a stainless-steel coin cell case. The detailed parameters of the two electrolytes are shown in Supplementary Table 11. The resulting SET of the conventional 1 M-LPF-EC/DMC and our 1 M-LPF-DMDOHD electrolytes is 85.4 and 0 s g⁻¹, respectively.

Supplementary Fig. 10 Flammability tests for the conventional 1 M-LPF-EC/DMC and the 1 M-LPF-DMDOHD electrolytes in the condition of ignition of a separator immersed in electrolyte in a stainless-steel coin cell case. The detailed parameters of the two electrolytes are shown in Supplementary Table 11. The resulting SET of the conventional 1 M-LPF-EC/DMC and our 1 M-LPF-DMDOHD electrolytes is 79.5 and 29.2 s g⁻¹, respectively.

Supplementary Table 11. Comparison of the self-extinguishing time (SET) of the 1 M-LPF-DMDOHD and 1 M-LPF-EC/DMC electrolytes under three different conditions.

Conditions	Electrolyte	Volume (μL)	Mass (g)	Burning time (s)	SET (s g^{-1})
Ignition of a suspended separator impregnated with electrolyte	1 M-LPF-EC/DMC	100	0.119	7	58.8
	1 M-LPF-DMDOHD	100	0.13	7	53.8
Ignition of an electrolyte in a stainless-steel coin cell case	1 M-LPF-EC/DMC	600	0.726	62	85.4
	1 M-LPF-DMDOHD	600	0.754	0	0
Ignition of a separator immersed in electrolyte in a stainless-steel coin cell case	1 M-LPF-EC/DMC	600	0.704	56	79.5
	1 M-LPF-DMDOHD	600	0.754	22	29.2

5. In the study of further improvement of high-pressure LM and Li-ion batteries containing cosolvents and additives, the coulomb efficiency of 1M-LPF-DMDOHD/FEC (4:1, v:v) electrolyte decreased instead, what is the possible reason for this?

Response: We thank the reviewer for his/her good question. It is indeed widely acknowledged that incorporating an appropriate quantity of FEC into the electrolyte facilitates the introduction of fluorine-containing components into the solid-electrolyte interface (SEI). The generation of fluorine-containing species contributes to the densification and robustness of the SEI, effectively mitigating the continuous side reactions between the electrolyte and lithium

metal, as well as suppressing the growth of lithium dendrites. Consequently, this enhances the Coulombic efficiency of lithium-metal batteries. However, Manthiram *et al.*^[12] observed a decrease in the Coulombic efficiency of Li-metal batteries with excessively high FEC content. This is primarily attributed to the high dielectric constant and polarity of FEC, similar to EC. FEC exhibits strong lithium-ion solvating ability, and the resulting solvation structure primarily exists as solvent-separated ion pairs (SSIPs). An increase in FEC content in the electrolyte leads to a reduction in the number of contact ion pairs (CIPs) and anion aggregates (AGGs) in the solvation structure. This ultimately results in an increase in organic species and a decrease in inorganic species in the SEI. However, as demonstrated in numerous studies,^[13,14] the diffusion of Li ions through the SEI is hindered by the presence of organic species due to their high bonding affinity, resulting in vertical lithium deposition and perforation the SEI. In addition, the organic SEI also possesses a porous structure that is not dense enough to completely block the diffusion of (fully, or partially solvated Li-ions) and thus further reaction between lithium metal and electrolyte solvent(s).^[13] Consequently, the SEI properties formed in DMDOHD-based electrolytes with excessive FEC remain unsatisfactory in preventing lithium dendritic growth, which may account for the decreased lithium plating/stripping Coulombic efficiency.

The corresponding discussions have been inserted in the suitable positions in our revised Supporting Information.

References

[12] Yi, M., Su, L., & Manthiram, A. (2023). Tuning and understanding the solvent ratios of localized saturated electrolytes for lithium-metal batteries. *J. Mater. Chem. A*, 11(22), 11889–11902.

[13] Liu, S, *et al.* (2021). An inorganic-rich solid electrolyte interphase for advanced lithium-metal batteries in carbonate electrolytes. *Angew. Chem. Int. Ed.*, 60(7), 3661–3671.

[14] Fang, C., *et al.* (2019). Quantifying inactive lithium in lithium metal batteries. *Nature*, 572(7770), 511–515

6. In the Introduction, “Finally, in the final preparation stages of this manuscript, J. Dahn and his team reported on the use of DMDOHD electrolyte solvent with LiFSI salt for 4 V class Li-ion cells such as graphite||LiFePO₄, graphite||Li[Ni_{0.5}Mn_{0.3}Co_{0.2}]O₂, and graphite||Li[Ni_{0.83}Mn_{0.06}Co_{0.11}]O_{2.40}.”, please check “graphite||Li[Ni_{0.83}Mn_{0.06}Co_{0.11}]O_{2.40}.”.

Response: We thank the reviewer for his/her careful review, and we have revised the relevant text as follows:

“Finally, in the final preparation stages of this manuscript, J. Dahn and his team reported on the use of DMDOHD electrolyte solvent with LiFSI salt for 4 V class Li-ion cells such as graphite||LiFePO₄, graphite||Li[Ni_{0.5}Mn_{0.3}Co_{0.2}]O₂, and graphite||Li[Ni_{0.83}Mn_{0.06}Co_{0.11}]O_{2.27}.”

7. The author is too vague when comparing the ionic conductivity of electrolyte with ionic liquid, solid ceramic and polymer electrolytes used in LMB or LIBs. I suggest that the author list specific substances and their properties to clarify this point.

Response: We thank the reviewer for his/her good suggestion. We have provided a Supplementary Table listing and comparing the ionic conductivity of our 1 M-LPF-DMDOHD electrolyte with ionic liquids, solid ceramic and polymer electrolytes used in LMBs or LIBs. As shown in below or Supplementary Table 10 in Supporting Information, the ionic conductivity of the 1 M-LPF-DMDOHD electrolyte (0.21 mS cm⁻¹) is indeed in the range of, or even higher than of many ionic liquids, solid ceramic and polymer electrolytes used in LMBs or LIBs, so that it can still meet the requirement of battery applications at low or moderate C-rates.

Supplementary Table 9. Comparison of the ionic conductivity of our 1 M-LPF-DMDOHD electrolyte with ionic liquids, solid ceramic and polymer electrolytes used in LMBs or LIBs.

Electrolytes	Ionic conductivity (mS cm ⁻¹)	Ref.
1 M LiPF₆ in DMDOHD	0.21 (25 °C)	This work
Solid ceramic		
Li _{6.75} La ₃ Zr _{1.75} Nb _{0.25} O ₁₂	0.092 (25 °C)	Adv. Eng. Mater. , 2019 , 21 , 1900055.
Li _{1.3} Al _{0.3} Ti _{1.7} (PO ₄) ₃	0.0778 (30 °C)	ACS Sustainable Chem. Eng. , 2019 , 7 , 4675.
Li _{1+x} Al _x Ti _{2-x} (PO ₄) ₃	0.052 (25 °C)	Nano Lett. , 2017 , 17 , 3182.
Li ₆ BaLa ₇ Ta ₂ O ₁₂	0.04 (22 °C)	Adv. Funct. Mater. , 2005 , 15 , 107.
Li ₇ La ₃ Zr ₂ O ₁₂	0.244 (25 °C)	Angew. Chem. Int. Ed. , 2007 , 46 , 7778.
Li _{6.28} Al _{0.24} La ₃ Zr ₂ O _{11.98}	0.027 (25 °C)	Mater. Today , 2018 , 21 , 594.
Ga-Li ₇ La ₃ Zr ₂ O ₁₂	0.12 (30 °C)	ACS Appl. Mater. Interfaces , 2019 , 11 , 26920.
Li _{0.34(1)} La _{0.51(1)} TiO _{2.94(2)}	0.02 (25 °C)	Solid State Commun. , 1993 , 86 , 689.
LiPON	0.0012 (25 °C)	Solid State Ion. , 2013 , 253 , 151.
Ionic liquid		
[nOctBu ₃ N] ⁺ [N(CF ₃ SO ₂) ₂] ⁻	0.13 (25 °C)	J. Phys. Chem. B , 1998 , 102 , 8858.

	$[\text{Bu}_3\text{Hex}_4\text{N}]^+[\text{N}(\text{CF}_3\text{SO}_2)_2]^-$	0.16 (25 °C)	J. Phys. Chem. B , 1998 , 102, 8858.
	LiIL-0	0.00530 (25 °C)	
	LiIL-1	0.0439 (25 °C)	
	LiIL-2	0.115 (25 °C)	
	LiIL-3	0.0532 (25 °C)	
	LiIL-4	0.104 (25 °C)	
	LiIL-5	0.0243 (25 °C)	Adv. Energy Mater. , 2023 , 13, 2202974.
	LiIL-6	0.145 (25 °C)	
	LiIL-7	0.187 (25 °C)	
	LiIL-8	0.189 (25 °C)	
	LiIL-9	0.0813 (25 °C)	
	LiIL-10	0.0940 (25 °C)	
Polymer electrolyte	PEO+P(STFSiLi)	0.013 (60 °C)	Nat. Mater. , 2013 , 12, 452.
	PDOL GPE	0.0616 (25 °C)	Mater. Today Energy , 2022 , 26, 100984.

poly(alkyl fluoroacrylate)	0.102 (25 °C)	ACS Appl. Mater. Interfaces , 2022 , 14 , 21018.
pDOL with Sn(OTf) ₂	0.0616 (25 °C)	Mater. Today Energy , 2022 , 26 , 100984
PVDA	0.033 (25 °C)	J. Electrochem. Soc. , 2022 , 169 , 090509.
Poly(ethylene glycol) methyl ether acrylate	0.1 (40 °C)	J. Am. Chem. Soc. , 2018 , 140 , 82.

REVIEWERS' COMMENTS

Reviewer #4 (Remarks to the Author):

Authors have revised the manuscript point by point. I recommend that the present version be accepted for publication.